# PRPS2 enhances RNA m⁶A methylation by stimulating SAM synthesis through enzyme-dependent and independent mechanisms

Lin Zhang [1,7], Xian Zhao[2,7], Jingyan Hu [1,7], Tingting Li[1], Hong-Zhuan Chen [3], Ao Zhang [4], Hao Wang[5] ✉, Jianxiu Yu [2] ✉ & Liang Zhang [1,6] ✉

Cancer cells exploit altered metabolic pathways to dynamically regulate epigenetic methylation and thus promote tumorigenesis and metastasis. In various human cancers, such as lung adenocarcinoma, the level of a key cellular metabolite, S-adenosylmethionine (SAM), is prominently upregulated for RNA hypermethylation as the methyl donor. However, the specific mechanisms by which cancer cells produce SAM to sustain RNA methylation remain elusive. Here, we demonstrate that PRPS2, a phosphoribosyl pyrophosphate synthetase isoform involved in the first and rate-limiting step of the purine biosynthesis pathway, exhibits distinct oncogenic functionality in regulating RNA methylation, unlike its homolog PRPS1. PRPS2 utilizes four non-conserved key residues to bypass the typical ADP/GDP allosteric feedback inhibition, enabling sustained excess production of newly synthesized ATP. Moreover, PRPS2 stabilizes methionine adenosyltransferase 2 A (MAT2A) through direct interactions to positively stimulate ATP utilization and SAM synthesis for RNA m⁶A specific methylation via the WTAP/METTL3/METTL14 methyltransferase complex, thereby promoting lung tumorigenesis. Our study links nucleotide biosynthesis with RNA epigenetics in cancer progression through the PRPS2-MAT2A-WTAP/METTL3/METTL14 axis, and elucidates both enzyme-dependent and independent functions of PRPS2. These findings have significant implications for developing targeted therapies for cancers associated with PRPS2 abnormalities.

Cancer metabolism intricately influences various cellular processes with distinctive features conducive to tumorigenesis, including biosynthetic, biogenetic, and epigenetic regulations, through the simultaneous engagement of canonical metabolic and nonmetabolic functions of metabolic enzymes[1-3]. In the context of tumorous

epigenetic regulation, abnormal hypermethylation of RNA m⁶A, the most abundant and prevalent internal RNA modification crucial for various RNA processes such as transcription, maturation, translation, splicing, degradation, and metabolism, promotes aberrant cell proliferation, metastasis, and tumorigenesis, marking it as an oncogenic

[1]Department of Pharmacology and Chemical Biology, State Key Laboratory of Systems Medicine for Cancer, School of Medicine, Shanghai Jiao Tong University, Shanghai 200025, China. [2]Department of Biochemistry and Molecular Cell Biology, Shanghai Key Laboratory of Tumor Microenvironment and Inflammation, Shanghai Jiao Tong University School of Medicine, Shanghai 200025, China. [3]Institute of Interdisciplinary Integrative Biomedical Research, Shuguang Hospital, Shanghai University of Traditional Chinese Medicine, Shanghai 201203, China. [4]Pharm-X Center, School of Pharmacy, Shanghai Jiao Tong University, Shanghai 200240, China. [5]The Division of Thoracic Surgery, Zhongshan Hospital, Fudan University, Shanghai 200032, China. [6]Department of Chemical Biology, School of Chemistry and Chemical Engineering, Shanghai Jiao Tong University, Shanghai 200240, China. [7]These authors contributed equally: Lin Zhang, Xian Zhao, Jingyan Hu. ✉e-mail: wang.hao@zs-hospital.sh.cn; Jianxiu.Yu@gmail.com; liangzhang2014@sjtu.edu.cn

signal in malignant tumors[4–9]. Predominantly catalyzed by the S-adenosylmethionine (SAM)-dependent RNA methyltransferase family (comprising the WTAP/METTL3/METTL14 complex or METTL16), the methylation of RNA m6A plays a pivotal role in these processes[10–12]. SAM, the key methyl group donor, is synthesized by the rate-limiting enzyme Methionine Adenosyltransferase 2 A (MAT2A) from methionine (Met) and adenosine triphosphate (ATP) in the methionine metabolic cycle[13]. The overexpression of MAT2A has been observed in various human cancers, including lung cancer, gastric cancer, liver cancer, and leukemia, to meet the biosynthetic demands associated with RNA m6A hypermethylation[14–16]. However, the regulatory mechanisms governing MAT2A in the context of cancer cell RNA m6A hypermethylation remain poorly understood.

Phosphoribosyl pyrophosphate synthetase (PRPS) stands as the first and rate-limiting enzyme in the purine biosynthesis pathway (Fig. 1a)[17,18]. It catalyzes the pyrophosphorylation of D-ribose-5-phosphate (R5P), obtained from the glucose pentose phosphate pathway (PPP), at the 1' hydroxyl group position, yielding the catalytic product 5-phosphoribosyl-α-1-pyrophosphate (PRPP). PRPP serves as an essential metabolic intermediate for purine biosynthesis via both de

novo and salvage pathways, as well as for the synthesis of pyrimidines, amino acids (tryptophan and histidine), and co-factors ($NAD^+$ and $NADP^+$)[19]. Thus, PRPS functions as a crucial "molecular rheostat" that connects predominant nutrient glucose metabolism to nucleotide and protein biosynthesis. Notably, PRPS enzymatic activity is strictly regulated through allosteric feedback inhibition by downstream nucleotide biosynthetic intermediates such as adenosine diphosphate (ADP) or guanosine diphosphate (GDP) in mammalian cells, ensuring precise cellular control[17,20–22].

In the human genome, three highly conserved PRPS homologous inter-isoforms (PRPS1, 2, and 3) have been identified, with PRPS1 and PRPS2 widely expressed in nearly all somatic tissues, while PRPS3 expression is limited to the testicles[23–25]. PRPS1 is considered the major determinant of PRPP levels, and the dysregulation of PRPS1 function is implicated in conditions such as deafness, optic neuropathy, gout, uric lithiasis, and childhood leukemia[26–29]. Conversely, PRPS2 displays ~94% identity with PRPS1 and shares exactly the same biochemical reaction (Supplementary Fig. 1). Knockdown of PRPS1 or PRPS2 in wild-type MEF cells led to similar decreases in the rates of RNA and DNA production, indicating that PRPS2 and PRPS1 play interchangeable roles in

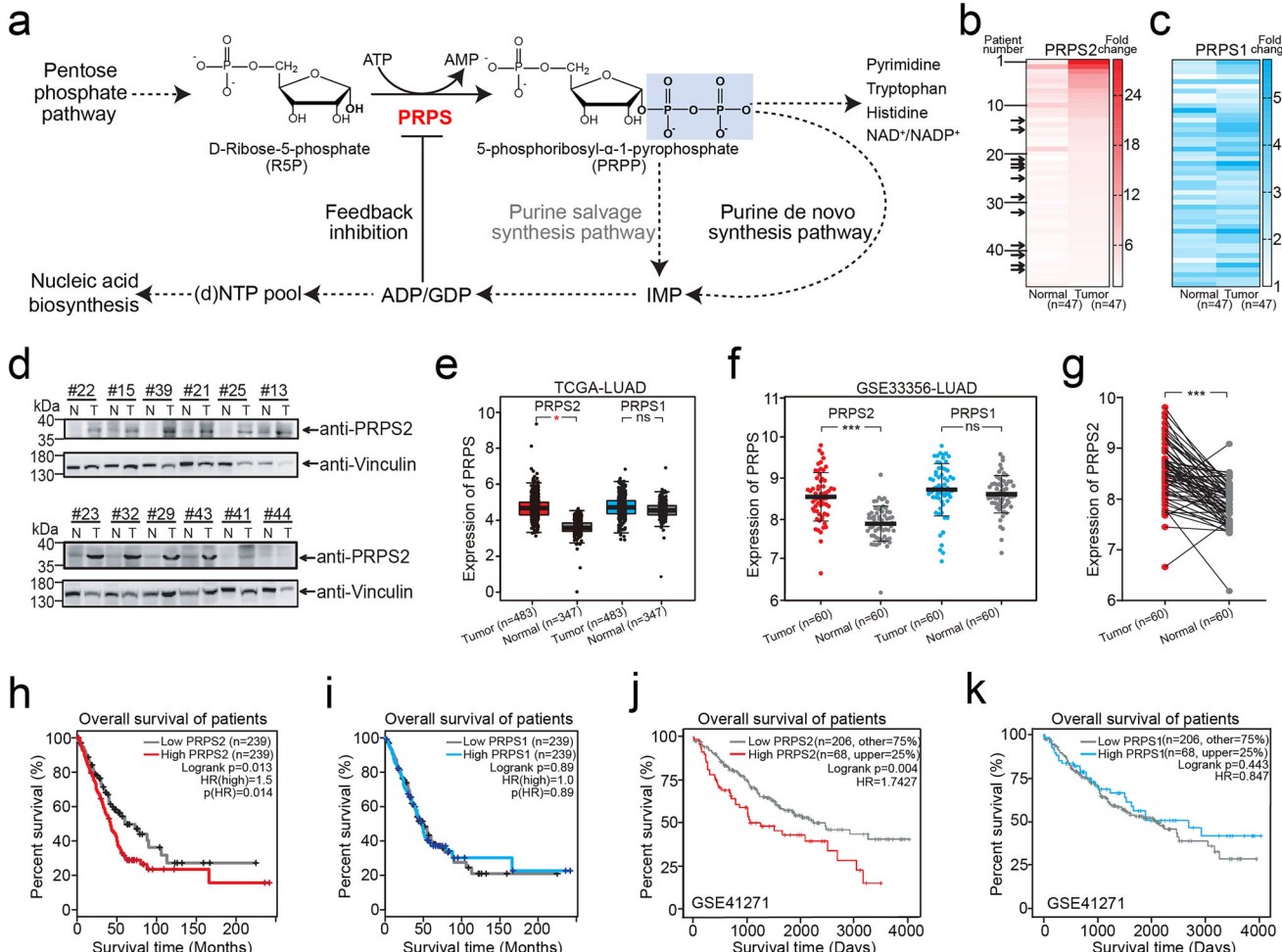

**Fig. 1 | PRPS2 is upregulated in lung malignancies clinically. a** Schematic diagram of PRPS enzymes in the purine biosynthesis pathway. **b, c** The endogenous mRNA expression levels of the *PRPS2* (**b**) or *PRPS1* (**c**) gene in 47 pairs of clinical lung adenocarcinoma specimens alongside adjacent normal tissues were analyzed. β-actin was used as an internal control. The black arrows indicate the samples randomly selected for subsequent detection of endogenous PRPS2 protein expression levels. The data were plotted as average values of biological triplicates. **d** The endogenous PRPS2 protein levels from the randomly selected 12 pairs of lung adenocarcinoma specimens with adjacent normal tissues were analyzed. Vinculin

served as an internal control. **e–g** The mRNA expression levels of the *PRPS2* or *PRPS1* gene in clinical lung adenocarcinoma specimen tissues (LUAD) were compared with normal tissues from TCGA (**e**) or GEO (**f, g**) databases, respectively. The data in (**e, f**) were plotted as the mean ± SDs. *$p < 0.05$, and ***$p < 0.001$ for pairwise comparisons calculated using a two-tailed Student's *t*-test in (**e–g**). **h–k** The effects of *PRPS2* or *PRPS1* mRNA expression on Kaplan-Meier survival curves of lung adenocarcinoma patients in the TCGA database (**h, i**) or GEO database (**j, k**) indicated that high mRNA expression of *PRPS2* is associated with poorer prognosis for lung adenocarcinoma patients.

normal cells[21,24,30]. Strikingly, PRPS2 is specifically upregulated to enhance nucleotide and protein biosynthesis in cancer cells with oncogene c-Myc overexpression, and the knockout of PRPS2 is synthetically lethal to such cancer cells[30–34]. However, the canonical metabolic and non-metabolic functions of PRPS2 in cancer development remain elusive.

In this study, we demonstrate that PRPS2, distinct from its highly inter-isoform conserved homolog PRPS1, promotes RNA m6A methylation in lung adenocarcinoma tumorigenesis through its canonical metabolic and non-metabolic functions. PRPS2 bypasses the typical allosteric feedback inhibition by downstream products ADP/GDP to sustain an excess ATP supply for SAM biosynthesis as a canonical metabolic enzyme. Moreover, it interacts directly with MAT2A to stabilize it, thereby enhancing ATP utilization and SAM biosynthesis, consequently promoting RNA m6A methylation for lung tumorigenesis and metastasis via the WTAP/METTL3/METTL14 methyltransferase complex. Our study elucidates the link between nucleotide metabolism and RNA epigenetics in tumorigenesis through the PRPS2-MAT2A-WTAP/METTL3/METTL14 axis, providing potential insights for drug design and discovery in cancers associated with PRPS2 abnormalities, such as lung adenocarcinoma.

## Results
### PRPS2 promotes lung tumorigenesis
To assess the oncogenic potential of PRPS isoforms, we initially compared the mRNA levels of PRPS1 and PRPS2 in 47 cases of lung adenocarcinoma specimens with paired adjacent normal tissues (Supplementary Tables 1, 2). Our analysis revealed a significant upregulation of PRPS2 mRNA levels in 38 cases (81%) of lung adenocarcinoma compared to their adjacent normal tissues (Fig. 1b), while no significant alteration was observed for PRPS1 mRNA levels (Fig. 1c). Subsequently, we developed a PRPS2-specific antibody capable of distinguishing PRPS2 from PRPS1 (Supplementary Fig. 2) and assessed PRPS2 protein expression levels in 12 pairs of randomly selected lung adenocarcinoma specimens (indicated by black arrows in Fig. 1b). Consistent with the mRNA findings, endogenous PRPS2 protein expression was notably elevated in lung adenocarcinoma specimens compared to paired adjacent normal tissues (Fig. 1d). Further Kaplan-Meier analysis of clinicopathological specimens from the Cancer Genome Atlas (TCGA) and GEO databases confirmed the significant upregulation of PRPS2 expression, in lung adenocarcinoma specimens compared to adjacent normal tissues (Fig. 1e-g). Moreover, overexpression of PRPS2 was strongly associated with poorer patient prognosis across various cancer types, including lung cancer, breast cancer, pancreatic adenocarcinoma, brain lower-grade glioma cancer, mesothelioma, and esophageal carcinoma (Fig. 1h-k, Supplementary Fig. 3-5). These findings suggest that PRPS2, rather than its conserved inter-isoform PRPS1, is specifically upregulated in various malignancies and correlates with unfavorable patient outcomes.

To validate these clinical observations, we examined endogenous PRPS2 expression in 12 types of human lung cancer cell lines and a normal human lung fibroblast cell line, WI38/VA13. As shown in Fig. 2a, PRPS2 expression varied among the cell lines, with 8 out of 12 cancer cell lines exhibiting high or moderate expression levels, while the normal lung fibroblast cell line showed undetectable low expression of PRPS2. Subsequently, we established stable cell lines ectopically expressing Flag-tagged PRPS2 or PRPS1 in cell lines with relatively low endogenous PRPS2 expression (H1299, A549, and WI38/VA13, as indicated by black arrows) (Supplementary Fig. 6a) and evaluated the biological consequences of overexpressing PRPS2 or PRPS1. Consistently, stable ectopic expression of PRPS2 in the H1299 and A549 cell lines exhibited significantly enhanced cell proliferation, colony formation capacity, disorganized growth, increased invasion into the extracellular matrix, and elevated migration compared to control cells that were transfected with an empty vector (EV) (Fig. 2b–h, Supplementary Fig. 6b–g). In contrast, stable ectopic expression of PRPS1

promoted cell proliferation at a moderate level. Remarkably, PRPS2 also stimulated cell proliferation and transformation of the normal human lung fibroblast cell line WI38/VA13, in contrast to PRPS1 (Fig. 2i, Supplementary Fig. 6h). These results confirm that the overexpression of PRPS2 promotes prominent lung cancer cell proliferation, colony formation, anchorage-independent growth, cell migration, and tumorigenesis progression.

### PRPS2 boosts the production of newly synthesized ATP/GTP through the purine biosynthesis pathway
PRPS enzymes play pivotal roles as conventional metabolic enzymes in the purine biosynthesis pathway. They utilize R5P derived from the glucose pentose phosphate pathway as a substrate to generate PRPP, which serves as a precursor for purine biosynthesis. To investigate whether the differential oncogenic effects observed between PRPS2 and PRPS1 result from their enzymatic catalysis affecting the abundance of metabolic flux through the purine biosynthesis pathway, we conducted liquid chromatography tandem mass spectrometry (LC-MS/MS)-based metabolite isotopic tracer profiling along the purine biosynthesis pathway in H1299 lung cancer cells ectopically expressing PRPS2 or PRPS1 (Fig. 3a). Cells were cultured with $^{13}$C stable isotope-labeled glucose ($^{13}C_6$-glucose, M + 6) and monitored for the subsequent labeling of the PRPS substrate R5P (M + 5) and the direct catalytic product PRPP (M + 5). As expected, the levels of newly synthesized R5P via the glucose pentose phosphate pathway significantly decreased within a 1 h timeframe compared to control cells, as R5P is the one of the main substrates of PRPSs for enzymatic catalysis (Fig. 3b). However, PRPS2 overexpression promoted the synthetic generation of PRPP, indicating accelerated consumption and catalytic utilization of R5P upon PRPS2 ectopic expression (Fig. 3c).

Subsequently, we evaluated the utilization of newly synthesized R5P by PRPS2 for subsequent purine synthesis by monitoring the synthetic levels of the major intermediate SAICAR (M + 5) in the de novo purine synthesis pathway, as well as the downstream product inosine monophosphate (IMP) (M + 5) from both de novo and salvage purine synthesis pathways. Again, PRPS2 overexpression prominently enhanced the new synthesis of SAICAR and IMP, while PRPS1 did not (Fig. 3d, e). The synthesized IMP further contributed to purine synthesis, including AMP/GMP, ADP/GDP, and ATP/GTP. As shown in Fig. 3f–k, PRPS2 overexpression promoted the generation of these purine products. These findings suggest that PRPS2 enzymatically stimulates the utilization of R5P, increases the abundance of the intermediates in the pathway, and enhances downstream end-product ATP/GTP production through the purine biosynthesis pathway.

### PRPS2 bypasses the allosteric feedback inhibition of the downstream products
To elucidate the enzymatic differences between PRPS2 and PRPS1 in ATP/GTP production, we characterized the enzymatic properties of recombinant PRPS isoforms. Surprisingly, PRPS2 exhibited -15-fold lower kinetic efficiency ($k_{cat}/K_m$) in R5P or ATP compared to PRPS1 (Fig. 4a, Supplementary Table 3). However, PRPS2 shares 94% amino acid identity with PRPS1, with three non-conserved residues (V103, G104, E105, referred to as 3AA hereinafter) located in the Loop2 region of PRPS2 as one of the major variations in their amino acid sequences, as they are absent in PRPS1. To investigate their distinct catalytic mechanisms, we solved the crystal structure of PRPS2 in complex with R5P, ApCpp (an inactive analog of ATP), and Cd$^{2+}$ ion (an inactive analog of Mg$^{2+}$) by co-crystallization (Supplementary Table 4). The overall structure of PRPS2 closely resembled the previously reported PRPS1 structure (PDB code: 2H06)[19]. However, the Loop2 region in the catalytic pocket of PRPS2 flipped -90° compared to that of PRPS1 due to the insertion of 3AA (Supplementary Fig. 7a). This conformational alteration ultimately reduced the overall size of the PRPS2 catalytic pocket for substrate access (148 Å$^3$ vs 302 Å$^3$ for that of PRPS2 or

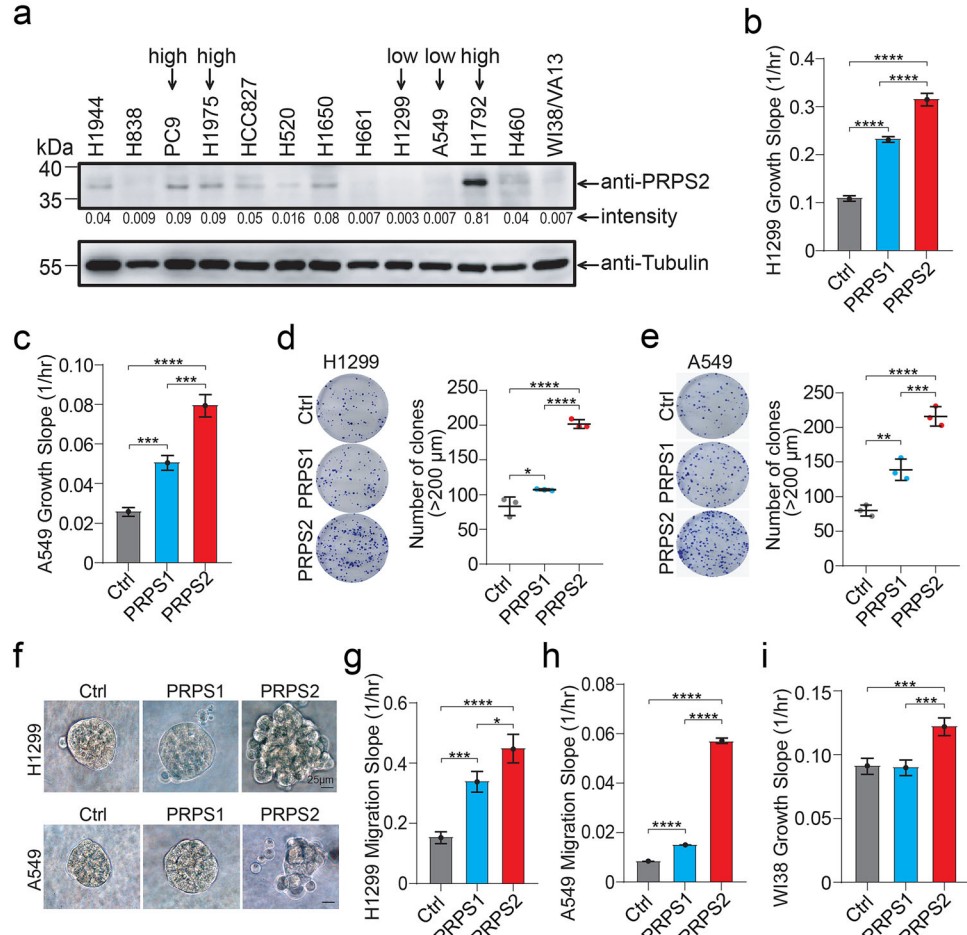

**Fig. 2 | Ectopic expression of PRPS2 promotes lung cancer and fibroblast cell proliferation and migration. a** The PRPS2 expression level was screened in 12 human lung cancer cell lines and the human normal lung fibroblast cell line WI38/VA13. Tubulin was used as an internal control. The black arrows indicated the selected cell lines with high (PC9, H1975, and H1792) or low (H1299 and A549) endogenous PRPS2 expression for subsequent ectopic expression or knockdown assays, respectively. **b, c** Cell proliferation in PRPS2 or PRPS1 ectopically over-expressed H1299 (**b**) and A549 (**c**) cell lines. Control cells refer to the cells that were transfected with an empty vector (EV). **d, e** Colony-formation assays on PRPS2 or PRPS1 ectopically expressed H1299 (**d**) and A549 (**e**) cell lines. **f** 3D culture assay for PRPS2 or PRPS1 ectopically overexpressed H1299 and A549 cell lines. Representative images of cell morphology in the extracellular matrix were taken. **g, h** Cell migration in PRPS2 or PRPS1 ectopically overexpressed H1299 and A549 cell lines. **i** Cell proliferation in PRPS2 or PRPS1 ectopically overexpressed normal lung fibroblast cell line WI38/VA13. The data in (**b–e**) and (**g–i**) were plotted as the mean ± SDs of biological triplicates. $*p < 0.05$, $***p < 0.001$, and $****p < 0.0001$ for multiple comparisons calculated using one-way ANOVA with Tukey's HSD test in (**b–e**) and (**g–i**).

PRPS1, respectively[35]) and introduced steric clashes with the C-terminal α12' helix, leading to its structural disorder in the complex (the prime indicates residues or secondary structures from monomers other than monomer A, Fig. 4b, Supplementary Fig. 8).

To elucidate the function of 3AA, we solved the crystal structure of a PRPS1 chimeric variant, PRPS1(+3AA) in complex with R5P, ApCpp, and $Cd^{2+}$ ion by co-crystallization, where 3AA was inserted into the corresponding position of the PRPS1 Loop2. The structure confirmed that the insertion of 3AA led to the flipping of Loop2 and the outward movement of the α12' helix (Fig. 4c, Supplementary Figs. 7b, 9, and Table 4). Notably, a non-conserved PRPS1 residue, K153' (Q156' in PRPS2), stabilized the α12' helix through a hydrogen bond with the C-terminal residue L321', which was well-observed in the PRPS1(+3AA) structure (Supplementary Fig. 7c). Further kinetic analysis of PRPS chimeric variants (PRPS1 with 3AA insertion and K153Q mutation, PRPS1(+3AA/K153Q); PRPS2 with 3AA deletion and Q156K mutation, PRPS2(-3AA/Q156A), hereinafter) exhibited switched enzymatic activities, where PRPS1(+3AA/K153Q) displayed ~2-fold or 16-fold lower kinetic efficiency in R5P or ATP, respectively, compared to PRPS2(-3AA/Q156A) (Fig. 4d, Supplementary Table 3). Meanwhile, the deletion of the α12' helix from PRPS1 or PRPS2 abolished their enzymatic activity. These results suggest that the four

non-conserved residues of PRPS (3AA and Q156'/K153') are essential for the conformational regulation of the C-terminal α12' helix and the subsequent enzymatic catalysis. The presence of 3AA and Q156' in PRPS2 resulted in lower enzymatic efficiency compared to PRPS1, which contradicts to metabolic observations.

In addition to conventional enzymatic catalysis, PRPS enzymes are downregulated through allosteric feedback inhibition by downstream nucleotide biosynthesis products ADP or GDP[21]. Notably, ADP is also a competitor of ATP, the substrate of PRPSs. The inhibition effect of ADP against PRPS1 would reflect a combination of competing the substrate ATP inside the catalytic pocket as a substrate competitor and binding at the allosteric pocket as an allosteric regulator. To avoid potential confusing, we chose to use GDP instead to ADP specifically for the allosteric inhibition regulation study as GDP does not bind to the catalytic pocket as a substrate competitor. Strikingly, in vitro feedback inhibition assays indicated that the enzymatic activity of PRPS1 was inhibited by GDP with an $IC_{50}$ value of 706 μM, while that of PRPS2 was unaffected by up to 10 mM GDP, suggesting that PRPS2 was insensitive to feedback inhibition (Fig. 4e). Indeed, PRPS2 binds with GDP with extremely weak affinity ($K_d > 200$ μM), whereas PRPS1 bound GDP with a $K_d$ value of 10 μM (Fig. 4e).

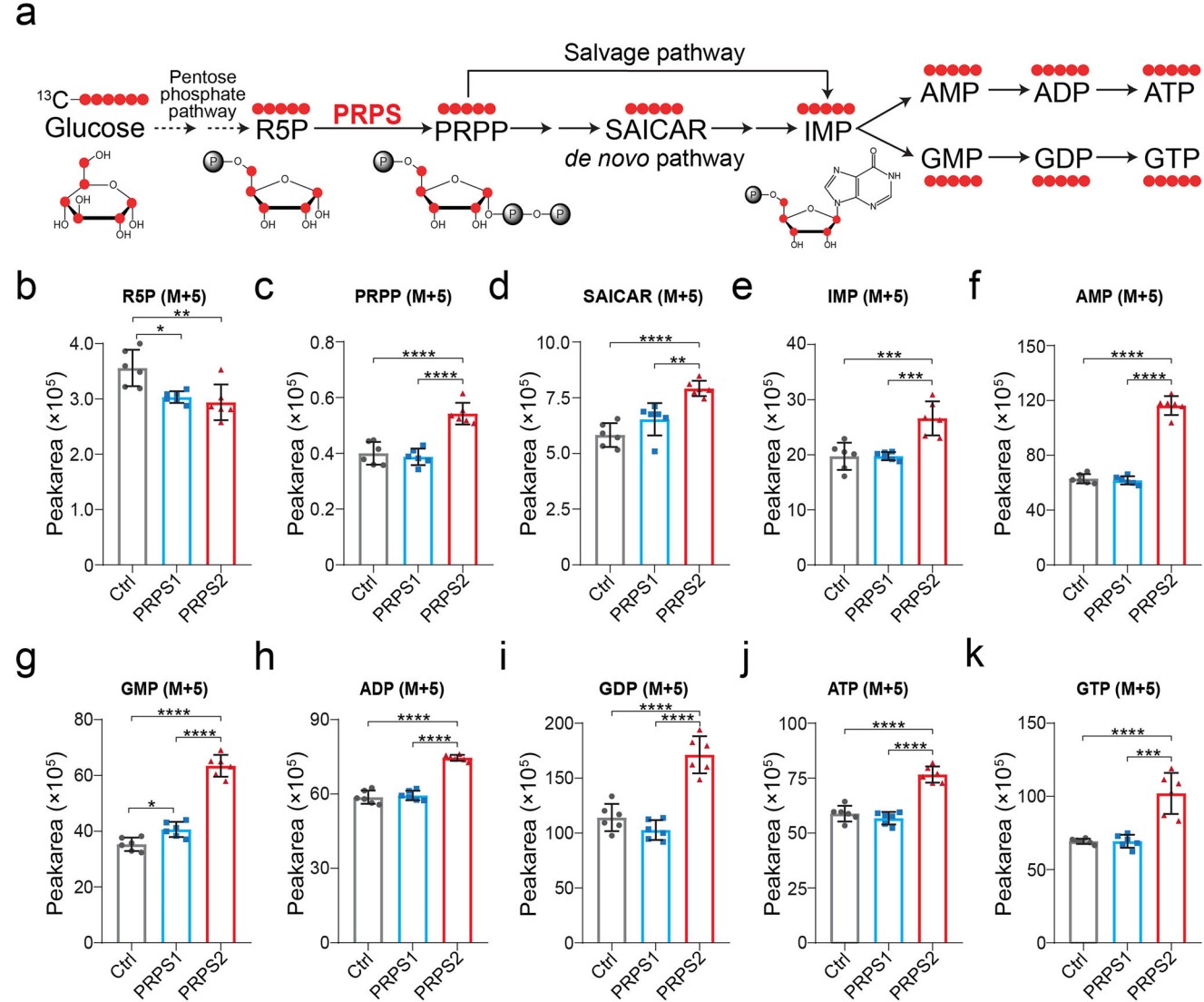

**Fig. 3 | PRPS2 boosts the production of newly synthesized ATP/GTP through the purine biosynthesis pathway. a** Schematic representation of $^{13}$C stable isotope-labeled carbon flow from $^{13}C_6$-glucose through the purine biosynthesis pathway. **b**–**k** Peak areas of $^{13}$C-labeled metabolites in the purine biosynthesis pathway measured by targeted LC-MS/MS from the PRPS2 or PRPS1 ectopic stable expression cell line H1299 in 1 h time-frame. The data in (**b**–**k**) were plotted as the mean ± SDs of six biological replicates. $*p < 0.05$, $**p < 0.01$, $***p < 0.001$, and $****p < 0.0001$ for multiple comparisons calculated using one-way ANOVA with Tukey's HSD test in (**b**–**k**).

To explain these unexpected variations, we determined the structure of PRPS1 in complex with R5P, ApCpp, Cd$^{2+}$ ion, and GDP by co-crystallization (Supplementary Fig. 10 and Table 4). The GDP ligand was positioned at the allosteric site between Loop2 and the C-terminal α12′ helix of PRPS1 with pocket volume 33 Å$^3$, stabilizing the conformations of Loop2 and the α12′ helix for catalysis (Fig. 4g, Supplementary Fig. 7d). In contrast, 3AA in PRPS2 sterically blocked GDP binding at the allosteric site, abolishing the feedback inhibition response of PRPS2. These findings were further confirmed by feedback inhibition assays on chimeric variants of PRPS1(+3AA/K153Q) and PRPS2(-3AA/Q156K), which exhibited switched GDP feedback inhibition responses (Fig. 4h). These biochemical and structural results well explained the observations that stable ectopic expression of PRPS1 moderately promotes cell proliferation, as PRPS1 would continuously produce ADP/GDP for cell proliferation until the concentrations of ADP/GDP reach certain amounts to shut down the enzymatic activity of PRPS1. The observations also indicated that the four non-conserved residues of PRPS2 (3AA and Q156′) enable its enzymatic function by bypassing the conventional downstream products allosteric feedback inhibition.

## PRPS2 stimulates lung tumorigenesis through increased nucleotide biosynthetic production

To validate our biochemical and structural findings, we generated stable H1299 and A549 cell lines expressing chimeric variants PRPS1(+3AA/K153Q) and PRPS2(-3AA/Q156K), as well as enzymatically inactive mutants PRPS1(R96A) and PRPS2(R96A) (Supplementary Fig. 11). As anticipated, the ectopic expression of PRPS2(-3AA/Q156K) in both cell lines exhibited phenotypes similar to PRPS1, significantly attenuating cell proliferation, colony formation, anchorage-independent growth in soft agar, and cell migration compared to PRPS2 (Fig. 5a–d, Supplementary Fig. 12). Conversely, the ectopic expression of PRPS1(+3AA/K153Q) mirrored PRPS2 phenotypes, displaying stronger growth and migratory abilities than PRPS1. The enzymatically inactive mutants PRPS2(R96A) and PRPS1(R96A) showed phenotypes similar to the control. Subsequently, steady-state metabolite isotopic tracer profiling of H1299 cells cultured with $^{13}C_6$-glucose (M + 6) confirmed that ectopic expression of either PRPS2 or PRPS1(+3AA/K153Q) upregulated the synthesis of intermediates and end-products in the purine biosynthesis pathway. In contrast, the

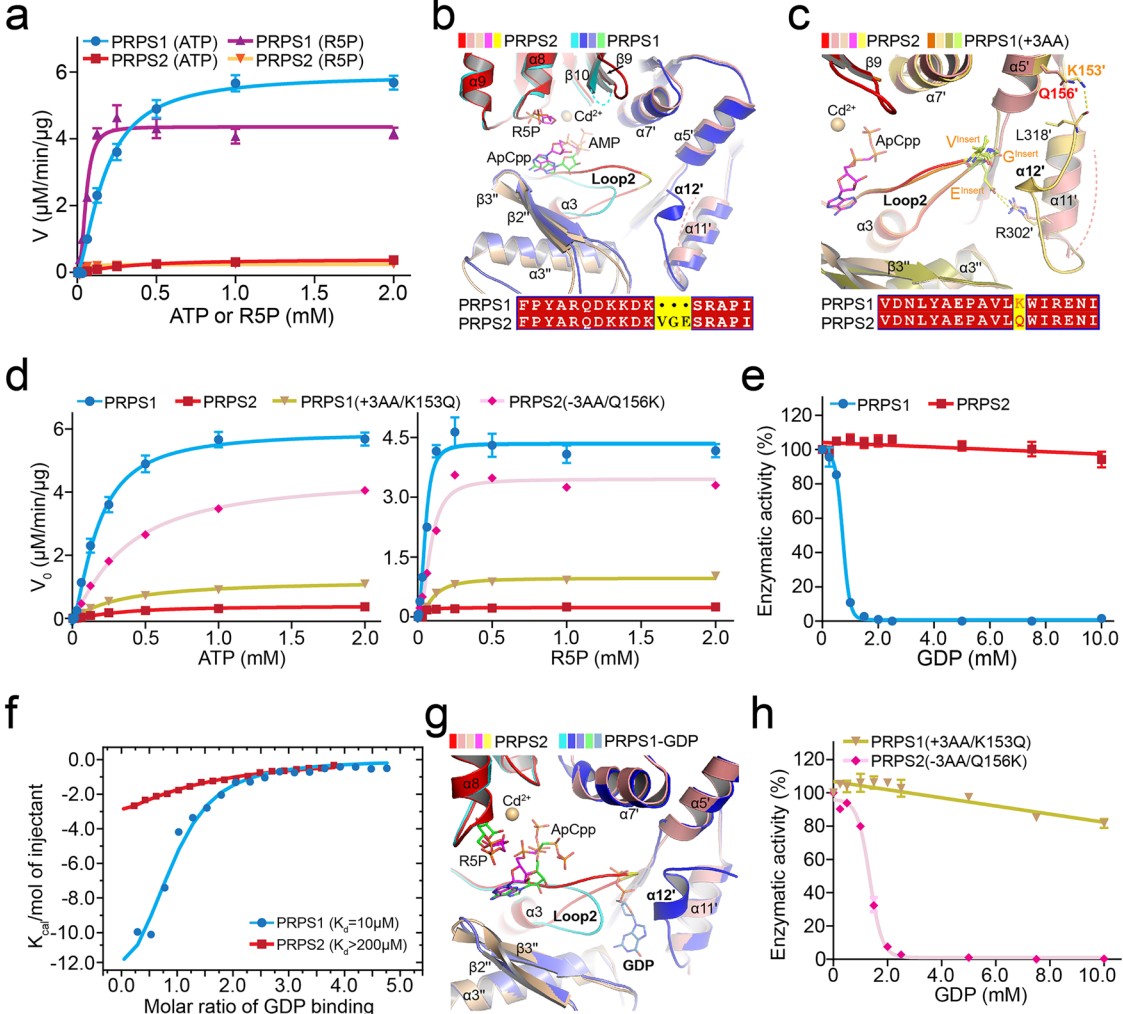

**Fig. 4 | Biochemical and structural characterization indicated vastly distinct properties between PRPS2 and PRPS1. a** Kinetic characterization of PRPS2 and PRPS1. The catalytic efficiencies of PRPS2 and PRPS1 on either ATP or R5P were characterized. **b** Structural superposition of PRPS2 and PRPS1 catalytic pockets among protomers A, B, and F. The structural region of the three additional residues Val, Gly, and Glu (3AA) in PRPS2 was colored in yellow, and the corresponding sequence alignment of the Loop2 region between PRPS2 and PRPS1 was also shown. The dashes indicated the structurally disordered Loop1 region of PRPS1 (cyan) or the C-terminal α12' helix of PRPS2 (salmon), respectively. The primes or double primes indicated the residues from monomer B or C of PRPS hexamers; otherwise, they referred to from monomer A. **c** Structural superposition of PRPS2 and PRPS1( + 3AA) catalytic pockets among protomers A, B, and F. The inserted three residues on PRPS1( + 3AA) were shown as sticks, labeled as $V^{insert}$, $G^{insert}$, and $E^{insert}$, and colored in lime. The corresponding sequence alignment of the α5' helix

between PRPS2 and PRPS1 was also shown. The yellow dashes indicated the H-bonds, and the salmon dashes indicated the structurally disordered C-terminal α12' helix of PRPS2. **d** Kinetic characterization of PRPS1( + 3AA/K153Q) and PRPS2(−3AA/Q156K) in catalyzing either ATP or R5P. **e** Sensitivities of PRPS2 and PRPS1 to GDP feedback inhibition under increasing concentrations of GDP up to 10 mM in vitro. **f** GDP binding of PRPS2 or PRPS1 were evaluated using isothermal titration calorimetry (ITC). Plots of molar enthalpy change against the GDP-PRPS1 or PRPS2 enzyme molar ratio were shown. **g** Structural superposition of PRPS2 and PRPS1 catalytic pockets among protomers A, B, and F indicated that the key 3AA from the PRPS2 Loop2 blocks the allosteric GDP binding site of PRPS1. The GDP ligand that binds to PRPS1 was depicted as sticks and colored in sky blue. **h** Sensitivities of PRPS1( + 3AA/K153Q) and PRPS2(−3AA/Q156K) to GDP feedback inhibition under increasing concentrations of GDP up to 10 mM in vitro. The data in (**a–h**) were plotted as the mean ± SDs of biological triplicates.

expression of PRPS1, PRPS2(-3AA/Q156K), or the inactive mutants PRPS1(R96A) and PRPS2(R96A) did not alter metabolite levels compared to the control (Fig. 5e).

These observations were further validated in mouse xenograft tumor models. As shown in Fig. 5f, g, and Supplementary Fig. 13, ectopic expression of PRPS2 significantly stimulated tumor growth, whereas PRPS1 did not. The in vivo functions of PRPS2 and PRPS1 could be switched by altering their four critical residues, consistent with in vitro structural and biochemical results as well as cell assays. These results demonstrated that the enzymatic function of PRPS2, which bypasses the allosteric feedback inhibition of downstream products, is crucial for its oncogenic role in lung tumorigenesis.

## PRPS2 stabilizes MAT2A in a nonenzymatic manner

While PRPS2 upregulated purine biosynthesis by bypassing feedback inhibition, knockdown of PRPS2 using shRNA in high endogenous PRPS2-expressing cell lines (H1792, PC9, and H1975) significantly suppressed cell proliferation compared to the control (Fig. 6, Supplementary Fig. 14), but not in the low endogenous PRPS2-expressing H1299 cell line (Supplementary Fig. 15). In contrast, knockdown of PRPS1 showed minimal impact, suggesting that the oncogenic function of PRPS2 also involves enzyme-independent mechanisms (Fig. 6a-b, Supplementary Fig. 16a−d).

To explore the enzyme-independent function of PRPS2, we performed Flag-tag based high-affinity pulldown assays in H1299 and HEK293FT cells ectopically expressing Flag-tagged PRPS2 or PRPS1 to identify PRPS2 interacting proteins. As shown in Fig. 6c, seven

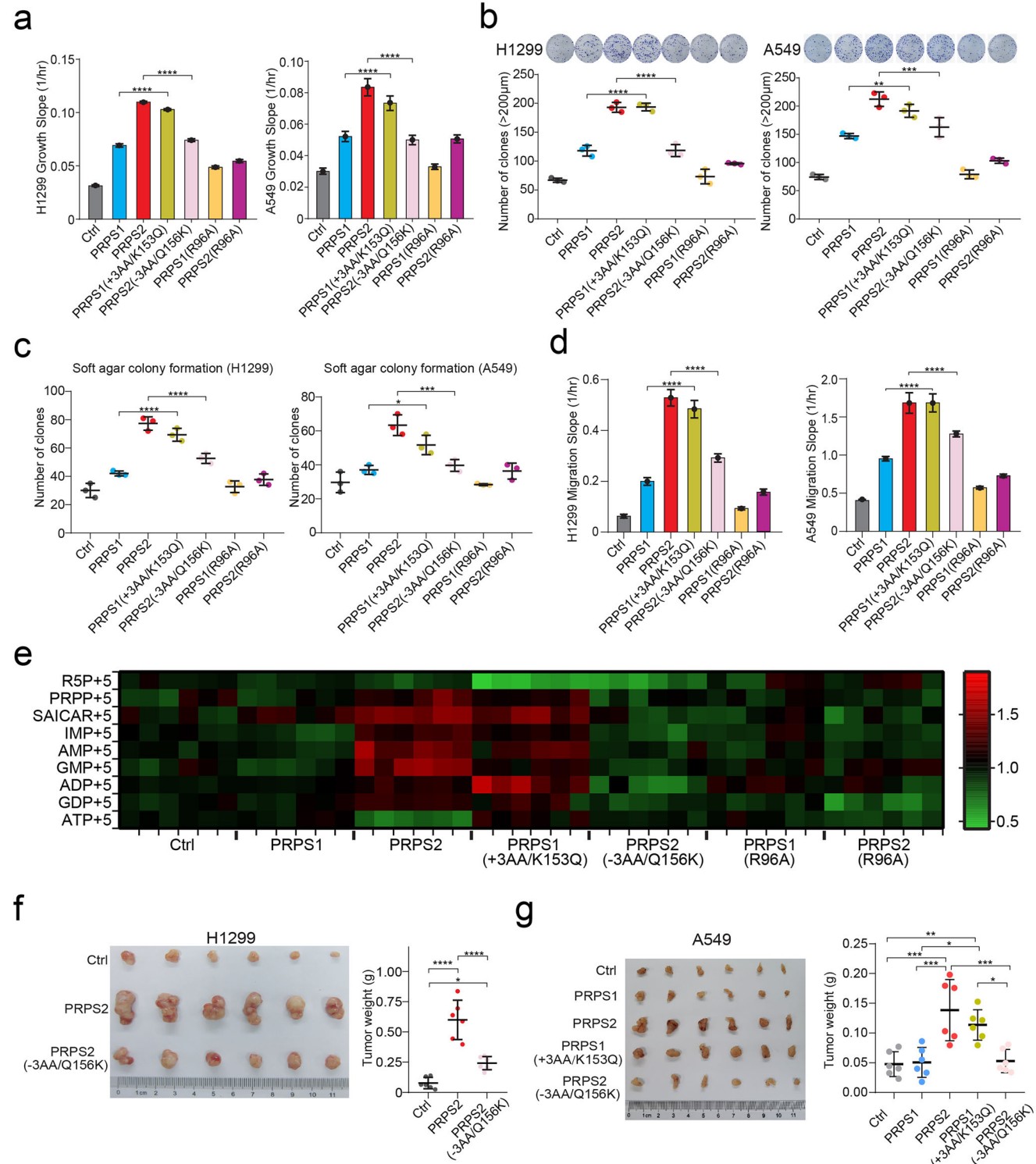

**Fig. 5 | Ectopic expression of PRPS2 or PRPS1(+3AA/K153Q) promotes lung cancer cell proliferation, migration, and tumor growth. a** Cell proliferation of PRPS1, PRPS2, PRPS1(+3AA/K153Q), PRPS2(−3AA/Q156K), PRPS1(R96A), or PRPS2(R96A) ectopically overexpressed H1299 and A549 cell lines. **b** The colony-formation assay showed that PRPS1(+3AA/K153Q) ectopic expression promotes H1299 and A549 cell proliferation. **c** Soft agar assay. **d** Cell migration of PRPS1(+3AA/K153Q),PRPS2(−3AA/Q156K), PRPS1(R96A), or PRPS2(R96A) ectopically expressed H1299 and A549 cell lines. **e** Steady-state metabolite profiles of the ectopically expressed H1299 cell lines (*n* = 6). **f, g** Xenograft tumor growth assay. The data in (**a–d**) and (**f, g**) were plotted as the mean ± SDs of biological triplicates. *$p < 0.05$, **$p < 0.01$, ***$p < 0.001$, and ****$p < 0.0001$ for multiple comparisons calculated using one-way ANOVA with Tukey's HSD test in (**a–d**) and (**f, g**).

potential interacting proteins were identified, specifically targeting PRPS2 (Supplementary Table 5). Among these, SAM synthetase MAT2A, which catalyzes the synthesis of SAM using methionine and ATP (a key end-product of the purine biosynthesis pathway[13]), was selected as a candidate for interaction with PRPS2.

To confirm the interactions between PRPS2 and MAT2A, we performed co-immunoprecipitation (Co-IP) and immunostaining assays. Flag-tagged PRPS2 and GFP-HA-tagged MAT2A were overexpressed in HEK293FT cells, and Co-IP results showed that ectopically expressed PRPS2 co-immunoprecipitated with MAT2A, and vice versa (Fig. 6d).

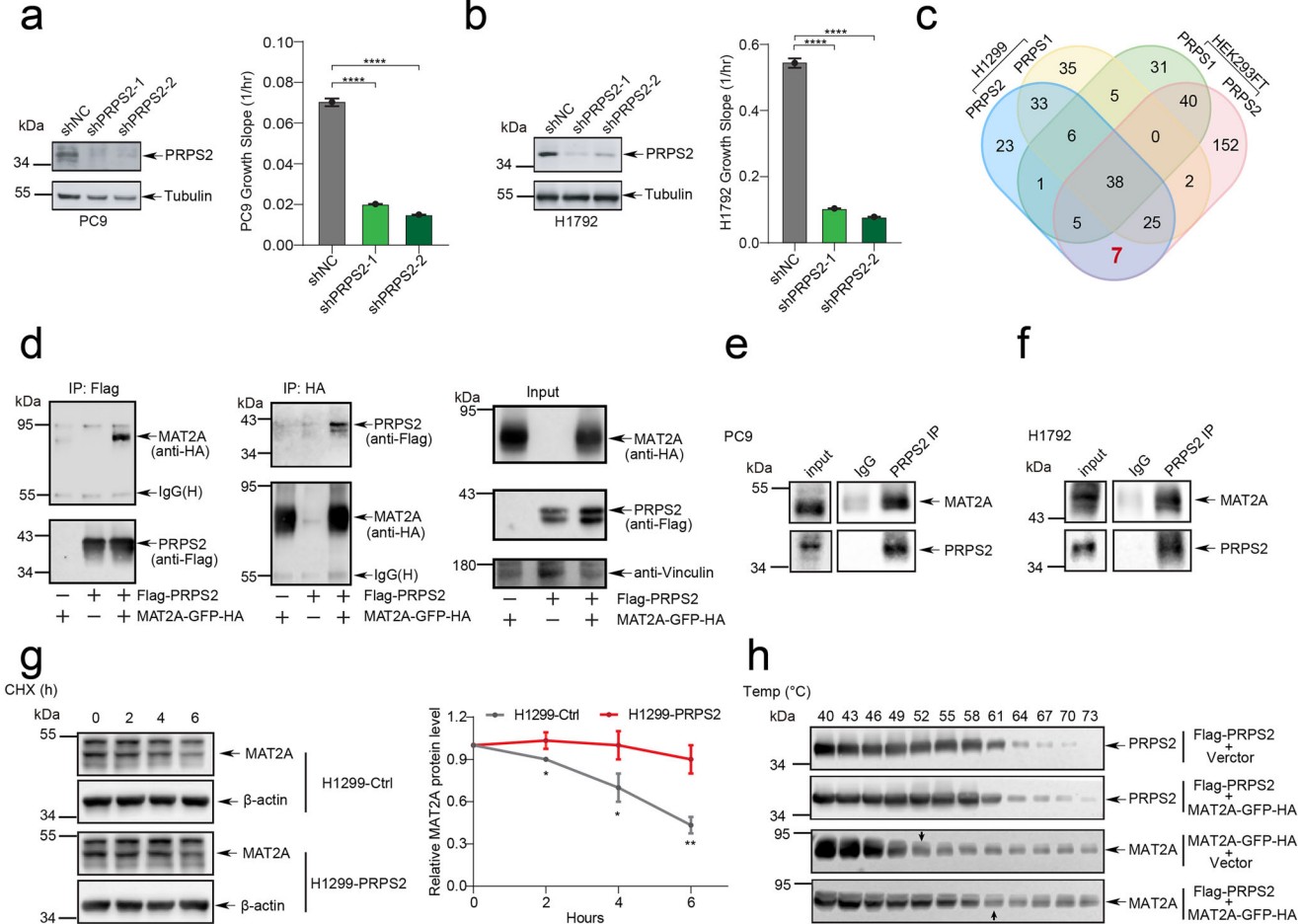

**Fig. 6 | PRPS2 stabilizes MAT2A in a non-enzymatic manner. a, b** Western blot analysis and cell proliferation of PRPS2 shRNA knockdown PC9 (**a**) and H1792 (**b**) cell lines. **c** Flag-tag based high-affinity pulldown from Flag-tagged PRPS2 or PRPS1 ectopically expressed H1299 and HEK293T stable cell lines for PRPS2 endogenous interaction protein trapping. **d** Immunoprecipitation and Western blot analysis showed the interaction between PRPS2 and MAT2A in HEK293T cells transfected with FLAG-PRPS2 and HA-MAT2A. **e-f** The interaction of endogenous PRPS2 and endogenous MAT2A in PC9 (**e**) and H1792 (**f**) cell lines. **g** MAT2A expression levels in PRPS2 ectopically expressed H1299 cells treated with cycloheximide (CHX).

**h** PRPS2 stabilized MAT2A in cellular thermal shift assay in HEK293T cells transfected with FLAG-PRPS2 and HA-MAT2A. The black arrows indicate the temperature points at which protein degradation was observed. The data in (**a, b**) were plotted as the mean ± SDs of biological triplicates. ****$p < 0.0001$ for multiple comparisons calculated using one-way ANOVA with Tukey's HSD test in (**a, b**). The data in (**g**) were plotted as the mean ± SEM of three independent replicates. Western blot data are representative of at least 3 biological repeats (**d–f, h**). *$p < 0.05$, and **$p < 0.01$ for pairwise comparisons calculated using a two-tailed Student's *t*-test in (**g**).

Additionally, interactions between endogenous MAT2A and PRPS2 were observed in PC9 and H1792 cells (Fig. 6e-f). Subsequent immunostaining assays demonstrated that transfected PRPS2 and endogenous MAT2A localize to the cytoplasm, corroborating the Co-IP results and indicating interactions between PRPS2 and MAT2A (Supplementary Fig. 16e).

To investigate the impact of PRPS2 on MAT2A stability, we performed cycloheximide (CHX) chase assays and cellular thermal shift assays (CETSA). In the presence of the protein synthesis inhibitor CHX, the stability of MAT2A was significantly increased in PRPS2-overexpressing H1299 cells (Fig. 6g). Furthermore, temperature-induced degradation of MAT2A was suppressed upon PRPS2 binding in HEK293FT cells, but not by PRPS1 (Fig. 6h, Supplementary Fig. 17). These results suggest that the interaction between PRPS2 and MAT2A markedly enhances the stability of MAT2A, extending its lifetime in cells.

**PRPS2 enhances the methionine metabolic cycle and upregulates SAM synthesis**

MAT2A is a rate-limiting enzyme in the methionine metabolic cycle, responsible for synthesizing S-adenosylmethionine (SAM), a major methyl donor for various biomolecular modifications, using methionine and ATP (Fig. 7a). Stabilizing MAT2A directly influences the utilization of methionine and ATP for the synthesis of SAM. When PRPS2 is overexpressed in H1299 cells, ATP accumulation is initially observed within a 1 h timeframe (Fig. 3j). However, over a 6 h timeframe, ATP levels sharply decrease (Figs. 5e, 7b). This suggests that ectopically expressed PRPS2 stabilizes MAT2A, accelerating the utilization and consumption of ATP by MAT2A in the methionine metabolic cycle.

We further analyzed the methionine metabolic flux in H1299 cells using $^{13}C_6$-glucose and $^{13}C_1$-methionine as substrates over a 6 h period. The tracing results revealed rapid elimination and utilization of methionine for SAM biosynthesis, along with increased production of SAM (M + 5, M + 1) and its demethylated analog S-adenosylhomocysteine (SAH, M + 5, M + 1) upon PRPS2 overexpression, while methionine (M + 1) levels remained unchanged due to excessive uptake (Fig. 7c–h). Similar phenotypes were observed in cells overexpressing PRPS1( + 3AA/K153Q) (Supplementary Fig. 18). These results indicate that PRPS2 enhances the methionine metabolic cycle and promotes SAM generation by increasing the stability of MAT2A in an enzyme-independent manner.

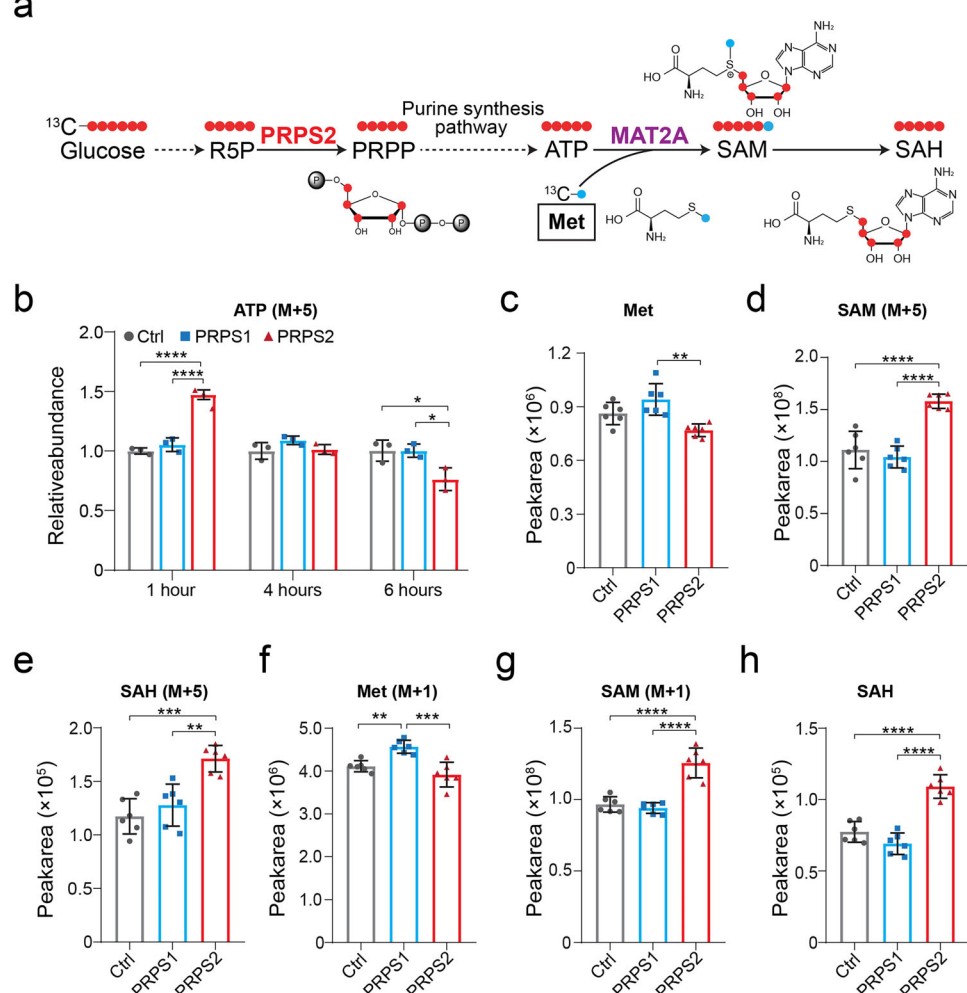

**Fig. 7 | PRPS2 enhances the methionine metabolic cycle and upregulates SAM synthesis. a** Schematic of $^{13}C$ stable isotope-labeled carbon flow from $^{13}C_6$-glucose or $^{13}C_1$- methionine through the methionine metabolic cycle. **b** $^{13}C$-labeled ATP elimination in 1-, 4-, and 6 h timeframes measured by targeted LC-MS/MS from PRPS2 or PRPS1 ectopic stable expression cell line H1299. **c**–**h** Peak areas of $^{13}C$-labeled metabolites through the methionine metabolic cycle. The data were monitored after a 6 h timeframe of $^{13}C_6$-glucose (red dots) or $^{13}C_1$-methionine (cyan dots) treatments. The data in (**b**) were plotted as the mean ± SDs of biological triplicates. The data in (**c**–**h**) were plotted as the mean ± SDs of six biological replicates. *$p < 0.05$, **$p < 0.01$, ***$p < 0.001$, and ****$p < 0.0001$ for multiple comparisons calculated using one-way ANOVA with Tukey's HSD test in (**b**–**h**).

## PRPS2 promotes RNA m6A methylation in lung adenocarcinoma

S-adenosylmethionine serves as a critical methyl donor for various biomolecular modifications, including DNA, RNA, and histone methylation. Aberrant SAM synthesis is closely associated with cancer development by elevating methylation levels within cells[36,37]. We examined the levels of key epigenetic methylation modifications—such as DNA 5-methyldeoxycytosine (5mC), RNA m6A, N1-methyladenosine (m1A), C5-methylcytosine (m5C), and histone H3 trimethylations (H3K4me3, H3K9me3, and H3K27me3)—following ectopic overexpression of PRPS2 or PRPS1 in H1299 cells. Strikingly, newly synthesized SAM was specifically utilized for RNA m6A methylation (M + 5 and M + 1) in PRPS2-overexpressing H1299 and A549 cells, as tracked by ribose backbone labeling from $^{13}C_6$-glucose or direct methyl group labeling from $^{13}C_1$-methionine (Fig. 8a–d, Supplementary Fig. 19a, d). Other methylation modifications remained unchanged (Supplementary Figs. 19–21). Additionally, stable isotope labeling assays using $^{13}C_6$-glucose for DNA and RNA synthesis indicated that neither DNA nor RNA synthesis was stimulated by PRPS2 overexpression. This suggests that newly synthesized ATP by PRPS2 through the purine synthesis pathway is primarily utilized for SAM synthesis and RNA m6A methylation rather than for DNA or RNA synthesis (Supplementary Fig. 22). Similarly, the RNA m6A level upon PRPS1( + 3AA/K153Q) overexpression is consistent with that of PRPS2, while that of PRPS1(R96A) and PRPS2(R96A) remains unchanged (Supplementary Fig. 23).

The WTAP/METTL3/METTL14 complex and METTL16 are known to be the predominant RNA m6A methyltransferases in humans[10–12]. To determine which methyltransferase is responsible for RNA m6A methylation in PRPS2-overexpressing cells, we knocked down METTL3, METTL14, or METTL16 by using siRNAs (Supplementary Figs. 24, 25a), or inhibited the enzymatic activity of METTL3/METTL14 complex with their specific inhibitor STM2457 in PRPS2 or PRPS1 overexpressing H1299 cells[38]. The results suggested that the knocking down or inhibition of METTL3/METTL14 significantly reduced RNA m6A levels in PRPS2-overexpressing cells to those observed in control and PRPS1-overexpressing cells (Fig. 8e, Supplementary Fig. 26). In contrast, knocking down METTL16 did not affect RNA m6A methylation levels (Supplementary Fig. 25b, c). These results suggest that PRPS2 promotes RNA m6A methylation predominantly via the WTAP/METTL3/METTL14 complex rather than METTL16.

## Discussion

Cancer metabolism is a pivotal aspect of tumorigenesis and metastatic progression. Cancer cells exploit pathways such as the

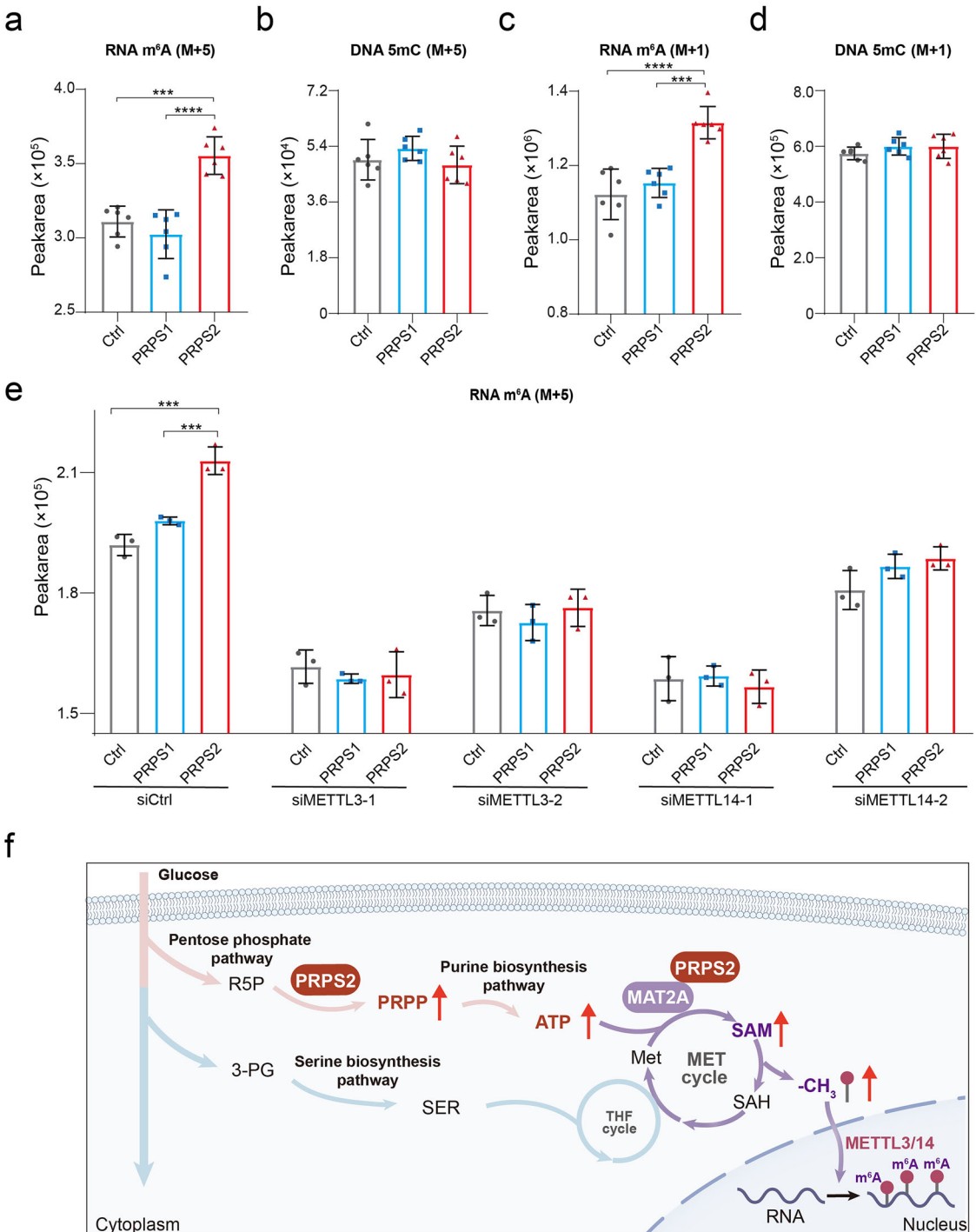

**Fig. 8 | PRPS2 promotes RNA m⁶A methylation. a–d** Peak areas of ¹³C-labeled methylated nucleotides from DNA 5mC or RNA m⁶A in PRPS2 or PRPS1 ectopically expressed H1299 cell lines. **e** Peak areas of ¹³C-labeled (¹³C₆-glucose) methylated nucleotides from RNA m⁶A upon siRNA knockdown of METTL3 or METTL14 in PRPS2 or PRPS1 ectopically expressed H1299 cell lines. **f** The schematic diagram depicts the contribution of PRPS2 in promoting RNA m⁶A methylation. The data in (**a–d**) were plotted as the mean ± SDs of six biological replicates. The data in (**e**) were plotted as the mean ± SDs of biological triplicates. ***$p < 0.001$, and ****$p < 0.0001$ for multiple comparisons calculated using one-way ANOVA with Tukey's HSD test in (**a, c, e**).

mTORC1 signaling to sense environmental cues and genetic signals, resulting in the upregulation of key oncogenes like Myc. This orchestrates cellular metabolic pathways, including nucleotide, amino acid, and lipid biosynthesis, as well as epigenetic regulation[30,39–41]. Previous studies have highlighted the role of c-Myc in upregulating PRPS2, but not its highly conserved inter-isoform homolog PRPS1, to meet the specific high demands for nucleotide synthesis in cancer cells[30,33]. Additionally, c-Myc promotes the expression of MAT2A and the

methyltransferase WTAP, enhancing SAM synthesis and RNA m⁶A hypermethylation for oncogenic behaviors[37,42]. However, the interplay between canonical metabolic functions and the non-metabolic roles of enzymes in sustaining SAM production for RNA m⁶A hypermethylation in cancer cells remains poorly understood.

Our study delves into the dual mechanisms of PRPS2, a key metabolic enzyme in the purine biosynthesis pathway, in regulating SAM synthesis and RNA m⁶A hypermethylation, crucial for the

development of lung tumorigenesis (Fig. 8). We found that PRPS2 utilizes specific residues (3AA and Q156) to evade allosteric feedback inhibition by downstream products, allowing sustained production of newly synthesized ATP. Moreover, PRPS2 interacts with MAT2A to stabilize it and enhance its lifespan, stimulating the utilization of excess amounts of ATP and methionine from PRPS2-overexpressed purine biosynthesis pathway and serine synthesis pathway, respectively, for excessive amount of SAM production. Subsequently, c-Myc upregulates the expression of WTAP alongside PRPS2 expression to enhance the expression levels of the WTAP/METTL3/METTL14 complex[36], in which METTL3 exhibits ~10-fold higher $K_m$ value (~0.1 μM) than other major SAM-dependent methyltransferases for DNA, RNA, and histone methylation (~1 μM)[43–46], enabling a higher priority of SAM utilization by the complex. Meanwhile, PRPS2, the WTAP/METTL3/METTL14 complex, RNA translation initiation factors (eIFs), and ribosomes would form a mega-sized protein complex machine that binds to the 5′ UTR region of mRNA to facilitate the specific utilization of SAM by the METTL3/14-WTAP complex for RNA m6A hypermethylation and subsequent promotion of translation initiation (Supplementary Fig. 27)[30,47–49].

In this work, additional knockdown/knockout of PRPS2 followed by rescue experiments may further strengthen our conclusion by excluding the potential off-target effects as well as measuring the changes in the amount of newly-made ATP in the absence of PRPS2 feedback inhibition. However, the orthogonal approaches for PRPS2 knockdown/knockout, including lentiviral CRISPR/gRNA, shRNA as well as siRNAs, are lethal to the cancer cell lines with high endogenous PRPS2 expression level, which is in consistent with previous findings[30]. To exclude the potential off-target effects, we performed shRNA knockdown assay against PRPS2 in H1299 cell line with low endogenous PRPS2 expression level instead, and the results suggested that knockdown of PRPS2 H1299 cell line does not interfere the cell proliferation, confirming that PRPS2 is indeed on-target and dependency in cell lines with high endogenous PRPS2 expression level, such as NSCLC (Supplementary Fig. 15). Another limitation of this study was the employment of the 13C-labeling isotope tracer experiment for ATP tracing, which would only observe the amount changes of ATP generated though the purine synthesis pathway (including de novo synthesis and salvage synthesis pathways) rather than those generated from the multiple energy supplemental metabolic pathways.

Nevertheless, our results highlight the unique and indispensable role of PRPS2 serving as an essential "molecular rheostat" that links glucose metabolism to diverse metabolite biosynthesis pathways vital for cancer development. This positions PRPS2 as an intriguing anti-cancer drug target due to its distinct oncogenic functionality. Therapeutic interruption of PRPS2 function through developing inhibitors that selectively bind to its distinctive Loop2 region for enzymatic inhibition, or protein-protein interaction (PPI)-based inhibitors to disrupt the PRPS2-MAT2A association could disrupt nucleotide biosynthesis and RNA epigenetic methylation, which are essential for tumor progression. Given that PRPS2 is only essential for cancer cells and is less expressed in normal tissues, such therapeutics may be more specific and exhibit fewer side effects. Consequently, our study underscores the potential of targeting PRPS2 enzymatic activity to halt tumor growth and defines a valuable and vulnerable target in cancer metabolism, essential for proliferation across various cancer types.

## Methods

### Ethics statement
The experiments using human samples were approved by the Ethics Committee of Zhongshan Hospital of Fudan University under ethical ID B2018-255, and was performed in accordance with the principles of the Declaration of Helsinki and relevant guidelines and regulations, and compensation was not provided.

### Patients and tissue specimens
From May 2019 to October 2019, 47 patients were recruited into this study, and the written informed consent was obtained from all patients. None of the patients had a history of radiotherapy or chemotherapy prior to the surgery. The data of the patients, including gender, age at diagnosis, tumor size at diagnosis, tumor (topography), lymph node, metastasis, and TNM stages, were collected and shown in Supplementary Table 1. The histologic types were independently assigned by at least 3 pathologists in a double-blinded fashion. Precast gels purchased from Thermo Fisher (NuPAGE™) were used for the detection of PRPS2 protein expression levels in the patient samples.

### Plasmid constructions
To construct the PRPS1 or PRPS2 expression vector, the coding sequences of PRPS1 or PRPS2 were cloned into the lentiviral expression vector pLV-CMVenh-EGFP-CD513B (System Biosciences) using *EcoRI* and *BamHI* restriction enzymes. The PRPS1(R96A), PRPS2(R96A), PRPS1(+3AA/K153Q), and PRPS2(-3AA/Q156K) mutants were generated by employing the KOD-plus-mutagenesis Kit (TOYOBO). To construct the shRNA vector for PRPS2, the synthetic double-stranded oligonucleotides were cloned into the pLKO-puro vector (Sigma-Aldrich) utilizing *AgeI* and *EcoRI* restriction enzymes. The sequences of the shRNA oligonucleotides are listed in Supplementary Table 2.

### Protein subcloning, expression and purification
The full lengths of the PRPS1, PRPS2, PRPS1(+3AA), PRPS2(-3AA), PRPS1-ΔC, and PRPS2-ΔC genes were synthesized and subcloned into the pSumo plasmid. The QuikChange Site-Directed Mutagenesis (Stratagene) strategy was used to create the PRPS1(R96A), PRPS2(R96A), PRPS1(+3AA/K153Q), and PRPS2(-3AA/Q156K) mutants. The proteins were transformed and expressed in BL21(DE3) cells, and were purified by using a Ni-NTA affinity column. The His-Sumo tag attached to the N-terminus of the protein was removed by recombinant ULP1 protease cleavage. The proteins were further purified using an ion-exchange column and gel-filtration Superdex-200 column. The final purified protein was quantified by using Bradford reagent purchased from Bio-Rad.

### Kinetics assay
Steady-state kinetics of wild-type and mutant PRPS were measured at 37 °C by monitoring the decrease in absorbance at 340 nm, as described previously[50]. The reaction mixtures contained 1 - 25 μg/ml enzyme in 50 mM Tris, pH 7.6, 16 mM sodium phosphate, 1 mM DTT, 10 mM MgCl₂, 0.5 mM PEP, 0.4 mM NADH, 2 U/ml MK, 2 U/ml PK, 2 U/ml LDH and various substrate concentrations. Each individual rate measurement was performed in triplicate. The data were analyzed by using GraphPad Prism with non-linear fitting to the allosteric sigmoidal equation (Supplementary Table 3).

### GDP allosteric feedback inhibition assay
The wild-type and mutant PRPS were assayed with various concentrations of GDP (0 - 10 mM) at 37 °C in the reaction buffer containing 50 mM Tris, pH 7.6, 16 mM sodium phosphate, 1 mM DTT, 10 mM MgCl₂, saturating concentrations of the substrates ATP (1 mM) and R5P (1 mM). Reactions were quenched with 12.5% trichloroacetic acid at ambient temperature for 15 min. After centrifugation (13,000 rpm at 25 °C for 20 min), 20 μl products was subjected to ion-pair HPLC (Agilent ZORBAX Eclipse XDB-C8, 5 μm, 4.6 × 150 mm column) to detect the production of AMP. Mobile phases were a solution of 10 mM NaH₂PO₄ and 10 mM TBAH in water (phase A) and 10 mM TBAH in methanol (phase B). The flow rate was 1.5 ml/min with online UV detection at 254 nm.

### Isothermal titration calorimetry (ITC) assay
An isothermal titration calorimetry assay was performed as previously described[51]. Briefly, the protein samples were loaded into the cell of a

MicroCal ITC200 titration calorimeter (GE Healthcare), and GDP was diluted before being loading into the syringe. An initial 0.2 µl injection was followed by 19 injections of 2 µl GDP solution at 3 min intervals under 25 °C. The ITC measurement data were assessed using the commercial Origin 7.0 program to analyze the stoichiometry (n), binding constants ($K_d$), and change in enthalpy for each enzyme-ligand interaction ($\Delta H$).

## Size-exclusion chromatography multi-angle light scattering (SEC-MALS)

The absolute molar masses of wild-type and mutant PRPS were assessed by size-exclusion chromatography multi-angle light scattering. Protein samples were loaded onto a Wyatt Technology WTC-030S5 column in 20 mM $NaH_2PO_4$, pH 7.0, 300 mM NaCl, 1 mM DTT, at 0.5 ml/min using HPLC system (Agilent Technologies). Light scattering (LS) and differential refractive index (dRI) profiles, with fitted molecular weights (Mw) plotted across elution peaks, were determined by using ASTRA 6 software (Wyatt Technology).

## Protein-ligands complex crystallization

500 µM purified wild-type and mutant PRPS were incubated with ApCpp (inactive analog of ATP) or GDP in a molar ratio of 1:5 at room temperature for 1 h. All the crystals were obtained by mixing 1 µl of the protein complex with an equal volume of reservoir solution and equilibrating against 500 µl of the reservoir solution at 277 K. Crystals of PRPS2 in complex with ligands were obtained in the reservoir solution containing 0.2 M Lithium sulfate and 35% MPD. Crystals of PRPS1 in complex with GDP were obtained in reservoir solution containing 0.2 M sodium acetate and 10% PEG3350. PRPS1( + 3AA) crystals were obtained in the reservoir solution containing 0.1 M Lithium sulfate and 20% PEG3350. Crystals were soaked for 1 h in the reservoir solution containing 1 mM $CdCl_2$. The crystals were then flash-frozen in liquid nitrogen with 27.5% glycerol (v/v) as the cryoprotectant solution.

## Data collection and structure determination

The crystal diffraction data of PRPS2, PRPS1-GDP, and PRPS1( + 3AA) were collected at the beamlines BL19U1 and BL18U of the Shanghai Synchrotron Radiation Facility (SSRF) (Supplementary Table 4), and processed with HKL3000[52]. The scaled data were used for molecular replacement. The phases were determined by using Phaser in the CCP4 suite with the crystal structure of PRPS1 as the search model (PDB code: 2H06)[19]. The structures were then refined by using Phenix-refine[53]. Electron density interpretation and model building were performed using the computer graphics program Coot[54]. The final structures were visualized by PyMOL software.

## Cell cultures

The human lung cancer cell lines NCI-H1299, NCI-H1792, NCI-H1975, and PC-9 were cultured in RPMI 1640 (Hyclone, Logan, UT, USA) containing 10% fetal bovine serum (Biowest, Kansas, MO, USA), 1% penicillin and streptomycin (Invitrogen, CA, USA). A549 cells were cultured in F-12K Medium (Kaighn's Modification of Ham's F-12 Medium) supplemented with 10% FBS and 1% penicillin and streptomycin at 37 °C with 5% $CO_2$. The normal human lung fibroblast cells WI38/VA13 were cultured in MEM medium supplemented with 10% FBS, 1% penicillin and streptomycin, 1% non-essential amino acids, 1 mM sodium pyruvate at 37 °C with 5% $CO_2$. The HEK293T cells were cultured in Dulbecco's modified Eagle's medium (DMEM, Hyclone) containing 10% FBS and 1% penicillin and streptomycin at 37 °C with 5% $CO_2$.

## RNA extraction, reverse transcription, and quantitative Real-Time PCR

Total RNA from patient tissues was extracted and isolated using the mRNA purification Kit (Takara, Tokyo, Japan). cDNA was generated with the PrimeScript RT Reagent Kit (Takara, Tokyo, Japan). Relative cDNA abundance was measured with the LightCycler 480 SYBR Green I Master (Roche, Welwyn Garden, UK), and the primers used for qPCR reactions are shown in Supplementary Table 2. The data were normalized by β-actin.

## Soft-agar colony formation assay

The ability of anchorage-independent growth was evaluated using soft agar colony formation assay. The agar medium mixture containing 0.6% low-melting point agarose (Amresco) and 10% FBS was added to six-well plates as the base agar gel. Each stable cell line was seeded at a density of $2 \times 10^3$ cells per well in 2 ml of culture medium containing 10% FBS and 0.35% low-melting point agarose on the top of the base gel as the colony formation gel. After 21–28 days, cell colonies were fixed and stained with 0.005% Crystal Violet. Photographs of the colonies developed in soft agar were taken, and the number of colonies was scored by using Photoshop CC 2018 (Adobe, USA).

## Colony-formation assay

Each stable cell line was trypsinized into a single-cell suspension and plated in 6-well plates at a density of 300 cells per well. After 14 days, cell colonies were fixed with methanol and stained with 0.5% crystal violet. Photographs of the colonies were taken, and the number of colonies >200 µm was scored using ImageJ V1.8.0 (NIH, USA). Each experiment was performed in triplicate.

## Three-dimensional (3D) culture assay

The 3D cell culture assay was performed as described previously[55]. Briefly, 5 µl of 3D culture matrix™ (Millipore) and 5 µl of cells ($1 \times 10^5$ cells/ml) were mixed and added into the inner well of µ-slides (ibidi GmbH, Martinsried, Germany). After polymerization, cell-free medium was added to fill the upper well. The µ-slides were incubated at 37 °C in a 5% $CO_2$ cell incubator for at least 7 days, and microscopy images were taken.

## Cell proliferation assay

Cell growth and migration assays were performed using Real-Time Cell Analysis with the xCELLigence RTCA-DP system (Roche) and the IncuCyte Live Cell Analysis system (Sartorius). The xCELLigence system enables real-time cell analysis (RTCA) through the RTCA-DP instrument equipped with CIM-Plate 16 or E-Plate 16. The RTCA-migration assay was conducted as described previously (Cell Death Dis. 2013, 4, e928-e942). Briefly, cells were serum-starved for 6 h before the migration assay. $4 \times 10^4$ of cells (per well) were resuspended in 100 µl of serum-free medium (SFM) and loaded into the upper chamber of the CIM-Plate 16, while the lower chamber was filled with complete medium containing 10% FBS. For the growth assay, $2 \times 10^4$ of cells (per well) resuspended in 100 µl of culture medium were added to the E-Plate 16. The CIM-Plate 16 or E-Plate 16 was placed onto the RTCA-DP Analyzer inside the incubator at 37 °C for measurement at 15 min intervals. The growth or migration curves were recorded and the cell index slope (1/hr) for a specific time period/range was calculated according to the RTCA Software Manual (Version 1.2). For the IncuCyte system, cells were seeded in 96-well plates at a density of 5000 cells per well and incubated for 100–148 h, scanning every 4 h.

## Immunofluorescence staining

Cells grown on glass coverslips were fixed in 4% paraformaldehyde for 15 min, and then permeabilized in 0.2% Triton X-100 for 30 min. The cells were subsequently blocked in PBS containing 5% BSA for 12 h, before incubating with the primary antibody at 4 °C overnight. Following incubation with the secondary antibody in PBST at room temperature for 1 h, nuclei were counterstained with DAPI (Thermo, P36931). The coverslips were then mounted onto slides. Images were visualized and collected using a Leica SP8 confocal microscope.

## Xenograft tumor model

Each of the H1299 and A549 stable cell lines (at a final concentration of $2.5 \times 10^6$ cells/each) was injected subcutaneously into 6 week-old BALB/c nude mice (all male gender) individually. Mice were bred and reared under specific-pathogen-free (SPF) conditions with a 12 h light-dark cycle at 20 - 22 °C and 50Accelerated transcription of PRPS1in X-linked overactivity of normal human phosphoribosyl pyrophosphate synthetase60% humidity at the animal facilities of Shanghai Jiao Tong University Medical. Mice were euthanized after 4 weeks, and tumors were weighed and photographed. All animal studies were conducted with the approval and guidance of the Shanghai Jiao Tong University Medical Animal Ethics Committees.

## Metabolic flux analysis

For metabolic flux analysis, $7 \times 10^5$ of cells (per well) were seeded onto a six-well plate inside the incubator at 37 °C for 24 h. Then the culture media were replaced with tracer chemically defined medium, which consisted of glucose-free 1640 (Gibco, 11879020) supplemented with $^{13}C_6$-glucose (Cambridge Isotope Laboratories, CLM-1396-1) or $^{13}C_1$-methionine (Sigma, 299146) at 37 °C for 1, 4, and 6 h. Afterwards, cells were collected and stored at -80 °C before LC-MS/MS analysis.

## LC-MS/MS evaluation

For LC-MS analysis, the samples were treated with 200 µL of cold methanol/$H_2O$ (4:1, v/v) for 1 h and centrifuged at 15,000 rpm at 4 °C for 20 min to obtain supernatants. The supernatants were collected and for analysis using a UHPLC system (1290 Infinity LC, Agilent Technologies) coupled to a Triple Quad 6500 System (AB Sciex 6500). The stability and repeatability of the instrument analysis were monitored using quality control (QC) samples.

## Co-immunoprecipitation (Co-IP) assay

For IP of FLAG-tagged proteins, cells were treated with the lysis buffer containing 20 mM tris-HCl (pH 7.5), 150 mM NaCl, 0.5 mM EDTA, 0.5% NP-40, and 1 × protease inhibitor cocktail (Thermo Fisher, 78439). For IP of FLAG-tagged proteins, whole-cell extracts were incubated with anti-FLAG M2 magnetic beads (Sigma-Aldrich, M8823) overnight at 4 °C. For IP of HA-tagged proteins, whole-cell extracts were incubated with anti-HA M2 magnetic beads (Thermo Fisher, 88837) overnight at 4 °C. Beads were then washed five times with PBS buffer, and proteins bound to beads were eluted in 2 × loading sample buffer before SDS-PAGE. For co-IP of endogenous PRPS2 with endogenous MAT2A, whole-cell extracts were incubated with 1Accelerated transcription of PRPS1in X-linked overactivity of normal human phosphoribosyl pyrophosphate synthetase4 µg of anti-PRPS2 or normal rabbit immunoglobulin G (IgG) (Sigma, I5006-10 MG) overnight followed by incubation with Protein G Plus/Protein A-Agarose (Sigma, IP10-10 ML). Beads were washed five times with PBS, and proteins bound to the beads were eluted in 2 × loading sample buffer before SDS-PAGE (Supplementary Table 5).

## Cycloheximide chase

H1299 cells with stable ectopic expression of PRPS2, PRPS1, or an empty vector (EV) were seeded into six-well plates and incubated at 37 °C for 24 h. Subsequently, 300 µg/ml CHX was added to the cell medium at different time points before harvesting. Cell samples were then subjected to western blot analysis, and the protein expressions were quantified by using ImageJ software.

## Cellular protein thermal shift assay

HEK293FT cells were transiently transfected with HA-GFP-MAT2A, Flag-PRPS2, Flag-PRPS1, or EV for 48 h. The cells were then resuspended in PBS buffer, and aliquoted into PCR tubes (100 µl/tube). Samples were heated to the designated temperature (40-73 °C) for 3 min. After centrifugation, 10 µl samples were loaded into the gel for western blot analysis.

## Analysis of TCGA and GEO data

The online database Gene Expression Profiling Interactive Analysis (GEPIA, http://gepia.cancerpku.cn/index.html) was used to analyze the RNA sequencing expression of PRPS1 and PRPS2 in lung adenocarcinoma (LUAD) and breast invasive carcinoma (BRCA) patients based on The Cancer Genome Atlas (TCGA) and the Genotype-Tissue Expression (GTEx) projects. The overall survival analyses were performed by using the GEPIA web tool, and the recurrence-free survival analyses of the BRCA patients were executed by using the KM plotter (http://kmplot.com/analysis/) web tool. The microarray gene expression profiling data sets (GSE33356, GSE43458, GSE10192, GSE7670, GSE32867, GSE1037, GSE75037, GSE27262) were downloaded from the Gene Expression Omnibus (GEO, https://www.ncbi.nlm.nih.gov/geo/) database.

## Reporting summary

Further information on research design is available in the Nature Portfolio Reporting Summary linked to this article.

# Data availability

The atomic coordinates and structure factors for the crystal structures generated in this study have been deposited to the Protein Data Bank (PDB) under the accession codes 8YPY (PRPS2), 8YPZ (PRPS1), and 8YQ0 (PRPS1 chimera). The online database Gene Expression Profiling Interactive Analysis (GEPIA, http://gepia.cancerpku.cn/index.html) was used to analyze the RNA sequencing expression of PRPS1 and PRPS2 in lung adenocarcinoma (LUAD) and breast invasive carcinoma (BRCA) patients based on The Cancer Genome Atlas (TCGA) and the Genotype-Tissue Expression (GTEx) projects. The overall survival analyses were performed by using GEPIA web tool, and the recurrence-free survival analyses of the BRCA patients were executed by using KM plotter (http://kmplot.com/analysis/) web tool. The microarray gene expression profiling data sets (GSE33356, GSE43458, GSE10192, GSE7670, GSE32867, GSE1037, GSE75037, GSE27262) were downloaded from Gene Expression Omnibus (GEO, https://www.ncbi.nlm.nih.gov/geo/) database. Source data are provided with this paper.

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

## Acknowledgements

This project was supported by grants from the National Natural Science Foundation of China (Nos. 22077081, 21722802, 91853118 to Liang Zhang; 22107067 to Lin Zhang; 82230100, 32271310 to J.X.Y., and 82273143 to X.Z.); Science and Technology Commission of Shanghai Municipality (No. 22S11900600 to Liang Zhang); "Shuguang Program" supported by Shanghai Education Development Foundation and Shanghai Municipal Education Commission (No. 20SG16 to Liang Zhang); Innovative research team of high-level local universities in Shanghai (No. SHSMU-ZDCX20212702 to Liang Zhang). We thank the staffs from BL19U1 and BL18U beamlines of National Facility for Protein Science in Shanghai (NFPS) at Shanghai Synchrotron Radiation Facility (SSRF) for their assistance during data collection, the supports from the Instrument Analysis Center of Shanghai Jiao Tong University for mass spectrometry in protein analysis, and the Core Facility of Basic Medical Sciences at Shanghai Jiao Tong University School of Medicine for LC-MS/MS data collection and analysis. We thank Dr. Gerard Bricogne and Dr. Ian J. Tickle from Global Phasing Ltd. for their kindly help in diffraction data analysis and discussion.

## Author contributions

Liang Zhang and Lin Zhang conceived the original idea and designed the experiments; Lin Zhang preformed the biochemistry and structural experiments; Lin Zhang and X.Z. performed the cell biology and mice assays; Lin Zhang, J.Y.H., and T.T.L. performed the LC-MS/MS based experiments; H.W. provided all the clinic lung adenocarcinoma specimens; Lin Zhang performed the western blot and qPCR assays against the clinic specimens; Lin Zhang, X.Z., H.Z.C., A.Z., H.W., J.X.Y., and Liang Zhang wrote the paper. All authors discussed and commented on the manuscript.

## Competing interests

The authors declare no competing interests.
