## [Transparent Peer Review file · Nature Communications]

PRPS2 Enhances RNA m6A Methylation by Stimulating SAM Synthesis Through Enzyme-Dependent and Independent Mechanisms

Corresponding Author: Professor Liang Zhang

Version 0:

Reviewer comments:

Reviewer #1

(Remarks to the Author)

The manuscript "PRPS2 Enhances RNA m6A Methylation by Stimulating SAM Synthesis Through Enzyme-Dependent and Independent Mechanisms" uncovers a new pathway in an unexpected setting, namely a PRPP-synthesizing isoenzyme that is not subject to feedback inhibition, and which, upon (over)expression enhances ATP production, from the increased ATP enhances SAM production, and the increased SAM levels are then preferentially used to methylate polyA RNA (mostly mRNA), in turn associated to tumor growth.

I find the context stunning, and against my initial skepticism, most of the data were convincing. Some of the above connections look implausible at first sight, be the authors have indeed elaborated experimental support for the different steps in this chain, including metabolomics with stable isotope labeling, and structural biology of the respective enzymes, coupled to mutational analysis and associated biochemistry.

To me the story is convincing, except for two parts, that in my opinion, need better elaboration:

1. ATP levels: increased ATP levels can be a result of many circumstances, and starvation would e.g. be expected to lower them (in favor of ADP). How does this fit in? Does starvation indeed lower m6A? and tumor growth? Can the stimulation of SAM synthesis be dissociated from ATP increase and still promote m6A synthesis?

2. What is the rationale/hypothesis, that the mettl3-14-WTAP complex should consume more of the "add-on" SAM than the 200-300 other MTases in the cell? I hold this an important question that needs a plausible answer.

Altogether, I support eventual publication after these issues are solved.

Reviewer #2

(Remarks to the Author)

In the article "PRPS2 Enhances RNA m6A Methylation by Stimulating SAM Synthesis Through Enzyme-Dependent and Independent Mechanisms," Zhang and colleagues link human phosphoribosyl pyrophosphate 2 (PRPS2) to RNA m6A methylation through both the enzyme activity of PRPS2 and the direct interaction of PRPS2 and methionine adenosyltransferase 2A. The authors show increased abundance of PRPS2 in cancerous tissues when compared to normal tissue in a lung cancer patient cohort. Using data from two repositories, they also identify elevated levels of PRPS2 in several cancers and a correlation of high PRPS2 with decreased patient survival. The authors then evaluate the levels of PRPS2 in human lung cancer cell lines, finding a mix of high and low PRPS2 expression. Using a variety of cell and metabolic flux assays, the authors argue that overexpression of PRPS2, but not PRPS1, in lung cancer cell lines with low endogenous PRPS2 promotes further transformation of the cancerous cells through the increased production of PRPP and downstream nucleotides. The authors use biochemical assays to show lower catalytic activity but resistance to feedback inhibition for the wild type PRPS2 enzyme when compared to PRPS1. Comparing structures determined by X-ray crystallography, the authors argue the allosteric "Loop2" insertion of PRPS2 occupies the binding site of the inhibitor GDP, preventing inhibition. The authors support these arguments by creating PRPS chimeras (adding the PRPS2 Loop2 insertion to PRPS1 or removing it from PRPS2) and testing the protein in various assays. Using knockdown experiments, the authors suggest that PRPS2 is also affecting the cancer properties of the cell lines through catalysis-independent mechanisms. The

authors show that PRPS2 directly interacts with methionine adenosyltransferase 2A and suggest this interaction stabilizes methionine adenosyltransferase 2A which leads to an increase in both S-adenosylmethionine, a major methyl donor, and RNA m6A methylation, an oncogenic signal in malignant cancers.

The article presents cross-disciplinary findings that further support a role for PRPS2 in cancer progression and demonstrates notable differences between human PRPS1 and PRPS2. However, the study requires improvement. The paper does not provide enough information to evaluate the authors claims about the positioning and function of the allosteric "Loop2" and the changing of the active site size. The data suggest that PRPS1 may have some overlapping effects, but the authors do not address this trend in their data. Later in the paper, it is unclear why the authors chose to use a subset of cell lines for each xenograph experiment. The authors use a Student's t-test for most of their statistical analyses outside of the Kaplan-Meier curves, but many of these are multiple comparisons, where a t-test is inappropriate. Below, please find a list of remarks to review.

Major Remarks:

1. The authors argue that PRPS2, but not PRPS1, promotes several cancer cell traits (cell proliferation, colony formation, etc). However, many of the quantifications of these traits suggest that PRPS1 also increases these traits, though to a lesser degree (Fig. 2b-c,e-f and supplementary figures), which counters their argument. The authors should discuss these results and their implications in the paper.
2. While the authors evaluate the expression of PRPS2 by western blot for the patient cohort shown in Fig. 1b, they do not show PRPS1 levels or total PRPS levels. While it might not be possible to distinguish PRPS1 from PRPS2 (due to lack of specific antibody for PRPS1), the authors could compare the total PRPS levels in normal versus tumor tissue. A similar experiment could be performed to define the levels of PRPS1 in the lung cancer cell lines. Overall PRPS levels should be elevated in tumor tissues and cell lines if PRPS1 levels remain the same while PRPS2 levels alone increase.
3. Many of the panels throughout the figures only show partial statistical analysis for the multiple comparisons within the plots. Additionally, unpaired two-sided Student's t-tests were used to evaluate significance for most plots. Student's t-tests are inappropriate for multiple comparisons; another statistical method should be used for evaluating significance. In addition, the p-value indicator for all comparisons should be shown or defined if not shown. While the asterisk values are defined in the methods, these are currently supplementary material. The authors should define them in all figure legends.
4. The authors extensively discuss the allosteric Loop2 in their results section and draw conclusions from its position and the potential interactions that some of the side chains make with amino acids outside of the loop. However, the authors do not show examples of electron density for Loop2 from any of their structures. In the validation report for PRPS2, many residues from these loops are flagged as containing geometric outliers or flagged for poor fit to the electron density or both. This is also true (though to a lesser degree) for the PRPS1 chimera structure containing the PRPS2 allosteric loop. At a minimum, the authors should add panels showing the fit of Loop2 in the electron density for all structures and specify which of the protomers are being shown in these figures and other figures containing structures. Ramachandran plot numbers (favored/allowed/outliers), percentage of rotamer outliers, and the Molprobit and Clash Scores should be added to Supplementary Table 4 and this table should be moved to the main text. The authors may want to revisit their models and improve the geometry unless there is very strong evidence in the map that the loop residues adopt unusual geometries.
5. The authors suggest that Loop2 positioning in PRPS2 decreases the catalytic pocket size, but they do not provide any measurements. The authors should generate these measurements by using software like CASTp (<http://sts.bioe.uic.edu/castp/index.html?2cpk>) or similar better support their argument.
6. The authors use a variety of lung cancer and other cell lines for their cellular and xenograph experiments. Specifically, they use lines H1299 and A549 where they overexpress wild type or chimeric PRPS1 or PRPS2 to demonstrate how PRPS2 transforms the cells. However, for some experiments, only a subset of the conditions are shown. For example, in Fig. 5f, why did the authors choose to use the PRPS2 and the PRPS2-chimera lines in H1299 but PRPS1 and the PRPS1-chimera lines in A549 for the xenograph experiments? This should be explained in the text.
7. The authors should state the sex of the mice used in the xenograph experiments in the methods (all male/female or equal mix of both sexes).
8. The authors argue that the presence of PRPS2 slows the degradation of MAT2A, but these experiments (CHX and CETSA, Fig. 6g-h) are accomplished by overexpressing PRPS2 in cancer cell lines. However, overexpression of PRPS2 may change MAT2A degradation kinetics independent of a direct interaction. To rule this out, the authors could compare the degradation of MAT2A in the same cell lines with another protein overexpressed. A good comparison would be overexpression of wild-type PRPS1, which would also support the effect being specific to PRPS2 and not the result of PRPS activity.
9. The authors do not provide methods for the cycloheximide chase or cellular thermal shift assays. These should be added.
10. The authors show immunostaining experiments in Supp. Fig. 13e, stating that the results show colocalization of PRPS2 and MAT2A. However, in the images provided, both proteins look like they are expressed throughout the cell, suggesting overlap is coincidental rather than the result of a specific interaction.
11. The authors use the inhibitor STM2457 to ask which methyltransferase complex is responsible for RNA m6A methylation. The molecule specifically inhibits methylation by WTAP/METTL3/METTL14 complex. The authors argue that "Treatment with STM2457 significantly reduced RNA m6A levels in PRPS2 overexpressing cells to those observed in control and PRPS1 overexpressing cells" (lines 319-320). However, treatment with the inhibitor reduces methylation in all conditions, not just the methylation in the PRPS2 overexpressing cells. Additionally, both plots (Fig. 8e-f) lack statistical analyses. Would the authors expect the difference seen among control/PRPS1 versus PRPS2 in the DMSO conditions for these plots to be robust enough to be detectable in the inhibitor conditions? Is it possible that the inhibited conditions have reached the limit of detection for these experiments?
12. In the discussion, the authors state, "Consequently, our study underscores the potential of targeting PRPS2 enzymatic activity to halt tumor growth and defines a novel vulnerable target in cancer metabolism essential for proliferation across various cancer types." PRPS2 involvement in cancer is not a novel finding (see 2014 Cunningham et al, Cell;

10.1016/j.cell.2014.03.052, which the authors do cite elsewhere) and the statement should be amended accordingly.

Minor Remarks

13. Error bars and the metric of single points (average?) are not defined in some figures (example: Supplementary Figure 6, plots). Please define for all figures.
14. In a few places (Fig. 1b, Fig. 2b, etc), panels are split in ways that make it difficult to know that the subcomponents are part of the same panel. Please update the figure layouts to avoid this confusion.
15. The figure legend for Fig. 1b mentions that individual data points and mean abundance levels are shown, but these do not appear to be present in the heat maps or lower panel next to panel 1e.
16. Fig. 1d, left panel has two lines at the bottom of the plot in each column. The figure legend should indicate what these lines are showing. Additionally, it would be better if both panels of Fig. 1d were on the same y-axis scale.
17. Full blots or full relevant lanes should be included as supplemental material for all westerns shown in the paper (unless journal standards dictate otherwise).
18. It would be helpful if the authors define what the y-axes of the growth and migration plots represent within the first figure legend where the plots occur. What is meant by "growth slope (1/hr)" and "migration slope (1/hr)?"
19. The authors state, "As anticipated, the levels of newly synthesized R5P via the glucose pentose phosphate pathway significantly decreased within a 1-hour timeframe compared to control cells (Figure 3b)." It should be clarified why this is anticipated.
20. The statement found in lines 169-172 is unclear. "These findings suggest that PRPS2, but not PRPS1, enzymatically stimulates the utilization of R5P, increases metabolic flux abundance, and enhances downstream end-product ATP/GTP production through the purine biosynthesis pathway." Fig. 3b suggests PRPS1 also consumes R5P in these conditions. What is meant by "metabolic flux abundance?" Please clarify.
21. The statement in lines 177-180 should be rephrased. "However, PRPS2 shares 94% amino acid identity with PRPS1, except for three non-conserved residues (V103, G104, E105, referred to as 3AA hereinafter) located in the Loop2 region of PRPS2, which are absent in PRPS1 and represent major variations in their amino acid sequences." As written, it is ambiguous and suggests that the only difference between PRPS1 and PRPS2 is the 3 amino acid insertion, which is incorrect.
22. In line 247/248, Fig. 5 is indicated instead of Fig. 6.
23. There are arrows in Fig. 6h. What are they indicating?
24. The y-axes in Figure 8 vary in that they do not all show the full scale. It is unclear why the authors sometimes show the full axes (Fig. 8d) and sometimes do not (Fig. 8c).
25. Supplementary Table 1 contains column labels that are ambiguous. Please define "pT," "pN," "M," and "pTNM stage."
26. Supplementary Figure 5 does not define the dotted lines in the plots. Please update.
27. Supplementary Figure 7f: the white dashes on a red/white/blue figure are very difficult to see. Please update.
28. It is unclear what Supplementary Figure 11h and j are showing; both the PDF on the screen and the printed version just look like blue circles.

Thank you for the opportunity to read the manuscript.

Reviewer #3

(Remarks to the Author)

Reviewer #4

(Remarks to the Author)

This work is clearly worthy of research and interesting both from mechanistic basic science and disease perspectives. A nice integration of structure/function biochemistry and various cell biological approaches, including disease relevant models, constitutes positive strengths of this paper. However, the manuscript in its current form has several issues, as key conclusions are not well backed up by the data shown. Alternative considerations with a more holistic view would also be beneficial. Experimental design and choice of control constructs and rationales should also be brought up to snuff. I hope the authors will do their due diligence to sure up the science. I'd be happy to review a revised paper that addresses the following concerns/queries in due course as Editors see fit.

Major:

The model is that PRPS2, and not the more well-studied PRPS1, promotes disease phenotypes through enzymatic and non-enzymatic contributions. The former is claimed to arise through a sustained increase in ATP production enabled by isoform-selective resistance to feedback inhibition (ADP/GDP negative allosteric feedback) along the PPP. The latter moonlighting/non-enzymatic function stems from selective stabilization of the rate-limiting SAM-biosynthesis enzyme MAT2A, claimed to co-opt the overproduced ATP (from the above), to drive upregulated SAM production which then results in the observed disease-phenotypes linked to m6A hypermethylation. The model is at first glance stimulating, but in the context of numerous intersecting/inter-dependent pathways that anabolize as well as catabolize these commodity metabolites, the authors do need to take a step back and take a high-level perspective and critique their own interpretations

with a more critical set of eyes..

ENZYMATICALLY-DRIVEN PHENOTYPIC OUTPUTS:

1. what are the K_m 's of MAT2A for ATP, and for Met, and approx. cellular concentrations of ATP and Met in relevant OE systems/cell lines being investigated. The relative values of these parameters vs. how much of the increase in ATP is actually being gained from the loss of negative feedback inhibition, do need to be factored in. For instance, with cellular ATP ~10 mM, it is likely well above K_m of MAT2A for ATP, so MAT2A is likely operating on a pseudo first-order mechanism on [Met] with ATP in excess. So, is the role of enzymatic contribution through ATP upregulation feeding into SAM synthesis still functionally relevant for your model? (Engaging in K_m for Met is also relevant to see how critical stabilization of MAT2A really is in attributing to PRPS2 moonlighting: see section further below)
2. in isotope tracing experiments, the relative flux / rate analysis along glycolytic and mitochondrial TCA-cycle (including assessment of the effects on key rate-limiting enzymes such as ATP synthase), which are the two major sources of cellular ATP production, should be analyzed / quantified to see if some differential effects are being observed through these pathways as opposed to PPP-dependent ATP. The authors would also need to rule out that overexpression (OE) of certain constructs is not selectively perturbing those primary ATP synthesis pathways in the cell. The authors should use knockdown/knockout of key enzymes in the pathway, as opposed to multiple OE systems to draw biological conclusions. OE is known to be perturbing/invasive to native biology and often introduces artefacts that we are often blind to, and not easily ruled out by WT / mutant OE comparisons.
3. Can exogenous adenosine stimulation, known to raise cellular [ATP] rapidly (within minutes), phenocopy the output (against appropriate control constructs), to further corroborate the overall increased ATP production truly being necessary for the phenotype
4. Fig.1a and text indicate both ADP/GDP are negative feedback regulators but the loss of feedback loop was only shown for GDP and not for ADP? Furthermore, given the poor catalytic efficiency of PRPS2 [~15 fold lower than PRPS1, I assume the authors mean k_{cat}/K_m (?)], in the context of the cell (even assuming that the loss of feedback inhibition is for both ADP and GDP), is this even sufficient to perturb the large amount of ATP molecules we have in the cell (and also directly needed by MAT2A)? I understand the tracer assay shows time-dependent expected increase but this assay does not easily tell us anything about how much perturbation / relative quantities, are being observed/perturbed against the bulk ATP pools (plus, these tracer assays are highly sensitive to pick up modest changes).

MOONLIGHTING FUNCTION-ASSOCIATED PHENOTYPIC OUTPUTS:

1. Final section "PRPS2 promotes m6A methylation in lung adenocarcinoma": the punchline on m6A modification needs to be strengthened.
 - (i) first, it is not strictly valid, or at least not rigorous, to compare the effects of the involvement of Protein A vs. B, etc., by assessing the outcomes from small-molecule inhibition on Protein A, but by contrast, effects of knockdown on Protein B! Unfortunately this is what the authors had done for nailing down on METTL isoform! For a start, all small-molecule inhibitors have off-target consequences (many are simply yet-unknown). At least with KO/KD approach, one can minimize off-target effects, using different sets of on-target shRNAs/gRNAs. Thus, the assay where the authors pinpoint the phenotype to a specific METTL isoform/complex should be done using several on-target shRNAs/gRNAs (against several non-targeted sh/gRNA controls) for each relevant METTL isoform.
 - (ii) data from this section require comparisons with rescue loop/point mutants as well as enzymatically dead mutants to tease apart enzyme dependent vs independent functions as claimed. Currently the data are limited to comparison between WT PRPS1 and PRPS2 only.
2. Moonlighting function involving MAT2A association should be corroborated by "knock-in" of enzymatically-dead mutants of PRPS2 and PRPS1. First, the endogenously present protein variant (even if they may be undetectable on WB) could still bear high enzymatic or non-enzymatic functions, especially if they can confer dominant phenotypes on the overexpressed variant. Thus, reconstituted system in the KO background gives a cleaner dataset in general, without potential interference from native copy. The authors should demonstrate a more solid set of evidence in a non-OE system. These issues concern the following sections:

"PRPS2 stabilizes MAT2A in a non-enzymatic manner"

"PRPS2 enhances Met meabolic cycle upregulation »"

data in these sections need to be rigidified by replication and inclusion of enzymatically-dead construct as positive control. Similar to above, TCA and glycolytic flux rates could be analyzed
3. "stabilization of MAT2A directly influences utilization of Met and ATP, as well as synsthesis of SAM" –this is not necessarily true but it depends on enzymatic activity of stabilised pools of MAT2A.
 - (i) the authors need to back this up by measuring the relevant activity (in lysate originating from these cells) over a time course of stabilization, against appropriate controls.

(ii) t1/2 of MAT2A needs to be quantified (with appropriate SEM / error analysis/ statistical-treatment), from CHX assays comparing various relevant OE constructs, including catalytically-dead construct. (Also does PRPS1 +3AA K153Q mutant fail to bind MAT2A?)

Stand-alone claims that are scientifically not valid/correct and it is misleading.

1. Line 159-161: "only PRPS2 OE, but not PRPS1, promoted the synthetic generation of PRPP, indicating accelerated consumption and catalytic utilization of R5P upon PRPS2 ectopic expression (Fig 3C)."

Line 166-167: "again, only PRPS2 OE predominantly enhanced the production of...."

Line 169-171: "these findings suggest that PRPS2 but not PRPS1 enzymatically stimulates the utilization of R5P, xxxxxx"

We cannot claim any of these points, unless you have shown that enzymatic activity (normalized by expression level differences, if any) of these two isoforms, under these conditions in these cell lines, is equally roughly equal. These data need to be provided or that this sentences as such above need to be removed. (NOTE: the fact that PRPS1/2's in vitro enzymatic activities, in the absence of feedback regulation, are vastly different, which was described in the immediately following section, makes it even more confusing and worrying for the readers. Given that feedback regulation loss only came up later, the first two sections read as if the logic is not there either). Alternatively, the authors could also input assumptions. Or that the authors may consider restructuring the order and logic of the results section entirely.

2. Line 209: "PRPS2 did not bind GDP (Kd > 200 uM)"

– incorrect statement to make such a general bind / no bind decision. Several essential metabolic enzymes have Kd/Km several tens to hundreds of micromolars but they are still positively and negatively regulated by these dN(D)TPs/N(D)TPs....

Data presentation-, interpretation-, and analysis-related questions:

1. In all OE lines they created, the authors should do quantitative assessment of expression level differences of various constructs as well as enzymatic activities in cell lysates. This is because it is well known OE induces artefacts – many of the molecules of enzymes do not have the right activity or proper folding. The wild-type, rescue mutants and enzymatic dead mutants should be compared for both enzymatic activity and expression level differences in each overexpressed system before necessary conclusions can be made

2. Quantification and appropriate statistical treatment across 3 independent triplicates should accompany all blots and all data in reality. That has been not the case in this paper. These are minimum required standards in most peer-reviewed broader-readership journals. Please fix across all Main and SI data / blots shown. (Also, at some point in one of the main fig legends, the authors wrote the following " the results were repeated (by) 3 times... and one of them was shown here and the other two shown in SI xxx" but this should be explicit across all data

3. the whole time-courses should be shown for all growth and migration assays, instead of bar plots of designated 1/h slope (e.g., Fig 2B, E, F).

4. BrdU/EdU flow-based assays are more sensitive/reliable for proliferation rate and also more immediate effects can be captured compared to the assays used by the authors: is there a reason why the authors go for phenotypic assessments over longer-duration: is there any difference on cell cycle and DNA replication, especially if the authors are also claiming enhanced / sustained ATP overproduction and NTP pools perturbation?

5. Fig 6, co-IP blots would benefit from cross comparisons and critical (re)evaluations: e.g., PRPS2 sometimes show multiple bands with similar intensity but in other cases only one major band. The same can be said for MAT2A. Legends make no comments on what appears to be inconsistent behaviors of these proteins. Once again, as mentioned above, all replicates (n=3 independent runs) should be shown openly as supporting full-view blots data and quantification and proper statistical analysis should be provided in all cases, so readers can better understand the extent of run-to-run variability. Furthermore, KO/KD lines should be used to corroborate the expected loss of signal. PRPS1 should be probed in all relevant blots, beyond PRPS2, to rule out isoform specific binding to endogenous MAT2A.

6. Results section and key main Fig 1b began with patient samples being claimed as 'randomly selected' but if you zoom into the selected samples marked by arrows, it is quite clear that the selection is far from random but selected for samples that are going in the "right way" as desirable. While it may look more fancy to start with human cancer patients, it's better the authors make a stronger emphasis on TCGA/GEO database data sets. Alternatively, they should provide data from patient samples that have been genuinely selected in an unbiased way: e.g., pick the samples that correspond to the bands that don't look so obviously different between 'normal' lane and 'tumor' lane in the authors' desirable direction...

7. On structural data analysis: to be able to compare multiple structures meaningfully, the growth conditions (e.g., soaking vs co-xtalization) being the same and presence of identical substrates/cofactors/metal ions, etc., are necessary for the readers

to know if these are even valid comparisons. These parameters should be briefly but clearly mentioned alongside any limitations, for the 3 structures the authors compare from line 173 to 196

8. Final paragraph of discussion: the authors could make some realistic projections of ways to potentially target PRPS2 in isoform specific manners based on their structure/function/pathway level understanding thus far. Could a specific binder to selectively inhibit PRPS2 leveraging these differential residues/loops they have identified structurally? Alternatively PPI inhibitors to suppress PRPS2-MAT2A association, and several others data-guided proposals could be interesting here? Currently the Discussion is largely a repeat of the later part of the introduction and the end paragraph of Discussion could end on a high note with some specific lines of proposal: currently it is a bit mundane and general. And therapeutic potential is simply thrown in there more like an after-thought...

9. Reading the manuscript, it gives readers a strong sense that results description is not following the order of experimental investigations the authors had undertaken, which is common for many projects. But I feel that the authors could do a better job of filling in the logical flow and segues between different sections. For instance, currently, some of the enzymatically-dead mutants and +/-3AA etc. rescue mutants come and go in a rather random/illogical manner from one section to the next. All relevant controls should be included and missing comparisons be added across all sections, as stated elsewhere.

Additional points:

1. Abstract: the sentence spanning Line no. 35 to 39 in the merged manuscript doc needs an upgrade. The jump to lung tumor came in too abruptly and it sounds as if this mechanism is the single cause. It would be more logical and broader/accessible if readers can be primed up for lung-specific tumor relevant investigations earlier on in the abstract
2. Out of curiosity, could the authors include a phylogenetic analysis across several closer species across taxa for PRPS2 and PRPS1. Do these key residues on PRPS2 exist in mice chimps etc – are there lung disease related reports in those models.
3. The usage of the word non-conserved vs conserved could be clearer: inter-isoform conservation between PRPS1 and PRPS2 (such as in Abstract), vs. cross-species conservation across orthologs of one isoform across for instance man and mice (such as in line 97 of last paragraph in Introduction).
4. PRPS3 in human is limited to expression in testicles so this work does not invoke into this isoform. But Ref 31 and 34 may relate to Y-chromosome specific disease, yet they are on PRPS2, instead of PRPS3? Looking at Fig S1, PRPS3 is akin to PRPS1 at least in terms of amino acid sequence homology?
5. Introduction should briefly engage/elaborate on 'functional redundance': e.g., can KO animals of PRPS1 survive/rescued by PRPS2, and vice versa?
6. In OE assays, what is "ctrl" is not clear. Is this empty vector (EV), or OE of a non-native gene? EV is a more appropriate control
- 7: Line 198-200: requires quantitative description – « exhibited switched enzymatic activities “ -- is it now 15-fold more active? Please compare with what you mentioned above (quantitatively). “deletion of the a12 helix from PRPS...” requires clarity – which isoform?
8. Cell line changes in a few cases are not explained well: HEK293T and 293FT vs H1299 in section: Line 251 to 271

Writing:

may I also request that the language (including grammatical issues) be proof-checked and edited across the board. Just to give one example: throughout legends “randomly chose” should be “randomly chosen”

- Line 149: delete the word 'catalytic' – 'substrate' is sufficient

-Line 176: remove the word "catalyzing"

Reviewer #5

(Remarks to the Author)

This manuscript by Zhang et al. explores the the role and functional contributions of PRPS2, a phosphoribosyl pyrophosphate synthetase isoform involved in the first and rate-limiting step of the purine biosynthesis pathway, in NSCLC. They start by finding that PRPS2 but not PRPS1 is upregulated in human NSCLC samples and that PRPS2 expression correlates with worse survival. They then show that PRPS2>PRPS1 increases growth, migration, and colony formation of NSCLC cell lines. They then do LC-MS and find that PRPS2 overexpression increases GTP/ATP production. Mechanistically, they interestingly find structural differences between PRPS2 and PRPS2 which include a VGE insertion in PRPS2 and K>Q mutation that could explain specific activity of PRPS2 over PRPS1. Making these mutations in each paralog, they find that mutants of PRPS1 relatively phenocopy PRPS2 and mutants of PRPS1 act like PRPS2 with respect

to enzymatic activity, proliferation phenotypes, and migration phenotypes. Moreover, the PRPS2 mutant attenuates tumor growth and the PRPS1 mutant increases tumor growth. They find that PRPS2 knockdown decreases growth. They then also find that PRPS2 binds to MAT2A to promote MAT2A stability which increases SAM leading to m6A mRNA methylation through METTL3/14.

This is overall a really strong study particularly the mechanistic studies elucidating structural differences between PRPS2 and PRPS1 that contribute to PRPS2 specific activity. Moreover, the specificity of PRPS2 contributing to m6A mRNA methylation is fascinating. I think the mechanistic work is rigorous with appropriate controls to make strong conclusions. I do have a few suggestions to improve the manuscript particularly related to phenotypes associated with suppression of PRPS2.

1. Is PRPS2 required for NSCLC survival in NSCLCs that highly express PRPS2? The authors show in 2 NSCLC cell lines in Fig. 6A-B that suppression of PRPS2 with shRNAs decreases growth of PC9 and H1792 cells. While these phenotypes are significant, it is unclear whether this is an on-target effect of losing PRPS2 or related to an off-target effect. What does CRISPR inactivation of PRPS2 and PRPS1 show in PC9 and H1792 cells? Are they able to express an shRNA or sgRNA resistant cDNA of PRPS2 and rescue the proliferation phenotypes in Figure 6A-B? Lastly, it would be helpful to show both the CRISPR and shRNA experiments in 1-2 additional NSCLC models with high PRPS2 expression and also see if there are differences in the requirement for PRPS2 in models of NSCLC that have low PRPS2 expression. A major conclusion (last line of their abstract and conclusion of their discussion) is that PRPS2 targeting should be considered in cancers associated PRPS2 abnormalities but this aspect of the paper is not strong as is.

2. M6A RNA methylation-It's fascinating that the authors show in Figure 8 increases in m6A methylation without major changes in other methylation that utilize SAM. This was only done in H1299 cells. These studies should be expanded to another NSCLC cell line with high PRPS2 expression to see if this specificity is generalizable.

Version 1:

Reviewer comments:

Reviewer #1

(Remarks to the Author)

My opinion is still very positive.

One of my two questions has been answered, namely the preferential use of newly synthesized SAM via the lower K_m value of mettl3/14 versus other MTases.

However, even the revised discussion does not provide a clear rationale (nor does the rebuttal letter) wrt to other ATP-depleting physiological events that should, according to the present rationale, overrule the entire pathway when lowering ATP levels. What would happen to RNA methylation when oxphos is lowered by an uncoupling agent? would that inhibit cancer growth, and if not, why? The authors have added the term "newly synthesized" before ATP in many places throughout the manuscript, but that does not yet heal the conceptual problem. Are there indications that the "newly synthesized" ATP is funneled towards SAM synthesis without entering the generally available pool?

Reviewer #2

(Remarks to the Author)

Zhang and colleagues have addressed almost all of my concerns. There is one point where small revisions to the text are needed.

Question-11: The authors show immunostaining experiments in Supp. Fig. 13e, stating that the results show colocalization of PRPS2 and MAT2A. However, in the images provided, both proteins look like they are expressed throughout the cell, suggesting overlap is coincidental rather than the result of a specific interaction.

Response: Thanks for your comments. The purpose of our immunostaining experiments was to check the localization of PRPS2 and MAT2A, as the localization of PRPS2 in cell is unclear so far. Both of our mass-spec and co-IP results confirmed that PRPS2 interacts with and stabilizes MAT2A.

Reviewer Response: While other experiments within this paper suggest that PRPS2 and MAT2A interact, the immunofluorescence-based colocalization assay data (now Supplementary Figure 16) only shows that both proteins are distributed throughout the cytoplasm. This experiment does not "corroborat[e] the Co-IP results and indicat[e] a specific interaction between PRPS2 and MAT2A" (lines 282-284). If the authors are claiming that both PRPS2 and MAT2A can be found in the cytoplasm together, they should state this, rather than claiming colocalization as a support for direct interaction.

Three sentences should be amended to reflect that both proteins localize to the cytoplasm together and this data should not be used to argue for "specific interactions:"

- Supplementary Figure 16. Cell proliferation of PRPS1 shRNA knockdown in H1792 and PC9 cell lines and ****localization**** of PRPS2 and endogenous MAT2A in H1299 cells.

- (e) Immunofluorescence assay showing the ****location**** of PRPS2 and endogenous MAT2A from H1299 cells transfected with FLAG-PRPS2.

- Subsequent immunostaining assays demonstrated that ****transfected**** PRPS2 and ****endogenous**** MAT2A ****localize**** to the cytoplasm (Supplementary Figure 16e).

Any other reference to this data should also be amended accordingly.

Thank you.

Reviewer #3

(Remarks to the Author)

Reviewer #4

(Remarks to the Author)

PDF review attached

The authors have addressed some of my concerns. There are a few pending questions that should be addressed comprehensively for clarity as well as rigor and also to rigidify some of the conceptual underpinnings and to help tie up the loose ends. Providing all these points below can be thoroughly remedied, I would be pleased to recommend publication. I do understand that not all data and assays may necessarily support the conclusions and models, but nonetheless, it would be necessary that alternative possibilities be ruled out and data be interpreted with reservations, care, and rigor, outlining limitations wherever applicable, and showing all acquired datasets openly etc.

a)- the authors made several implications that ¹³C experiments only report on the newly-synthesized ATP arising exclusively from de novo pathway; and also that externally introduced adenosine would not affect the de novo or salvage pathway, and so on. These statements above are limited / incorrect. Referring to a commonly-encountered text-book figure (reproduced below is such a figure that appears in a recent publication, 11th July 2024 Tran et al) illustrating both de novo and salvage pathway, the authors' detected ¹³C amount at the m/z corresponding to ATP, could still arise from pathways that are not exclusively de novo ATP synthesis pathway. For instance, nothing is preventing, in the authors' experimental conditions/cells deployed, for the first set of newly-made ¹³C-labeled ATP molecules, to be transformed into adenosine (see green arrow in figure, via nucleotidase, thereby reducing the ultimate measured amount of total 'newly-made' ¹³-C labeled ATP molecules); nothing is also preventing the so-produced ¹³-C labeled adenosine to also be reconverted, to some unknown extent, to ¹³-C labeled AMP and thus ATP molecules, thus (re)increasing or altering the ultimate measured total ¹³-C labeled ATP molecules; and nothing is also preventing the so-made ¹³-C-labeled Adenosine to ¹³-c-labeled Adenine that can enter salvage pathway (see orange arrow) to some unknown degree, to contribute to ¹³-c-labeled AMP and thus altering the detected ¹³-C labeled ATP being measured in a way that is unknown.....

Indeed these issues are often intrinsic limitations of isotope tracer experiments, but from the rebuttal responses provided in Page 11 and 13 of the rebuttal PDF, it makes me worry that the authors are unaware of these points first of all.

Thus Adenosine is indeed able to influence the newly-synthesised ATP from both de novo and salvage pathway. If one increases adenosine by exogenous addition, for instance, it will raise the cellular ATP levels and should phenocopy what the authors are reporting (my previous Q4). To nail down that this is salvage pathway independent they can easily perform these adenosine-administered experiments in APRT native and KO/KD cells see if phenotypes are replicated.

b) potential interference of endogenous wild-type protein variant that could impose dominant negative effects differentially on overexpressed wt and mutant constructs remains a concern. This is not eliminated by the revisions due to inability to obtain KD/KO lines of this protein. To this end: I wonder if the authors have attempted transient KD by siRNA, instead of near-complete KO or integrated shRNA KD.

Should transient KD still result in complete lack of any level of KD achievable in these cells, then this issue above resulting in entirely overexpression-focused studies, should be mentioned in the paper. Reasons behind why KD/KO lines are inaccessible by both lentiviral CRISPR/gRNA, shRNA as well as siRNAs, also should be briefly engaged in the discussion as part of the usual 'limitations of the study' paragraph..

More generally, some of the limitations and assumptions outlined in the rebuttal should be integrated into the manuscript to help the ultimate readers.

Likewise, for the choice of the use of GDP over ADP should be briefly explained in the relevant results section.

Similarly, technical limitations in measuring the changes in the amount of newly-made ATP in the absence of PRPS2 feedback inhibition should be mentioned.

c) siRNA KD efficiencies of all siRNAs used in fig 8E should be measured using western blot and results shown and quantified and reported in Supplemental Info, against ctrl-siRNA-administered and non-siRNA-administered native cells

d) my Q8 previously asked for quantification of the bands since the changes are minute / modest. The authors have not yet quantified nor provided the quantitative changes to the half-life of protein (derived from appropriate half-life (t_{1/2}) curve fitting) under different conditions. Please quantify the bands across multiple independent replicates along with appropriate

statistical treatments for data in both A and B of Supp Fig S17

e) it is an important due diligence that all antibodies be validated by knockdown/knockout unless they have been previously validated as such by either previous data in the published literature, or by the image data available on manufacturers' websites. Especially that there are multiple bands, and the authors responded by saying that these are due to non-specific bands: how does one 100% sure that any non-specific bands do not accidentally appear on the same/similar MW/kD as the protein being probed. This coincidental issue is far too common for many antibodies (where the band of correct MW fails to disappear after targeted KO/KD!) and has resulted in a lot of data misinterpretations of western blot data with numerous antibodies. Thus the authors should thus do the due diligence of KO/KD validations of Abs that report on their key enzymes and isoforms at least PRPS1/2 and MAT2A, key proteins essential for their mechanistic model. This is the only valid way to verify the outcomes.

f) please also provide full-view blots of all blots shown as supplementary information, which is not an uncommon practice in biological publications.

g) WRITING: writing improvement was requested previously (Q28)-- it would be necessary that professional proof-editor be deployed to help improve the writing, grammatical correctness and idiomatic use of the language across the board. There are several places where language used could lead to incorrect interpretations: just as one-of-many examples, the authors consistently use in the manuscript text and rebuttal document, the word 'in' mistakenly, whenever they tend to mean that something is ("in") consistent with something else. Should the space be accidentally missing between "in" and "consistent", this would mean the complete opposite of what they intend to describe!

I must say I do not quite understand why the authors replicated their responses to the referees in a repeated manner (in some cases already 3 times: see, for instance "We do have looked through the metabolites in multiple metabolic pathways, such as glycolysis, TCA cycle, NADPH metabolic pathway, etc.," on page 11, page 13, and page 16 – one would wonder why?). In general, what was already said in Page 11 (and indeed read by this reviewer with great care and interest) was repeatedly described in subsequent pages, yet once again. This makes the rebuttal look doubly lengthy and of greater importance, it is not helpful and confusing to say the least for the reviewers to sieve through new information if any against repeating texts...

I am also unsure if no. of years spent on a project is a valid yard-stick to justify to what extent the work is rigorous. I understand that the authors have done a large extent of work but it is better that the responses be centered upon scientific points that can be integrated to the manuscript where necessary.

On the flip side, it makes the readers worry that after spending over 8 yrs into this project, why then the authors did not think about clearly possible solutions that would render a lot of these investigations / results to be much more solid: such as, on the multiple reviewers' overlapping questions, where the response has been: "commercial PRPS1 or PRPS2 antibodies could not discriminate PRPS1 from PRPS2".... >8 yrs is several folds more than sufficient time for the authors to raise isoform-specific antibodies in-house or have them be custom-made.

Reviewer #5

(Remarks to the Author)

The authors have substantially improved their manuscript and addressed some of my critiques. I still think the manuscript could be improved with more rigorous genetic experiments aimed at determining whether PRPS2 is actually a dependency in NSCLC lines. I originally asked for orthogonal approaches such as CRISPR or doing rescue experiments with shRNA resistant cDNAs of PRPS2 to make sure their proliferation phenotypes were on-target. PC9 is a cell line that is very easy to CRISPR and therefore these experiments should be feasible.

Version 2:

Reviewer comments:

Reviewer #4

(Remarks to the Author)

I commend the authors for the good effort they have clearly put into what I hope as their final revision round. Based on the authors' rebuttal responses including website links provided, the authors appear to unfortunately think that seeing a band on western blot at the "expected" molecular weight is considered as "validation" for a given antibody. In reality, the only valid method of validation for any antibody is via KD/KO... I note that many Ab's used in this paper (many are commercially available) have thus not been validated. I hope that the authors could keep this important point in mind. I hope that they will take a more cautious approach and improve rigor of Ab validation in their future research. I am happy to recommend the manuscript for publication and wish the authors all good luck for their ongoing research endeavours.

Manuscript number: NCOMMS-24-35483-T

Reviewer #1

The manuscript “PRPS2 Enhances RNA m⁶A Methylation by Stimulating SAM Synthesis Through Enzyme-Dependent and Independent Mechanisms” uncovers a new pathway in an unexpected setting, namely a PRPP-synthesizing isoenzyme that is not subject to feedback inhibition, and which, upon (over)expression enhances ATP production, from the increased ATP enhances SAM production, and the increased SAM levels are then preferentially used to methylate polyA RNA (mostly mRNA), in turn associated to tumor growth. I find the context stunning, and against my initial skepticism, most of the data were convincing. Some of the above connections look implausible at first sight, but the authors have indeed elaborated experimental support for the different steps in this chain, including metabolomics with stable isotope labeling, and structural biology of the respective enzymes, coupled to mutational analysis and associated biochemistry. To me the story is convincing, except for two parts, that in my opinion, need better elaboration:

Question-1: (1) ATP levels: increased ATP levels can be a result of many circumstances, and starvation would e.g. be expected to lower them (in favor of ADP). How does this fit in? Does starvation indeed lower m⁶A? and tumor growth? (2) Can the stimulation of SAM synthesis be dissociated from ATP increase and still promote m⁶A synthesis?

Response: Thanks for your positive comments. (1) In this work, we mainly focused on the newly synthesized ATP through the *de novo* purine synthesis pathway by tracing C¹³ labeled ribose of ATP derived from C¹³ labeled glucose (Figure 3a), which are distinct from those generated through ADP to ATP conversion in multiple energy supplemental metabolic pathways such as glycolysis, and TCA-cycle *etc.* Starvation would be very likely to decrease ATP production through the *de novo* purine synthesis pathway as the glucose deprivation would directly decrease the product R5P through the pentose phosphate pathway, which is one of the major substrates of PRPSs for downstream purine synthesis and ATP production. However, the starvation stress in a short time period would trigger many metabolic circumstances (glycolysis, TCA cycle, and fatty acid oxidation, *etc.*) to compensate the loss of ATP by converting ADP to ATP, and maintain a sufficient overall ATP levels for biological behavior achievements. In contrast, long term starvation would induce cell apoptosis and death through multiple progresses. Previous study showed that m⁶A modification is indeed up-regulated to promote autophagosome formation and lysosomal degradation upon nutrient deficiency in mouse embryonic fibroblasts (MEFs) (*Nat. Commun.*, 2022, 13, 5845). These results suggested that the cellular RNA m⁶A level is highly dynamic regulated by complicated mechanisms, and its variation observed upon starvation in cancer cells could not be explained simply by the lack of ATP and SAM. (2) Simply increasing the cellular level of ATP would not certainly stimulate SAM synthesis. As you mentioned, there are many

ATP utilization circumstances identified so far and SAM synthesis pathway is just one of them. Our results demonstrated that the enzymatic-independent function of PRPS2 is critical for SAM synthesis stimulation by stabilizing MAT2A via protein-protein interactions. Hence, the combination of both enzymatic-dependent and independent functions of PRPS2 enables the utilization of newly synthesized ATP specifically for SAM synthesis and downstream RNA m⁶A modification.

Question-2: What is the rationale/hypothesis, that the mettl3-14-WTAP complex should consume more of the “add-on” SAM than the 200-300 other MTases in the cell? I hold this an important question that needs a plausible answer. Altogether, I support eventual publication after these issues are solved.

Response: Thanks for your comments. (1) In the enzymatic level, METTL3 exhibits ~10-fold higher Km value (~0.1 μM) than other major SAM dependent methyltransferases for DNA, RNA and histone methylation (~1 μM), enabling a higher priority of SAM utilization among the SAM dependent methyltransferases (*Biochem. 2023*, 62, 494; *J. Biomo. Screen.*, 2016, 21, 290; *Nucleic Acids Res.*, 2001, 29, 3506; *Assay Drug Dev. Technol.*, 2013, 11, 227). (2) In the protein expression level, previous studies have demonstrated that c-Myc upregulates the expression level of WTAP accompany with PRPS2 overexpression to enhance the expression levels of WTAP/METTL3/METTL14 complex for RNA m⁶A methylation (*Cell*, 2014, 157, 1088; *Mol. Cell*, 2019, 75, 1147). (3) In the protein regulation level, PRPS2 was found to interact with RNA translation initiation factors such as eIF4E (*Cell*, 2014, 157, 1088), and our co-IP results also observed potential binding of eIFs with PRPS2 (Supplementary Table 5), while METTL3 was found to interact with eIF3 (*Mol. Cell*, 2016, 62, 335). Notably, these eIFs were found to further form a complex with ribosome for mRNA 5'-UTR recognition (*Cell*, 2022, 185, 4474; *Mol. Cell*, 2016, 62, 335). These observations strongly imply that PRPS2, WTAP/METTL3/METTL14 complex, eIFs, and ribosome may form a mega size complex to achieve the specific RNA m⁶A modification for translation initiation and promotion. We have revised the discussion section in the main text accordingly.

Reviewer #2

In the article “PRPS2 Enhances RNA m⁶A Methylation by Stimulating SAM Synthesis Through Enzyme-Dependent and Independent Mechanisms,” Zhang and colleagues link human phosphoribosyl pyrophosphate 2 (PRPS2) to RNA m⁶A methylation through both the enzyme activity of PRPS2 and the direct interaction of PRPS2 and methionine adenosyltransferase 2A. The authors show increased abundance of PRPS2 in cancerous tissues when compared to normal tissue in a lung cancer patient cohort. Using data from two repositories, they also identify elevated levels of PRPS2 in several cancers and a correlation of high PRPS2 with decreased patient survival. The authors then evaluate the levels of PRPS2 in human lung cancer cell lines, finding a mix of high and low PRPS2 expression. Using a variety of cell and metabolic flux assays, the authors argue that overexpression of PRPS2, but not PRPS1, in lung cancer cell lines with low endogenous PRPS2 promotes further transformation of the cancerous cells

through the increased production of PRPP and downstream nucleotides. The authors use biochemical assays to show lower catalytic activity but resistance to feedback inhibition for the wild type PRPS2 enzyme when compared to PRPS1. Comparing structures determined by X-ray crystallography, the authors argue the allosteric “Loop2” insertion of PRPS2 occupies the binding site of the inhibitor GDP, preventing inhibition. The authors support these arguments by creating PRPS chimeras (adding the PRPS2 Loop2 insertion to PRPS1 or removing it from PRPS2) and testing the protein in various assays. Using knockdown experiments, the authors suggest that PRPS2 is also affecting the cancer properties of the cell lines through catalysis-independent mechanisms. The authors show that PRPS2 directly interacts with methionine adenosyltransferase 2A and suggest this interaction stabilizes methionine adenosyltransferase 2A which leads to an increase in both S-adenosylmethionine, a major methyl donor, and RNA m⁶A methylation, an oncogenic signal in malignant cancers.

Question-1: The article presents cross-disciplinary findings that further support a role for PRPS2 in cancer progression and demonstrates notable differences between human PRPS1 and PRPS2. However, the study requires improvement: The paper does not provide enough information to evaluate the authors claims about the positioning and function of the allosteric “Loop2” and the changing of the active site size. The data suggest that PRPS1 may have some overlapping effects, but the authors do not address this trend in their data. Later in the paper, it is unclear why the authors chose to use a subset of cell lines for each xenograph experiment. The authors use a Student’s t-test for most of their statistical analyses outside of the Kaplan-Meier curves, but many of these are multiple comparisons, where a t-test is inappropriate. Below, please find a list of remarks to review.

Response: Thanks for your positive comments. We have revised the manuscript according to your comments, and please find the details below for your review.

Major Remarks:

Question-2: The authors argue that PRPS2, but not PRPS1, promotes several cancer cell traits (cell proliferation, colony formation, *etc.*). However, many of the quantifications of these traits suggest that PRPS1 also increases these traits, though to a lesser degree (Fig. 2b-c, e-f and supplementary figures), which counters their argument. The authors should discuss these results and their implications in the paper.

Response: Thanks for your comment. We have revised the PRPS1 related content in the main text accordingly. Our results demonstrated that GDP inhibits the enzymatic activity of PRPS1 with a IC₅₀ value of 706 μM (Figure 4e). Hence, it is expected to see a moderate level increase of cell proliferation upon stable ectopic expression of PRPS1, as PRPS1 would keep producing ADP/GDP for cell proliferation until the concentrations of ADP/GDP reach certain amounts to shut down the enzymatic activity of PRPS1. In contrast, PRPS2 kept producing excess amounts of downstream products due to the lack of feedback inhibition regulation, and specifically stabilized MAT2A for SAM synthesis stimulation and RNA m⁶A hypermethylation for continuously cell proliferation promotion.

Question-3: While the authors evaluate the expression of PRPS2 by western blot for the patient cohort shown in Fig. 1b, they do not show PRPS1 levels or total PRPS levels. While it might not be possible to distinguish PRPS1 from PRPS2 (due to lack of specific antibody for PRPS1), the authors could compare the total PRPS levels in normal versus tumor tissue. A similar experiment could be performed to define the levels of PRPS1 in the lung cancer cell lines. Overall PRPS levels should be elevated in tumor tissues and cell lines if PRPS1 levels remain the same while PRPS2 levels alone increase.

Response: Thanks for your comments. We performed the western blot experiments for the patient cohort samples in the year of 2019. Frankly speaking, the commercial antibodies we purchased showed no specificity among PRPSs along with significant off-target issues, which were not qualified for the quantification of PRPS1, PRPS2, or total PRPSs expression levels. For the above reasons, we generated homemade PRPS2 specific antibody (Supplementary Figure 2) combined with qPCR assays to look into the PRPSs expression levels. To make the results and conclusions more reliable, we further introduced PRPS1 as a control for subsequent enzymatic, structural, cell proliferation, migration, cellular metabolic, and mouse xenograft studies. All our results confirmed that the function of PRPS2 is significantly distinct from that of PRPS1.

Question-4: Many of the panels throughout the figures only show partial statistical analysis for the multiple comparisons within the plots. Additionally, unpaired two-sided Student's t-tests were used to evaluate significance for most plots. Student's t-tests are inappropriate for multiple comparisons; another statistical method should be used for evaluating significance. In addition, the p-value indicator for all comparisons should be shown or defined if not shown. While the asterisk values are defined in the methods, these are currently supplementary material. The authors should define them in all figure legends.

Response: Thanks for your comments. We have revised the figures and legends accordingly.

Question-5: The authors extensively discuss the allosteric Loop2 in their results section and draw conclusions from its position and the potential interactions that some of the side chains make with amino acids outside of the loop. However, (1) the authors do not show examples of electron density for Loop2 from any of their structures. In the validation report for PRPS2, many residues from these loops are flagged as containing geometric outliers or flagged for poor fit to the electron density or both. This is also true (though to a lesser degree) for the PRPS1 chimera structure containing the PRPS2 allosteric loop. At a minimum, the authors should add panels showing the fit of Loop2 in the electron density for all structures and specify which of the protomers are being shown in these figures and other figures containing structures. (2) Ramachandran plot numbers (favored/allowed/outliers), percentage of rotamer outliers, and the Molprobit and Clash Scores should be added to Supplementary Table 4 and (3) this table should be moved to the main text. (4) The authors may want to revisit their models and improve the geometry unless there is very strong evidence in the map that the loop residues adopt unusual geometries.

Response: Thanks for your comments. (1) We have added a supplementary figure to show the omit electron density map around Loop2 regions of PRPS2, PRPS1 or PRPS1(+3AA) structures, respectively (Supplementary Figure 7). As shown in the figure, the Loop2 regions of PRPS1 and PRPS1(+3AA) were perfectly fitted into the electron densities. Despite the electron density around residues Lys99, Lys100, and Asp101 of PRPS2 Loop2 were weak, it would not change the overall conformation of Loop2, which was further confirmed by the structural superposition of the Loop2 regions between PRPS2 and PRPS1(+3AA) (Figure 4c). (2) We have added these data to Supplementary Table 4 accordingly. (3) We have to keep the table in the supplementary file as there is not enough space in main text (eight figures in main text already) due to the page limitation for journal publication. (4) The resolutions of the structures were relatively low (2.75Å, 3.0Å, and 3.1Å, respectively), which limited the structural refinement accuracy. This is a common issue for the relatively low-resolution structures in protein data bank (PDB), and is acceptable by the policy of PDB. According to our response above, the conformation of PRPS2 Loop2 region has been well confirmed by that of PRPS1(+3AA) structure, as well as subsequent results.

Question-6: The authors suggest that Loop2 positioning in PRPS2 decreases the catalytic pocket size, but they do not provide any measurements. The authors should generate these measurements by using software like CASTp (<http://sts.bioe.uic.edu/castp/index.html?2cpk>) or similar better support their argument.

Response: Thanks for your suggestion. We have calculated the catalytic pocket sizes of PRPS1 and PRPS2 by the server you suggested and revised the main text accordingly. The results suggested that the catalytic pocket volume of PRPS2 (148 Å³) is approximately half of that of PRPS1 (302 Å³) due to the conformation variations of Loop2 region in these enzymes.

Question-7: The authors use a variety of lung cancer and other cell lines for their cellular and xenograph experiments. Specifically, they use lines H1299 and A549 where they overexpress wild type or chimeric PRPS1 or PRPS2 to demonstrate how PRPS2 transforms the cells. However, for some experiments, only a subset of the conditions are shown. For example, in Fig. 5f, why did the authors choose to use the PRPS2 and the PRPS2-chimera lines in H1299 but PRPS1 and the PRPS1-chimera lines in A549 for the xenograft experiments? This should be explained in the text.

Response: Thanks for your comments. We have performed additional xenograft experiments, and revised the main text, Figure 5, and Supplementary Figure 13 accordingly.

Question-8: The authors should state the sex of the mice used in the xenograft experiments in the methods (all male/female or equal mix of both sexes).

Response: Thanks for your suggestion. All the mice we used were male mice. We have revised the methods accordingly.

Question-9: The authors argue that the presence of PRPS2 slows the degradation of MAT2A, but these experiments (CHX and CETSA, Fig. 6g-h) are accomplished by

overexpressing PRPS2 in cancer cell lines. However, overexpression of PRPS2 may change MAT2A degradation kinetics independent of a direct interaction. To rule this out, the authors could compare the degradation of MAT2A in the same cell lines with another protein overexpressed. A good comparison would be overexpression of wild-type PRPS1, which would also support the effect being specific to PRPS2 and not the result of PRPS activity.

Response: Thanks for your comments. We performed the degradation assay for PRPS1 as the control (Supplementary Figure 17). The results demonstrated that PRPS1 does not interact with and stabilize MAT2A.

Question-10: The authors do not provide methods for the cycloheximide chase or cellular thermal shift assays. These should be added.

Response: Thanks for your suggestion. We have added them to the methods section accordingly.

Question-11: The authors show immunostaining experiments in Supp. Fig. 13e, stating that the results show colocalization of PRPS2 and MAT2A. However, in the images provided, both proteins look like they are expressed throughout the cell, suggesting overlap is coincidental rather than the result of a specific interaction.

Response: Thanks for your comments. The purpose of our immunostaining experiments was to check the localization of PRPS2 and MAT2A, as the localization of PRPS2 in cell is unclear so far. Both of our mass-spec and co-IP results confirmed that PRPS2 interacts with and stabilizes MAT2A.

Question-12: (1) The authors use the inhibitor STM2457 to ask which methyltransferase complex is responsible for RNA m⁶A methylation. The molecule specifically inhibits methylation by WTAP/METTL3/METTL14 complex. The authors argue that “Treatment with STM2457 significantly reduced RNA m⁶A levels in PRPS2 overexpressing cells to those observed in control and PRPS1 overexpressing cells” (lines 319-320). However, treatment with the inhibitor reduces methylation in all conditions, not just the methylation in the PRPS2 overexpressing cells. (2) Additionally, both plots (Fig. 8e-f) lack statistical analyses. (3) Would the authors expect the difference seen among control/PRPS1 versus PRPS2 in the DMSO conditions for these plots to be robust enough to be detectable in the inhibitor conditions? Is it possible that the inhibited conditions have reached the limit of detection for these experiments?

Response: Thanks for your comments. (1) Given our results have demonstrated that the excess methylation of RNA m⁶A was mediated via the ectopic overexpression of PRPS2 through the METTL3/14 pathway, the blockage of METTL3/14 would trim down the RNA m⁶A level to comparable levels of that of PRPS1 overexpression or control. In addition to the inhibition assay against METTL3/METTL14 by using inhibitor STM2457 (Supplementary Figure 26), we further performed siRNA knock down assay against either METTL3 or METTL14 (Figure 8e). Both the results confirmed that the inhibition of METTL3/14 by STM245 or siRNA knock down of either METTL3 or METTL14 in PRPS2 overexpressing cell line decreased the RNA

m⁶A levels to comparable levels with that of PRPS1 overexpression or control, suggesting that the excess methylation of RNA m⁶A observed in PRPS2 overexpressing cell line was generated specifically through METTL3/14 dependent methylation. (2) We have revised Figure 8 and figure legend accordingly. (3) These data were indeed within the effective detection range of mass-spec (peak area is higher than the value of 10⁴).

Question-13: In the discussion, the authors state, “Consequently, our study underscores the potential of targeting PRPS2 enzymatic activity to halt tumor growth and defines a novel vulnerable target in cancer metabolism essential for proliferation across various cancer types.” PRPS2 involvement in cancer is not a novel finding (see 2014 Cunningham et al, Cell; 10.1016/j.cell.2014.03.052, which the authors do cite elsewhere) and the statement should be amended accordingly.

Response: Thanks for your suggestion. We have revised the context accordingly.

Minor Remarks

Question-14: Error bars and the metric of single points (average?) are not defined in some figures (example: Supplementary Figure 6, plots). Please define for all figures.

Response: Thanks for your comment. We have revised the figure legends accordingly.

Question-15: In a few places (Fig. 1b, Fig. 2b, etc), panels are split in ways that make it difficult to know that the subcomponents are part of the same panel. Please update the figure layouts to avoid this confusion.

Response: Thanks for your comment. We have revised the figures and legends accordingly.

Question-16: The figure legend for Fig. 1b mentions that individual data points and mean abundance levels are shown, but these do not appear to be present in the heat maps or lower panel next to panel 1e.

Response: Thanks for your comment. The data presented in Fig. 1b were shown as average values of biological triplicates. We have revised the figure legend accordingly.

Question-17: Fig. 1d, left panel has two lines at the bottom of the plot in each column. The figure legend should indicate what these lines are showing. Additionally, it would be better if both panels of Fig. 1d were on the same y-axis scale.

Response: Thanks for your comment. We have revised the figure accordingly.

Question-18: Full blots or full relevant lanes should be included as supplemental material for all westerns shown in the paper (unless journal standards dictate otherwise).

Response: Thanks for your comment. The raw data for the results have been submitted along with the raw data file.

Question-19: It would be helpful if the authors define what the y-axes of the growth and migration plots represent within the first figure legend where the plots occur. What

is meant by “growth slope (1/hr)” and “migration slope (1/hr)?”

Response: Thanks for your comment. The corresponding growth and migration curves of the histograms were shown in Supplementary figures (for instance, Figure 2b to Supplementary Figure 6b). In these curves (Supplementary Figure 6b), the x-axes referred to time (hours), while the y-axes referred to cell index. For the histograms (Figure 2b), the y-axes referred to the slopes of the curves, and the unit is the derivative of the curve x-axis (1/hour). Please see the figure below from the RTCA manufacturer's manual for your review.

10.5.1.1. Slope

The **Slope** is used to describe the steepness, incline, gradient, or changing rate of a curve within the given time window. The *Slope* parameter may be used for cell proliferation, cell adhesion, cytotoxicity, receptor activation or other cell-based assays. For example, if one is interested in the rate of change of the Cell Index for cells after cytotoxic compound treatment, then Slope could be chosen for such analysis.

For each selected well, the Software calculates the Slope of the Cell Index (or Normalized Cell Index, or Delta Cell Index) curve in the chosen *Time* period, after fitting the points to a straight line. The slope values are shown in the bar chart. Please see the example illustrated in the screenshot below:

Question-20: The authors state, “As anticipated, the levels of newly synthesized R5P via the glucose pentose phosphate pathway significantly decreased within a 1-hour timeframe compared to control cells (Figure 3b).” It should be clarified why this is anticipated.

Response: Thanks for your comment. We have revised the main text accordingly.

Question-21: The statement found in lines 169-172 is unclear. “These findings suggest that PRPS2, but not PRPS1, enzymatically stimulates the utilization of R5P, increases metabolic flux abundance, and enhances downstream end-product ATP/GTP production through the purine biosynthesis pathway.” Fig. 3b suggests PRPS1 also consumes R5P in these conditions. What is meant by “metabolic flux abundance?” Please clarify.

Response: Thanks for your comment. We have revised the main text accordingly.

Question-22: The statement in lines 177-180 should be rephrased. “However, PRPS2 shares 94% amino acid identity with PRPS1, except for three non-conserved residues (V103, G104, E105, referred to as 3AA hereinafter) located in the Loop2 region of PRPS2, which are absent in PRPS1 and represent major variations in their amino acid sequences.” As written, it is ambiguous and suggests that the only difference between PRPS1 and PRPS2 is the 3 amino acid insertion, which is incorrect.

Response: Thanks for your comment. We have revised the main text accordingly.

Question-23: In line 247/248, Fig. 5 is indicated instead of Fig. 6.

Response: Thanks for your comment. We have revised it accordingly.

Question-24: There are arrows in Fig. 6h. What are they indicating?

Response: Thanks for your comment. The black arrows indicated that the temperature point that the protein degradation observed. We have revised the figure legend accordingly.

Question-25: The y-axes in Figure 8 vary in that they do not all show the full scale. It is unclear why the authors sometimes show the full axes (Fig. 8d) and sometimes do not (Fig. 8c).

Response: Thanks for your comment. Given the mass-spec responses on different nucleosides vary a lot, we have to adjust the y-axes accordingly to make the data presented clearly, which is a common presentation way for mass-spec data presentation in publications (*Cell Metab.*, 2022, 34, 106). The y-axis presentation does not change the trend and significance.

Question-26: Supplementary Table 1 contains column labels that are ambiguous. Please define “pT,” “pN,” “M,” and “pTNM stage.”

Response: Thanks for your comment. pT refers to pathological tumor size; pN refers to pathological lymph node; M refers to metastasis; pTNM stage refers to the combined evaluation of pT, pN and M. We have revised Supplementary Table 1 accordingly.

Question-27: Supplementary Figure 5 does not define the dotted lines in the plots. Please update.

Response: Thanks for your comment. We have removed these dotted lines accordingly to make the data clearer.

Question-28: Supplementary Figure 7f: the white dashes on a red/white/blue figure are very difficult to see. Please update.

Response: Thanks for your comment. We have revised the figure accordingly.

Question-29: It is unclear what Supplementary Figure 11h and j are showing; both the PDF on the screen and the printed version just look like blue circles.

Response: Thanks for your comment. We have revised the figure accordingly.

Reviewer #3

Response: Thanks for the positive comments from you and Reviewer #2.

Reviewer #4

This work is clearly worthy of research and interesting both from mechanistic basic science and disease perspectives. A nice integration of structure/function biochemistry and various cell biological approaches, including disease relevant models, constitutes positive strengths of this paper. However, the manuscript in its current form has several issues, as key conclusions are not well backed up by the data shown. Alternative considerations with a more holistic view would also be beneficial. Experimental design and choice of control constructs and rationales should also be brought up to snuff. I hope the authors will do their due diligence to sure up the science. I'd be happy to review a revised paper that addresses the following concerns/queries in due course as Editors see fit.

Response: Thanks for your constructive comments. We strongly agree with your opinion that we should do our due diligence to sure up the science, and this was what we have tried in this work. Indeed, we spent over eight years to look into the detailed mechanisms of PRPS2 in tumorigenesis promotion in this work. It was initiated with the observations from the bioinformatic and clinic data and ended with the enzymatic and structural studies in the first four years. These results piqued our curiosity on how the cancer cells utilize the excess amount of newly synthesized ATP by PRPS2 for tumorigenesis promotion. Hence, we spent another three years to establish mass-spec based research platform as we are not a traditional mass-spec lab, and studied the metabolic flux inside the cancer cell by using stable isotope labeling. Beside the mass-spec data we have shown in the manuscript, we do have looked through the metabolites in multiple metabolic pathways, such as glycolysis, TCA cycle, NADPH metabolic pathway, *etc.*, and all the results suggested that PRPS2 overexpression specifically stimulated SAM synthesis rather than other pathways we monitored. Based on these observations, we spent one more year to further study the enzymatic-independent function of PRPS2 in SAM synthesis regulation and RNA methylation. Although we understand that the deeper we looked into the mechanisms, the more scientific questions coming out, we believe that our efforts have provided valuable mechanistic basic science and disease perspectives to the field. We will definitely keep digging out the regulation mechanisms of this enzyme in cancer development in the near future. We have tired our best to address your concerns as much as possible to strengthen our conclusion, and we hope these efforts would make the work more rigorous and valuable to science.

Major:

Question-1: The model is that PRPS2, and not the more well-studied PRPS1, promotes disease phenotypes through enzymatic and non-enzymatic contributions. The former is claimed to arise through a sustained increase in ATP production enabled by isoform-selective resistance to feedback inhibition (ADP/GDP negative allosteric feedback) along the PPP. The latter moonlighting/non-enzymatic function stems from selective stabilization of the rate-limiting SAM-biosynthesis enzyme MAT2A, claimed to co-opt the overproduced ATP (from the above), to drive upregulated SAM production which then results in the observed disease-phenotypes linked to m⁶A hypermethylation. The model is at first glance stimulating, but in the context of numerous intersecting/inter-dependent pathways that anabolize as well as catabolize these commodity metabolites, the authors do need to take a step back and take a high-level perspective and critique their own interpretations with a more critical set of eyes.

Response: Thanks for your positive comments. We have addressed most of your concerns according to your suggestions by performing additional assays for the Questions 1-3, 6-8, and would like to explain specifically to some of the concerns as following:

(1) The relative flux/rate analysis along glycolytic and mitochondrial TCA-cycle: In this work, we mainly focused on the newly synthesized ATP through the *de novo* purine synthesis pathway, which are distinct from those generated through ADP to ATP conversion in multiple energy supplemental metabolic pathways such as glycolysis, and TCA-cycle *etc.* Specifically, we utilized C¹³ labeled glucose as the carbon source, and traced the C¹³ labeled ribose to evaluate the production of newly synthesized ATP through the *de novo* purine synthesis pathway (Figure 3a). In contrast, ATP molecules derived from ADP to ATP conversion would not be detected in this way as the ribose of these ATP molecules were not be labeled by C¹³. Nevertheless, we do have looked through multiple metabolic pathways, such as glycolysis, TCA cycle, NADPH metabolic pathway, *etc.*, and the results suggested that PRPS2 overexpression specifically stimulated SAM synthesis rather than other pathways we monitored. We also detected the overall ATP levels upon PRPS2 or PRPS1 overexpression vs control, and did not see significant variations.

(2) Knock down/knock out of PRPS2 for knock in based rescue assays: We have performed CRISPR based PRPS2 knock out assay, but we could not obtain any positive colony. We have also performed PRPS2 knock down assays by using shRNA in PC9, H1792 and H1975, while their proliferation upon PRPS2 KD were sharply suppressed and could not be passed for further evaluations as well. Indeed, previous study have demonstrated that KD/KO of PRPS2 is synthetic lethal to cancer cells with c-Myc overexpression that subsequently upregulates PRPS2 expression (*Cell*, 2014, 157, 1088), which is in consistent with our observations. Hence, the knock in based rescue assays could not be carried out successfully. Instead, we overexpressed many functional mutants along with wildtype enzymes to draw biological conclusions.

(3) Enzymatic activity determination in cell lysate, and BrdU/EdU assay for cell proliferation: The cellular metabolite levels and cell proliferation levels was measured

and evaluated with intact cell status in our work, which reflect the real situations inside living cells as much as possible. Breaking cells for enzymatic activity determination or fix cells for BrdU/EdU assay would destroy cell constructions, change the intra-cellular environments, disturb protein-protein interactions for functional studies. The target proteins precipitate immediately and continuously in cell lysate, whose concentration is hard to be quantified for subsequent enzymatic activity determination and comparisons. Additionally, our results have demonstrated that DNA and RNA synthesis did not show significantly differences upon PRPS2 overexpression compared to that of PRPS1 and control (Supplementary Figure 22).

ENZYMATICALLY-DRIVEN PHENOTYPIC OUTPUTS:

Question-2: what are the K_m 's of MAT2A for ATP, and for Met, and approx. cellular concentrations of ATP and Met in relevant OE systems/cell lines being investigated. The relative values of these parameters vs. how much of the increase in ATP is actually being gained from the loss of negative feedback inhibition, do need to be factored in. For instance, with cellular ATP ~10 mM, it is likely well above K_m of MAT2A for ATP, so MAT2A is likely operating on a pseudo first-order mechanism on [Met] with ATP in excess. So, is the role of enzymatic contribution through ATP upregulation feeding into SAM synthesis still functionally relevant for your model? (Engaging in K_m for Met is also relevant to see how critical stabilization of MAT2A really is in attributing to PRPS2 moonlighting: see section further below)

Response: Thanks for your comments. According to the previous studies, the K_m of MAT2A for ATP or Met is 50 μ M or 5 μ M, respectively (*Biochem.*, 2021, 60, 3621). We agree with your concern that the levels of cellular overall ATP and Met are normally maintained at levels beyond the K_m s of MAT2A, no matter the PRPSs overexpressed or not. Indeed, we have detected the overall ATP levels upon PRPS2 or PRPS1 overexpression vs control, and /did not see significant variations. However, our results demonstrated that PRPS1 overexpression did not stimulate SAM synthesis compared to that of PRPS2 or control, indicating that the increase of SAM level was specifically due to the newly synthesized ATP upon PRPS2 overexpression rather than simply driven by the high levels of cellular overall ATP and Met. Our subsequent results further demonstrated that PRPS2 stabilizes MAT2A through protein-protein interactions to stimulate SAM production, which facilitates the utilization of the newly synthesized ATP. Although we could not exclude another possibility that PRPS2-MAT2A further forms larger size protein complex with other potential partner regulators for ATP specific utilization and SAM synthesis stimulation, at least our results demonstrated that both of the enzymatic-dependent and independent function of PRPS2 is required for specific SAM synthesis stimulation.

Question-3: (1) in isotope tracing experiments, the relative flux/rate analysis along glycolytic and mitochondrial TCA-cycle (including assessment of the effects on key rate-limiting enzymes such as ATP synthase), which are the two major sources of cellular ATP production, should be analyzed/quantified to see if some differential effects are being observed through these pathways as opposed to PPP-dependent ATP.

(2) The authors would also need to rule out that overexpression (OE) of certain constructs is not selectively perturbing those primary ATP synthesis pathways in the cell. (3) The authors should use knockdown/knockout of key enzymes in the pathway, as opposed to multiple OE systems to draw biological conclusions. OE is known to be perturbing/invasive to native biology and often introduces artefacts that we are often blind to, and not easily ruled out by WT / mutant OE comparisons.

Response: Thanks for your comments. (1) As we mentioned in our response to your Question-1(1), we mainly focused on the newly synthesized ATP through the *de novo* purine synthesis pathway in this work rather than those generated from ADP to ATP conversion in multiple energy supplemental metabolic pathways. We do have looked through the metabolites in multiple metabolic pathways, such as glycolysis, TCA cycle, NADPH metabolic pathway, *etc.*, and detected the overall ATP levels upon PRPS2 or PRPS1 overexpression vs control, confirming the unique function of PRPS2 in SAM synthesis stimulation by its enzymatic-dependent and independent regulation. (2) We overexpressed PRPS1 and PRPS2 along with many functional mutants, and found that the increase of newly synthesized ATP level was only observed upon PRPS2 overexpression, further confirming the unique function of PRPS2 in SAM synthesis stimulation by its enzymatic-dependent and independent regulation rather than perturbing/invasion of the OE system. (3) Our results suggested that the stimulation of SAM synthesis is a combination effect of PRPS2 enzymatic-dependent and independent regulation. Given the *de novo* purine synthesis pathway is essential for cell development, knock down/knock out of these enzymes would interfere their enzymatic activities for subsequent ATP synthesis, which are not related to PRPS2 enzymatic-independent function.

Question-4: Can exogenous adenosine stimulation, known to raise cellular [ATP] rapidly (within minutes), phenocopy the output (against appropriate control constructs), to further corroborate the overall increased ATP production truly being necessary for the phenotype.

Response: Thanks for your comments. As we mentioned above, we mainly focused on the newly synthesized ATP through the *de novo* purine synthesis pathway in this work rather than those generated from ADP to ATP conversion in multiple energy supplemental metabolic pathways. Although exogenous adenosine would be converted to ATP by ADK kinase in the cell and lead to the increase of cellular ATP level, the phenotypes observed from exogenous adenosine would not directly correlate with PRPSs functions in cancer cells as adenosine is not involved in the *de novo* purine synthesis pathway or salvage pathway (Please see the figure below).

Question-5: (1) Fig.1a and text indicate both ADP/GDP are negative feedback regulators but the loss of feedback loop was only shown for GDP and not for ADP? (2) Furthermore, given the poor catalytic efficiency of PRPS2 [~ 15 fold lower than PRPS1], I assume the authors mean k_{cat}/K_m ? (3) in the context of the cell (even assuming that the loss of feedback inhibition is for both ADP and GDP), is this even sufficient to perturb the large amount of ATP molecules we have in the cell (and also directly needed by MAT2A)? I understand the tracer assay shows time-dependent expected increase but this assay does not easily tell us anything about how much perturbation/relative quantities, are being observed/perturbed against the bulk ATP pools (plus, these tracer assays are highly sensitive to pick up modest changes).

Response: Thanks for your comments. (1) Previous studies have considered both of ADP and GDP as the feedback inhibition regulators for PRPSs. Notably, ADP is also a competitor of ATP, the substrate of PRPSs. The inhibition effect of ADP against PRPS1 would reflect a combination of competing the substrate ATP inside the catalytic pocket as a substrate competitor and binding at the allosteric pocket as an allosteric regulator. To avoid potential confusing, we chose to use GDP instead to ADP specifically for the allosteric inhibition regulation study as GDP does not bind to the catalytic pocket as a substrate competitor. (2) The catalytic efficiencies of PRPSs were evaluate by k_{cat}/K_m values from the kinetics study. We have revised the main text accordingly to make it clearer. (3) The excess amount of newly synthesized ATP due to the lack of PRPS2 feedback inhibition would be hard to evaluate as the generation and consumption of these ATP is a fast and dynamic process. In a short period, for instance, within 1 hour, we detected 150% increase of the newly synthesized ATP compared to that of PRPS1 and control. While within 4 hours, the newly synthesized ATP has been largely utilized, and we did not observe the increase any more.

MOONLIGHTING FUNCTION-ASSOCIATED PHENOTYPIC OUTPUTS:

Question-6: Final section “PRPS2 promotes m⁶A methylation in lung adenocarcinoma”: the punchline on m⁶A modification needs to be strengthened. (1) first, it is not strictly valid, or at least not rigorous, to compare the effects of the involvement of Protein A vs. B, etc., by assessing the outcomes from small-molecule inhibition on Protein A, but by contrast, effects of knockdown on Protein B! Unfortunately, this is what the authors had done for nailing down on METTL isoform! For a start, all small-molecule inhibitors have off-target consequences (many are simply yet-unknown). At least with KO/KD approach, one can minimize off-target effects, using different sets of on-target shRNAs/gRNAs. Thus, the assay where the authors pinpoint the phenotype to a specific METTL isoform/complex should be done using several on-target shRNAs/gRNAs (against several non-targeted sh/gRNA controls) for each relevant METTL isoform. (2) data from this section require comparisons with rescue loop/point mutants as well as enzymatically dead mutants to tease apart enzyme dependent vs independent functions as claimed. Currently the data are limited to comparison between WT PRPS1 and PRPS2 only.

Response: Thanks for your comments. (1) According to your suggestion, we performed additional siRNA knock down assay against either METTL3 or METTL14 (Figure 8e). The results were in consistent with STM2457 treatment results (Supplementary Figure 26), and confirming that the hypermethylation of RNA m⁶A observed in PRPS2 overexpressing cell line was generated specifically through METTL3/14 dependent methylation. (2) According to your suggestion, we detected RNA m⁶A methylation levels upon overexpression of rescue loop/point mutants and enzymatic dead mutants. The results suggested that the dead mutants did not affect RNA m⁶A methylation, while rescue loop/point mutants showed similar phenotypes with wild type enzymes, respectively (Supplementary Figure 23).

Question-7: (1) Moonlighting function involving MAT2A association should be corroborated by “knock-in” of enzymatically-dead mutants of PRPS2 and PRPS1. First, the endogenously present protein variant (even if they may be undetectable on WB) could still bear high enzymatic or non-enzymatic functions, especially if they can confer dominant phenotypes on the overexpressed variant. Thus, reconstituted system in the KO background gives a cleaner dataset in general, without potential interference from native copy. The authors should demonstrate a more solid set of evidence in a non-OE system. (2) These issues concern the following sections: “PRPS2 stabilizes MAT2A in a non-enzymatic manner” “PRPS2 enhances Met metabolic cycle upregulation” data in these sections need to be rigidified by replication and inclusion of enzymatically-dead construct as positive control. (3) Similar to above, TCA and glycolytic flux rates could be analyzed.

Response: Thanks for your comments. (1) As we mentioned in our response to your Question-1(2), we have performed CRISPR based PRPS2 knock out assay, but we could not obtain any positive colony. We have also performed PRPS2 knock down assays by using shRNA, while the proliferation of the PRPS2 KD cells were sharply suppressed and could not be passed for further evaluations as well. Indeed, previous study have demonstrated that KD/KO of PRPS2 is synthetic lethal to cancer cells with

c-Myc overexpression that subsequently upregulates PRPS2 expression (*Cell*, 2014, 157, 1088), which is consistent with our observations. Hence, the knock in based rescue assays could not be carried out successfully. Instead, we overexpressed many functional mutants along with wildtype enzymes to draw biological conclusions. (2) According to the Supplementary Figures 18 and 23, we detected the levels of SAM synthesis and RNA m⁶A methylation, and the results are consistent with our conclusion. (3) As we mentioned above, we mainly focused on the newly synthesized ATP through the *de novo* purine synthesis pathway in this work rather than those generated from ADP to ATP conversion in multiple energy supplemental metabolic pathways. We do have looked through the metabolites in multiple metabolic pathways, such as glycolysis, TCA cycle, NADPH metabolic pathway, *etc.*, and detected the overall ATP levels upon PRPS2 or PRPS1 overexpression vs control, all the results suggested little significant variation and correlations, confirming the unique function of PRPS2 in SAM synthesis stimulation by its enzymatic-dependent and independent regulation.

Question-8: “stabilization of MAT2A directly influences utilization of Met and ATP, as well as synthesis of SAM” –this is not necessarily true but it depends on enzymatic activity of stabilized pools of MAT2A. (1) the authors need to back this up by measuring the relevant activity (in lysate originating from these cells) over a time course of stabilization, against appropriate controls. (2) t_{1/2} of MAT2A needs to be quantified (with appropriate SEM / error analysis/ statistical-treatment), from CHX assays comparing various relevant OE constructs, including catalytically-dead construct. (3) Also does PRPS1 +3AA K153Q mutant fail to bind MAT2A?

Response: Thanks for your comments. (1) As we mentioned above, the cellular metabolite levels were measured and evaluated with intact cell status in our work, which reflect the real situations inside living cells as much as possible. In contrast, breaking cells for enzymatic activity determination would destroy cell constructions, change the intra-cellular environments, disturb protein-protein interactions, and precipitate target proteins immediately and continuously in cell lysate, whose concentration is hard to be quantified for subsequent enzymatic activity determination and comparisons. (2) In CHX assay, MAT2A has been quantified, and the corresponding results have been shown in the raw data submitted. According to your suggestion and our response to Reviewer 2, Question-9, we performed the degradation assay for PRPS1 as the control (Supplementary Figure 17). The results demonstrated that PRPS1 does not stabilize MAT2A. (3) According to your suggestion, we detected the interactions between PRPS1(+3AA/K153Q) and MAT2A, and the results suggested that PRPS1(+3AA/K153Q) mimics the function of PRPS2 (Supplementary Figure 18).

Question-9: Stand-alone claims that are scientifically not valid/correct and it is misleading. Line 159-161: “only PRPS2 OE, but not PRPS1, promoted the synthetic generation of PRPP, indicating accelerated consumption and catalytic utilization of R5P upon PRPS2 ectopic expression (Fig 3C).”; Line 166-167: “again, only PRPS2 OE predominantly enhanced the production of.....”; Line 169-171: “these findings suggest that PRPS2 but not PRPS1 enzymatically stimulates the utilization of R5P, xxxxxx”. We cannot claim any of these points, unless you have shown that enzymatic activity

(normalized by expression level differences, if any) of these two isoforms, under these conditions in these cell lines, is equally roughly equal. These data need to be provided or that this sentences as such above need to be removed. (NOTE: the fact that PRPS1/2's in vitro enzymatic activities, in the absence of feedback regulation, are vastly different, which was described in the immediately following section, makes it even more confusing and worrying for the readers. Given that feedback regulation loss only came up later, the first two sections read as if the logic is not there either). Alternatively, the authors could also input assumptions. Or that the authors may consider restructuring the order and logic of the results section entirely.

Response: Thanks for your comment. We have revised the main text accordingly.

Question-10: Line 209: “PRPS2 did not bind GDP ($K_d > 200 \mu\text{M}$)”– incorrect statement to make such a general bind / no bind decision. Several essential metabolic enzymes have K_d/K_m several tens to hundreds of micromolars but they are still positively and negatively regulated by these dN(D)TPs/N(D)TPs....

Response: Thanks for your comment. We have revised the main text accordingly.

Data presentation-, interpretation-, and analysis-related questions:

Question-11: In all OE lines they created, the authors should do quantitative assessment of expression level differences of various constructs as well as enzymatic activities in cell lysates. This is because it is well known OE induces artefacts – many of the molecules of enzymes do not have the right activity or proper folding. The wild-type, rescue mutants and enzymatic dead mutants should be compared for both enzymatic activity and expression level differences in each overexpressed system before necessary conclusions can be made

Response: Thanks for your comments. The expression levels of protein have been quantified as shown in Supplementary Figure 12a, 12d. Although we agree with your comments that OE may induce artefacts, we performed overexpression of PRPS2 along with that of PRPS1 and empty vector (EV), among which the increase of ATP level was only observed upon PRPS2 overexpression, suggesting that the increase of ATP level was specifically due to PRPS2 overexpression rather than the perturbing/invasion of the OE system (Figure 5e). While improper folding or inactive of these enzymes would not see such variations.

Question-12: Quantification and appropriate statistical treatment across 3 independent triplicates should accompany all blots and all data in reality. That has been not the case in this paper. These are minimum required standards in most peer-reviewed broader-readership journals. Please fix across all Main and SI data / blots shown. (Also, at some point in one of the main fig legends, the authors wrote the following “ the results were repeated (by) 3 times... and one of them was shown here and the other two shown in SI xxx” but this should be explicit across all data.

Response: Thanks for your comments. We have revised the figure legends accordingly.

Question-13: the whole time-courses should be shown for all growth and migration

assays, instead of bar plots of designated 1/h slope (e.g., Fig 2B, E, F).

Response: Thanks for your comments. We have shown the corresponding whole time-courses in Supplementary file. For instance, Figure 2b corresponds to Supplementary Figure 6b, Figure 2g-2i corresponds to Supplementary Figure 6f-6h.

Question-14: BrdU/EdU flow-based assays are more sensitive/reliable for proliferation rate and also more immediate effects can be captured compared to the assays used by the authors: is there a reason why the authors go for phenotypic assessments over longer-duration: is there any difference on cell cycle and DNA replication, especially if the authors are also claiming enhanced/sustained ATP overproduction and NTP pools perturbation?

Response: Thanks for your comments. As we mentioned in our response to your Question-1(3), BrdU assays require specific antibodies, and both BrdU/EdU assays require multiple time-consuming steps including cells fix and many times wash, *etc.* The process is too complicated to obtain reliable results that reflect living cell proliferation status. While the cell proliferation rates we determined by RTCA and Incucyte were against the living cells, which may more accurately reflect the living cell proliferation status. Additionally, our results have demonstrated that DNA and RNA synthesis did not show significantly differences upon PRPS2 overexpression compared to that of PRPS1 and control (Supplementary Figure 22). We have revised the main text accordingly to make it clearer.

Question-15: (1) Fig 6, co-IP blots would benefit from cross comparisons and critical (re)evaluations: e.g., PRPS2 sometimes show multiple bands with similar intensity but in other cases only one major band. The same can be said for MAT2A. Legends make no comments on what appears to be inconsistent behaviors of these proteins. (2) Once again, as mentioned above, all replicates (n=3 independent runs) should be shown openly as supporting full-view blots data and quantification and proper statistical analysis should be provided in all cases, so readers can better understand the extent of run-to-run variability. (3) Furthermore, KO/KD lines should be used to corroborate the expected loss of signal. (4) PRPS1 should be probed in all relevant blots, beyond PRPS2, to rule out isoform specific binding to endogenous MAT2A.

Response: Thanks for your comments. (1) Different batches of PRPS2 and MAT2A antibodies have different efficiencies and non-specific bands may show up in some batches. We only recognize target proteins with correct molecular weights. (2) We have revised the figure legends and submitted the corresponding raw data. (3) As we mentioned above, the PRPS2 KD/KO assays could not be carried out successfully as both of our efforts and previous study have demonstrated that KD/KO of PRPS2 is synthetic lethal to cancer cells with c-Myc overexpression (*Cell*, 2014, 157, 1088). (4) According to our responses to Reviewer 2, Question-3, the commercial PRPS1 or PRPS2 antibodies could not discriminate PRPS1 from PRPS2. While our co-IP results suggested that PRPS1 does not interact with MAT2A in both of HEK293FT and H1299 cells (Supplementary Table 5).

Question-16: Results section and key main Fig 1b began with patient samples being claimed as ‘randomly selected’ but if you zoom into the selected samples marked by arrows, it is quite clear that the selection is far from random but selected for samples that are going in the “right way” as desirable. While it may look more fancy to start with human cancer patients, it’s better the authors make a stronger emphasis on TCGA/GEO database data sets. Alternatively, they should provide data from patient samples that have been genuinely selected in an unbiased way: e.g., pick the samples that correspond to the bands that don’t look so obviously different between ‘normal’ lane and ‘tumor’ lane in the authors’ desirable direction...

Response: Thanks for your comment. We do have selected the samples randomly. Indeed, not all the results shown were going in the “right way” as desirable. For instance, #43, the qPCR results were not in consistence with that of WB. The patient tumor samples were precious and small, we ensured the qPCR assays in a higher priority.

Question-17: On structural data analysis: to be able to compare multiple structures meaningfully, the growth conditions (e.g., soaking vs co-xtalization) being the same and presence of identical substrates/cofactors/metal ions, etc., are necessary for the readers to know if these are even valid comparisons. These parameters should be briefly but clearly mentioned alongside any limitations, for the 3 structures the authors compare from line 173 to 196

Response: Thanks for your comments. All the crystals for three structures obtained by incubating the ligands with protein for co-crystallization under different various conditions. We have showed that the overall structures of three structures were highly similar, and the observations and functions of the major variations around Loop2 region have been further confirmed by using Loop2 switch experiments and subsequent cell-based assays. These results suggested three structures we determined are reliable and the multiple structural comparison are meaningful. We have revised the main text accordingly.

Question-18: Final paragraph of discussion: the authors could make some realistic projections of ways to potentially target PRPS2 in isoform specific manners based on their structure/function/pathway level understanding thus far. Could a specific binder to selectively inhibit PRPS2 leveraging these differential residues/loops they have identified structurally? Alternatively PPI inhibitors to suppress PRPS2-MAT2A association, and several others data-guided proposals could be interesting here? Currently the Discussion is largely a repeat of the later part of the introduction and the end paragraph of Discussion could end on a high note with some specific lines of proposal: currently it is a bit mundane and general. And therapeutic potential is simply thrown in there more like an after-thought...

Response: Thanks for your comments. We have revised the discussion section accordingly.

Question-19: Reading the manuscript, it gives readers a strong sense that results description is not following the order of experimental investigations the authors had

undertaken, which is common for many projects. But I feel that the authors could do a better job of filling in the logical flow and segues between different sections. For instance, currently, some of the enzymatically-dead mutants and +/-3AA etc. rescue mutants come and go in a rather random/illogical manner from one section to the next. All relevant controls should be included and missing comparisons be added across all sections, as stated elsewhere.

Response: Thanks for your comments. The results description was following the order of experimental investigations the authors had undertaken (please see our response to your first comments). According to your suggestion, we added the results of the enzymatically-dead mutants and +/-3AA in moonlighting section (Supplementary Figures 18 and 23). We have revised the main text accordingly.

Additional points:

Question-20: Abstract: the sentence spanning Line no. 35 to 39 in the merged manuscript doc needs an upgrade. The jump to lung tumor came in too abruptly and it sounds as if this mechanism is the single cause. It would be more logical and broader/accessible if readers can be primed up for lung-specific tumor relevant investigations earlier on in the abstract

Response: Thanks for your comments. We have revised the abstract accordingly.

Question-21: Out of curiosity, could the authors include a phylogenetic analysis across several closer species across taxa for PRPS2 and PRPS1. Do these key residues on PRPS2 exist in mice chimps etc – are there lung disease related reports in those models.

Response: Thanks for your comments. We further performed additional sequence alignment against mice and chimps (Please see the Figure below). The results showed that either mice or chimps contain a similar version of human PRPS2 with the three key residues. To the best of our knowledge, we do not see any related disease case reported in these animal models as PRPS2 has been rarely studied even in human.

Question-22: The usage of the word non-conserved vs conserved could be clearer: inter-isoform conservation between PRPS1 and PRPS2 (such as in Abstract), vs. cross-species conservation across orthologs of one isoform across for instance man and mice (such as in line 97 of last paragraph in Introduction).

Response: Thanks for your comments. We have revised the main text accordingly.

Question-23: PRPS3 in human is limited to expression in testicles so this work does not invoke into this isoform. But Ref 31 and 34 may relate to Y-chromosome specific disease, yet they are on PRPS2, instead of PRPS3? Looking at Fig S1, PRPS3 is akin to PRPS1 at least in terms of amino acid sequence homology?

Response: Thanks for your comments. PRPS2 has been found to highly expressed in testicles and ovaries along with PRPS3 specific expression. Given the amino acid sequence of PRPS3 is highly similar to that of PRPS1 (PRPS3 is also named as PRPS1-like protein), we conjectured that PRPS3 would be also regulated by downstream feedback inhibition. Nevertheless, high expression of PRPS2 in testicles and ovaries is reasonable as high level of nucleic acid synthesis is essential for sperm generation and ovum development, where the characteristics of PRPS2 we elucidated in this study may promote these processes.

Question-24: Introduction should briefly engage/elaborate on ‘functional redundancy’: e.g., can KO animals of PRPS1 survive/rescued by PRPS2, and vice versa?

Response: Thanks for your comment. Previous study suggested that knockdown of PRPS1 or PRPS2 in wildtype MEFs cells produced similar decreases in the rates of RNA and DNA production, indicating that PRPS2 and PRPS1 play interchangeable roles in normal cells. In contrast, knockout of PRPS2 is synthetic lethal to cancer cells with c-Myc overexpression as they rely on the specific activity of PRPS2 to sustain their metabolic demands (*Cell*, 2014, 157, 1088). We have revised the introduction accordingly.

Question-25: In OE assays, what is “ctrl” is not clear. Is this empty vector (EV), or OE of a non-native gene? EV is a more appropriate control.

Response: Thanks for your comment. We have revised the main text accordingly.

Question-26: Line 198-200: requires quantitative description – « exhibited switched enzymatic activities “ -- is it now 15-fold more active? Please compare with what you mentioned above (quantitatively). “deletion of the a12 helix from PRPS...” requires clarity – which isoform?

Response: Thanks for your comment. We have revised the main text accordingly.

Question-27: Cell line changes in a few cases are not explained well: HEK293FT and 293FT vs H1299 in section: Line 251 to 271

Response: Thanks for your comment. We have revised the main text accordingly.

Question-28: Writing:

may I also request that the language (including grammatical issues) be proof-checked and edited across the board. Just to give one example: throughout legends “randomly chose” should be “randomly chosen”

- Line 149: delete the word ‘catalytic’ – ‘substrate’ is sufficient

-Line 176: remove the word “catalyzing”

Response: Thanks for your comment. We have revised the main text accordingly.

Reviewer #5

This manuscript by Zhang et al. explores the role and functional contributions of PRPS2, a phosphoribosyl pyrophosphate synthetase isoform involved in the first and rate-limiting step of the purine biosynthesis pathway, in NSCLC. They start by finding that PRPS2 but not PRPS1 is upregulated in human NSCLC samples and that PRPS2 expression correlates with worse survival. They then show that PRPS2>PRPS1 increases growth, migration, and colony formation of NSCLC cell lines. They then do LC-MS and find that PRPS2 overexpression increases GTP/ATP production. Mechanistically, they interestingly find structural differences between PRPS2 and PRPS1 which include a VGE insertion in PRPS2 and K>Q mutation that could explain

specific activity of PRPS2 over PRPS1. Making these mutations in each paralog, they find that mutants of PRPS1 relatively phenocopy PRPS2 and mutants of PRPS2 act like PRPS1 with respect to enzymatic activity, proliferation phenotypes, and migration phenotypes. Moreover, the PRPS2 mutant attenuates tumor growth and the PRPS1 mutant increases tumor growth. They find that PRPS2 knockdown decreases growth. They then also find that PRPS2 binds to MAT2A to promote MAT2A stability which increases SAM leading to m⁶A mRNA methylation through METTL3/14.

This is overall a really strong study particularly the mechanistic studies elucidating structural differences between PRPS2 and PRPS1 that contribute to PRPS2 specific activity. Moreover, the specificity of PRPS2 contributing to m⁶A mRNA methylation is fascinating. I think the mechanistic work is rigorous with appropriate controls to make strong conclusions. I do have a few suggestions to improve the manuscript particularly related to phenotypes associated with suppression of PRPS2.

Response: Thanks for your positive comments and suggestions. We have performed additional assays and revised the manuscript accordingly, and please find the details below for your review.

Question-1: (1) Is PRPS2 required for NSCLC survival in NSCLCs that highly express PRPS2? The authors show in 2 NSCLC cell lines in Fig. 6A-B that suppression of PRPS2 with shRNAs decreases growth of PC9 and H1792 cells. While these phenotypes are significant, it is unclear whether this is an on-target effect of losing PRPS2 or related to an off-target effect. (2) What does CRISPR inactivation of PRPS2 and PRPS1 show in PC9 and H1792 cells? Are they able to express an shRNA or sgRNA resistant cDNA of PRPS2 and rescue the proliferation phenotypes in Figure 6A-B? (3) Lastly, it would be helpful to show both the CRISPR and shRNA experiments in 1-2 additional NSCLC models with high PRPS2 expression and also see if there are differences in the requirement for PRPS2 in models of NSCLC that have low PRPS2 expression. A major conclusion (last line of their abstract and conclusion of their discussion) is that PRPS2 targeting should be considered in cancers associated PRPS2 abnormalities but this aspect of the paper is not strong as is.

Response: Thanks for your comments. (1) We performed additional assay to strengthen our conclusion according to your suggestion (Supplementary Figure 15). The shRNAs knocking down against PRPS2 in H1299 cell line with low endogenous PRPS2 expression level showed little interference for H1299 cell proliferation, confirming that the phenotypes observed in PC9 and H1792 cell lines were on-target effects of losing PRPS2 rather than off-target effects, and suggesting that PRPS2 is required for NSCLC survival in NSCLCs that highly expressing PRPS2. (2) We performed CRISPR based PRPS2 knock out assay, but we could not obtain any positive colony for further evaluations. We then performed PRPS2 knock down assays by using shRNA, while the proliferation of the PRPS2 KD cells were sharply suppressed and could not be passed for further evaluations as well. Indeed, previous study have demonstrated that KD/KO of PRPS2 is synthetic lethal to cancer cells with c-Myc overexpression that subsequently upregulates PRPS2 expression (*Cell*, 2014, 157, 1088), which is in

consistent with our observations. (3) According to your suggestion, we added the results of PRPS2 knock down assays by using shRNA in H1975 cell line (Supplementary Figure 14e-g), which was in consistent with that of PC9 and H1792.

Question-2: m⁶A RNA methylation-It's fascinating that the authors show in Figure 8 increases in m⁶A methylation without major changes in other methylation that utilize SAM. This was only done in H1299 cells. These studies should be expanded to another NSCLC cell line with high PRPS2 expression to see if this specificity is generalizable.

Response: Thanks for your comments. According to your suggestion, we further performed the assay in A549 cell line, and the results were in consistent with the observations in H1299 cell line (Supplementary Figure 19).

Manuscript number: NCOMMS-24-35483B

Editorial comments

Question-1: Please discuss further about the issue with ATP pool (R#1).

Response: Thanks for your comments. We have revised the discussion section to make it clear.

Question-2: Please mention colocalization of PRPS2 and MAT2A instead of specific interaction (R#2).

Response: Thanks for your suggestion. We have revised the manuscript accordingly.

Question-3: Please perform adenosine-administered experiments in APRT native and KO/KD cells to claim salvage-independent pathway, or claims should be removed (R#4).

Response: Thanks for your suggestion. We did not claim the newly synthesized ATP are predominantly from *de novo* pathway rather than the salvage pathway in the manuscript. We apologize for the misunderstanding we introduced by emphasizing the newly synthesized ATP came from the *de novo* pathway in our previous point to point response. We revised the discussion section to describe the technical limitation of ¹³C-labeling isotope tracer experiments in this study accordingly.

Question-4: Please provide transient KD data, or mention limitations of the system in the paper (R#4).

Response: Thanks for your suggestion. We have tried transient KD assay during the experiments, but the results were similar to that of shRNA (lethal to the cancer cell). We have revised the discussion section to describe the limitation of the KO/KD system.

Question-5: Please provide siRNA KD efficiencies of all siRNAs by western blotting (R#4).

Response: Thanks for your suggestion. We have added the western blot validation results in Supplementary Figure 24b and 24d accordingly.

Question-6: Please provide quantification and statistics for fig S17 (R#4).

Response: Thanks for your suggestion. We have added the quantification data to Supplementary Figure 17 accordingly.

Question-7: Provide antibody validation for PRPS1/2 and MAT2A by KD (R#4).

Response: Thanks for your comments. (1) For PRPS2 antibody, we confirmed its specificity by using recombinant purified proteins (Supplementary Figure 2a, 2b), ectopic overexpressed proteins (Supplementary Figure 2c), and knockdown of PRPS2 in H1792, PC9 and H1975 cell lines (Supplementary Figure 14a, c, e). (2) The MAT2A

antibody is a commercial antibody we purchased from “Proteintech” company, and the validation image is available on manufacturers’ website (<https://www.ptgcn.com/products/MAT2A-Antibody-55309-1-AP.htm>). (3) The information of other antibodies we used in this study were commercially available, and we listed their manufacturers’ websites (validation image) in the reporting summary file.

Question-8: Please improve writing of this paper (R#4).

Response: Thanks for your suggestion. We have gone through the manuscript, and improved the writing through the content accordingly.

Question-9: Please mention limitation of the conclusion regarding proliferation phenotype due to lack of orthogonal approaches such as CRISPR or rescue experiments with shRNA resistant cDNAs (R#5).

Response: Thanks for your suggestion. We have revised the discussion section to describe the limitation.

Question-10: Please confirm in your cover letter whether your study is compliant with the "Guidance of the Ministry of Science and Technology (MOST) for the Review and Approval of Human Genetic Resources", which requires formal approval for the export of human genetic material or data from China.

Response: Thanks for your comment. We have confirmed it in the cover letter.

Reviewer #1

My opinion is still very positive. One of my two questions has been answered, namely the preferential use of newly synthesized SAM via the lower K_m value of *mettl3/14* versus other MTAses.

Question-1: However, even the revised discussion does not provide a clear rationale (nor does the rebuttal letter) wrt to other ATP-depleting physiological events that should, according to the present rationale, overrule the entire pathway when lowering ATP levels. What would happen to RNA methylation when oxphos is lowered by an uncoupling agent? would that inhibit cancer growth, and if not, why? The authors have added the term "newly synthesized" before ATP in many places throughout the manuscript, but that does not yet heal the conceptual problem. Are there indications that the "newly synthesized" ATP is funneled towards SAM synthesis without entering the generally available pool?

Response: Thanks for your comments. It is really a complex relationship and context-dependent manner between lower ATP levels induced by nutrient deficiency/hypoxia/oxphos dysfunction-related signals and RNA m^6A regulating pathways in cancers. For instance, a recent study reported that hypoxia increases mRNA m^6A level in breast cancer cells to increase the cell proliferation and migration (*Oncotarget*, 2018, 9, 31231); However, other studies found that hypoxia decreases RNA m^6A levels in breast

cancer cells and lung adenocarcinoma cells to increase the cell proliferation, respectively (*Proc. Natl. Acad. Sci. U.S.A.*, 2016, 113, E2047; *Biochem. Biophys. Res. Commun.*, 2020, 521, 499). While starvation-induced FTO upregulation led to the decrease of RNA m⁶A modification and increase of PD-1 expression in melanoma cells, and thereby promoting melanoma tumorigenesis and anti-PD-1 therapy resistance (*Nat. Commun.*, 2019, 10, 2782). Additionally, many cancer cells compensate for the loss of ATP to adapt the metabolic stress through increasing glycolytic flux, and thus continuing to fuel cell growth and proliferation despite reduced mitochondrial function (*Nat. Rev. Cancer*, 2010, 10, 235; *J. Exp. Clin. Oncol.*, 2020, 39, 232). These debated observations suggest that stress-mediated RNA m⁶A modification exhibits cell context-dependent manner, and are various in each scenario.

We used the words “newly synthesized ATP” to differentiate different ATP originations between those generated through the purine synthesis pathway that we studied and those generated from multiple energy supplemental metabolic pathways. We did not claim in the manuscript that the “newly synthesized” ATP is funneled towards SAM synthesis without entering the generally available pool. Indeed, we agree the fact that these ATP are a part of the general ATP pool for the global regulation and utilization during cell proliferation. The ¹³C-labeling isotope tracer experiment we used in this study could only trace the ATP generated from R5P, and limited us to study the global changes of the ATP pool. Our results more focused on the enzymatic-dependent and independent functions of PRPS2 in the purine synthesis pathway in cancer development rather than the global ATP pool regulation. We have revised the discussion section accordingly to describe the limitation of this method to make it rigorous.

Reviewer #2

Zhang and colleagues have addressed almost all of my concerns. There is one point where small revisions to the text are needed.

Question-11: The authors show immunostaining experiments in Supp. Fig. 13e, stating that the results show colocalization of PRPS2 and MAT2A. However, in the images provided, both proteins look like they are expressed throughout the cell, suggesting overlap is coincidental rather than the result of a specific interaction.

Response: Thanks for your comments. The purpose of our immunostaining experiments was to check the localization of PRPS2 and MAT2A, as the localization of PRPS2 in cell is unclear so far. Both of our mass-spec and co-IP results confirmed that PRPS2 interacts with and stabilizes MAT2A.

Question-1: Reviewer Response: While other experiments within this paper suggest that PRPS2 and MAT2A interact, the immunofluorescence-based colocalization assay data (now Supplementary Figure 16) only shows that both proteins are distributed throughout the cytoplasm. This experiment does not “corroborat[e] the Co-IP results and indicat[e] a specific interaction between PRPS2 and MAT2A” (lines282-284). If the authors are claiming that both PRPS2 and MAT2A can be found in the cytoplasm

together, they should state this, rather than claiming colocalization as a support for direct interaction. Three sentences should be amended to reflect that both proteins localize to the cytoplasm together and this data should not be used to argue for “specific interactions:”

- Supplementary Figure 16. Cell proliferation of PRPS1 shRNA knockdown in H1792 and PC9 cell lines and **localization** of PRPS2 and endogenous MAT2A in H1299 cells.

- (e) Immunofluorescence assay showing the **location** of PRPS2 and endogenous MAT2A from H1299 cells transfected with FLAG-PRPS2.

- Subsequent immunostaining assays demonstrated that **transfected** PRPS2 and **endogenous** MAT2A **localize** to the cytoplasm (Supplementary Figure 16e).

Any other reference to this data should also be amended accordingly.

Response: Thanks for your constructive comments. We have revised the manuscript accordingly.

Reviewer #3

Response: Thanks for your time and comments in reviewing our work.

Reviewer #4

The authors have addressed some of my concerns. There are a few pending questions that should be addressed comprehensively for clarity as well as rigor and also to rigidify some of the conceptual underpinnings and to help tie up the loose ends. Providing all these points below can be thoroughly remedied, I would be pleased to recommend publication. I do understand that not all data and assays may necessarily support the conclusions and models, but nonetheless, it would be necessary that alternative possibilities be ruled out and data be interpreted with reservations, care, and rigor, outlining limitations wherever applicable, and showing all acquired datasets openly etc.

Question-1: the authors made several implications that ¹³C experiments only report on the newly-synthesized ATP arising exclusively from de novo pathway; and also that externally introduced adenosine would not affect the de novo or salvage pathway, and so on. These statements above are limited/incorrect. Referring to a commonly-encountered text-book figure (reproduced below is such a figure that appears in a recent publica on, 11th July 2024 Tran et al) illustrating both de novo and salvage pathway, the authors' detected ¹³C amount at the m/z corresponding to ATP, could still arise from

pathways that are not exclusively *de novo* ATP synthesis pathway. For instance, nothing is preventing, in the authors' experimental conditions/cells deployed, for the first set of newly-made ¹³C-labeled ATP molecules, to be transformed into adenosine (see green arrow in figure, via nucleotidase, thereby reducing the ultimate measured amount of total 'newly-made' ¹³C labeled ATP molecules); nothing is also preventing the so-produced ¹³C labeled adenosine to also be reconverted, to some unknown extent, to ¹³C labeled AMP and thus ATP molecules, thus (re)increasing or altering the ultimate measured total ¹³C labeled ATP molecules; and nothing is also preventing the so-made ¹³C-labeled Adenosine to ¹³C-labeled Adenine that can enter salvage pathway (see orange arrow) to some unknown degree, to reconstitute to ¹³C-labeled AMP and thus altering the detected ¹³C labeled ATP being measured in a way that is unknown.....

Indeed these issues are often intrinsic limitations of isotope tracer experiments, but from the rebuttal responses provided in Page 11 and 13 of the rebuttal PDF, it makes me worry that the authors are unaware of these points first of all.

Thus Adenosine is indeed able to influence the newly-synthesised ATP from both *de novo* and salvage pathway. If one increases adenosine by exogenous addition, for instance, it will raise the cellular ATP levels and should phenocopy what the authors are reporting (my previous Q4). To nail down that this is salvage pathway independent they can easily perform these adenosine-administered experiments in APRT native and KO/KD cells see if phenotypes are replicated.

Response: Thanks for your constructive comments. We agree with your comments that the newly-made ATP could be influence from both *de novo* and salvage pathways. Indeed, we did not claim the newly synthesized ATP are predominantly from *de novo* pathway rather than the salvage pathway in the manuscript. Our results in Figure 3d and 3e have also shown that overexpression of PRPS2 enhances both synthetic levels of the major intermediate SAICAR (M+5) in the *de novo* purine synthesis pathway, as well as the downstream product inosine monophosphate (IMP) (M+5) from both *de novo* and salvage purine synthesis pathways. We apologize for the misunderstanding we introduced by emphasizing the newly synthesized ATP came from the *de novo* pathway in our previous point to point response, as we were trying to explain the newly synthesized ATP were predominantly from the purine synthesis pathway (both *de novo* and salvage) rather than those generated from multiple energy supplemental metabolic pathways. We revised the discussion section to describe the technical limitation of ¹³C-labeling isotope tracer experiments in this study accordingly.

Question-2: potential interference of endogenous wild-type protein variant that could impose dominant negative effects differentially on overexpressed wt and mutant constructs remains a concern. This is not eliminated by the revisions due to inability to obtain KD/KO lines of this protein. To this end: I wonder if the authors have attempted transient KD by siRNA, instead of near-complete KO or integrated shRNA KD.

Should transient KD still result in complete lack of any level of KD achievable in these cells, then this issue above resulting in entirely overexpression-focused studies, should be mentioned in the paper. Reasons behind why KD/KO lines are inaccessible by both

lentiviral CRISPR/gRNA, shRNA as well as siRNAs, also should be briefly engaged in the discussion as part of the usual ‘ limitations of the study ‘ paragraph..

More generally, some of the limitations and assumptions outlined in the rebuttal should be integrated into the manuscript to help the ultimate readers.

Likewise, for the choice of the use of GDP over ADP should be briefly explained in the relevant results section.

Similarly, technical limitations in measuring the changes in the amount of newly-made ATP in the absence of PRPS2 feedback inhibition should be mentioned.

Response: Thanks for your comments. We have also tried transient KD assay by using siRNA during the experiments, but the results were similar to that of shRNA. According to the suggestions from you and the editor (Question-4), we revised the discussion section to describe the technical limitation of the conclusion regarding proliferation phenotype due to lack of orthogonal approaches including lentiviral CRISPR/gRNA, shRNA as well as siRNAs, and revised the results section to explain the choice of the use of GDP over ADP.

Question-3: siRNA KD efficiencies of all siRNAs used in fig 8E should be measured using western blot and results shown and quantified and reported in Supplemental Info, against ctrl-siRNA-administered and non-siRNA-administered native cells

Response: Thanks for your comments. We have added these western blot validation results in Supplementary Figure 24b and 24d accordingly.

Question-4: my Q8 previously asked for quantification of the bands since the changes are minute / modest. The authors have not yet quantified nor provided the quantitative changes to the half-life of protein (derived from appropriate half-life ($t_{1/2}$) curve fitting) under different conditions. Please quantify the bands across multiple independent replicates along with appropriate statistical treatments for data in both A and B of Supp Fig S17

Response: Thanks for your comments. We have added the quantification data to Supplementary Figure 17 accordingly.

Question-5: it is an important due diligence that all antibodies be validated by knockdown/knockout unless they have been previously validated as such by either previous data in the published literature, or by the image data available on manufacturers’ websites. Especially that there are multiple bands, and the authors responded by saying that these are due to non-specific bands: how does one 100% sure that any non-specific bands do not accidentally appear on the same/similar MW/kD as the protein being probed. This coincidental issue is far too common for many antibodies (where the band of correct MW fails to disappear after targeted KO/KD!) and has resulted in a lot of data misinterpretations of western blot data with numerous antibodies. Thus the authors should thus do the due diligence of KO/KD validations of Abs that report on their key enzymes and isoforms at least PRPS1/2 and MAT2A, key proteins essential for their mechanistic model. This is the only valid way to verify the outcomes.

Response: Thanks for your comments. (1) For PRPS2 antibody, we confirmed its specificity by using recombinant purified proteins (Supplementary Figure 2a, 2b), ectopic overexpressed proteins (Supplementary Figure 2c), and knockdown of PRPS2 in H1792, PC9 and H1975 cell lines (Supplementary Figure 14a, c, e). (2) The MAT2A antibody is a commercial antibody we purchased from “Proteintech” company, and the validation image is available on manufacturers’ websites (<https://www.ptgcn.com/products/MAT2A-Antibody-55309-1-AP.htm>). (3) The information of other antibodies we used in this study were commercially available, and we listed their manufacturers’ websites (validation image) in the reporting summary file.

Question-6: please also provide full-view blots of all blots shown as supplementary information, which is not an uncommon practice in biological publications.

Response: Thanks for your comments. We have submitted the raw data to the journal system according to the Nature group publication policy, which will be published as “Source Data” with the manuscript in the journal.

Question-7: WRITING: writing improvement was requested previously (Q28)-- it would be necessary that professional proof-editor be deployed to help improve the writing, grammatical correctness and idiomatic use of the language across the board. There are several places where language used could lead to incorrect interpretations: just as one-of-many examples, the authors consistently use in the manuscript text and rebuttal document, the word ‘in’ mistakenly, whenever they tend to mean that something is (“in”) consistent with something else. Should the space be accidentally missing between “in” and “consistent”, this would mean the complete opposite of what they intend to describe!

Response: Thanks for your comments. We have gone through the manuscript and improved the writing through the content accordingly.

Question-8: I must say I do not quite understand why the authors replicated their responses to the referees in a repeated manner (in some cases already 3 times: see, for instance “We do have looked through the metabolites in multiple metabolic pathways, such as glycolysis, TCA cycle, NADPH metabolic pathway, etc.,” on page 11, page 13, and page 16 – one would wonder why?). In general, what was already said in Page 11 (and indeed read by this reviewer with great care and interest) was repeatedly described in subsequent pages, yet once again. This makes the rebuttal look doubly lengthy and of greater importance, it is not helpful and confusing to say the least for the reviewers to sieve through new information if any against repeating texts...

I am also unsure if no. of years spent on a project is a valid yard-stick to justify to what extent the work is rigorous. I understand that the authors have done a large extent of work but it is better that the responses be centered upon scientific points that can be integrated to the manuscript where necessary.

On the flip side, it makes the readers worry that after spending over 8 yrs into this project, why then the authors did not think about clearly possible solutions that would

render a lot of these investigations / results to be much more solid: such as, on the multiple reviewers' overlapping questions, where the response has been: "commercial PRPS1 or PRPS2 antibodies could not discriminate PRPS1 from PRPS2".... >8 yrs is several folds more than sufficient for the authors to raise isoform-specific antibodies in-house or have them be custom-made.

Response: Thanks for your comments. We agree with your comment that numbers of years spent on a project is a valid yard-stick to justify the quality of the work. In our previous corresponding responses, we were trying to explain that we do have tried our best to make this work as more rigorous as we can (e.g. eight years efforts, having looked through the data related to the metabolites in multiple metabolic pathways, etc). We tried to use these words to show our attitude in agreement of your comments "do our due diligence to sure up the science". During this work, we do have collaborated with several companies to generate specific antibody for PRPS1 or PRPS2, but the collaborations had to be halted multiple times during years due to the global pandemic issue unfortunately, and we had to use what we obtained (PRPS2 antibody) to push forward our study. We apologize for any misunderstanding we introduced in our previous point to point responses due to our inaccurate descriptions.

Reviewer #5

The authors have substantially improved their manuscript and addressed some of my critiques. I still think the manuscript could be improved with more rigorous genetic experiments aimed at determining whether PRPS2 is actually a dependency in NSCLC lines. I originally asked for orthogonal approaches such as CRISPR or doing rescue experiments with shRNA resistant cDNAs of PRPS2 to make sure their proliferation phenotypes were on-target. PC9 is a cell line that is very easy to CRISPR and therefore these experiments should be feasible.

Response: Thanks for your comments. Our efforts and previous study (Cell, 2014, 157, 1088) have shown that the traditional orthogonal approaches, including lentiviral CRISPR/gRNA, shRNA as well as siRNAs, for PRPS2 knockout/knockdown are lethal to these cell lines with high endogenous PRPS2 expression levels. To exclude the potential off-target effects, we performed shRNA knockdown assay against PRPS2 in H1299 cell line with low endogenous PRPS2 expression level (Supplementary Figure 15). The results suggested that the knockdown of PRPS2 in H1299 cell line does not interfere the cell proliferation, confirming that PRPS2 is indeed on-target by shRNA and dependency in cell lines with high endogenous PRPS2 expression level, such as NSCLC. According to the related suggestions from the editor (Question-4, 9) and Reviewer-4 (Question-2), we also revised the discussion section to describe the technical limitation of the conclusion regarding proliferation phenotype due to lack of orthogonal approaches.

The authors have addressed some of my concerns. There are a few pending questions that should be addressed comprehensively for clarity as well as rigor and also to rigidify some of the conceptual underpinnings and to help tie up the loose ends. Providing all these points below can be thoroughly remedied, I would be pleased to recommend publication. I do understand that not all data and assays may necessarily support the conclusions and models, but nonetheless, it would be necessary that alternative possibilities be ruled out and data be interpreted with reservations, care, and rigor, outlining limitations wherever applicable, and showing all acquired datasets openly etc.

a)- the authors made several implications that ¹³C experiments only report on the newly-synthesized ATP arising exclusively from de novo pathway; and also that externally introduced adenosine would not affect the de novo or salvage pathway, and so on. These statements above are limited / incorrect. Referring to a commonly-encountered text-book figure (reproduced below is such a figure that appears in a recent publication, 11th July 2024 Tran et al) illustrating both de novo and salvage pathway, the authors' detected ¹³C amount at the m/z corresponding to ATP, could still arise from pathways that are not exclusively de novo ATP synthesis pathway. For instance, nothing is preventing, in the authors' experimental conditions/cells deployed, for the first set of newly-made ¹³C-labeled ATP molecules, to be transformed into adenosine (see green arrow in figure, via nucleotidase, thereby reducing the ultimate measured amount of total 'newly-made' ¹³-C labeled ATP molecules); nothing is also preventing the so-produced ¹³-C labeled adenosine to also be reconverted, to some unknown extent, to ¹³-C labeled AMP and thus ATP molecules, thus (re)increasing or altering the ultimate measured total ¹³-C labeled ATP molecules; and nothing is also preventing the so-made ¹³-C-labeled Adenosine to ¹³-c-labeled Adenine that can enter salvage pathway (see orange arrow) to some unknown degree, to recontribute to ¹³-c-labeled AMP and thus altering the detected ¹³-C labeled ATP being measured in a way that is unknown.....

Indeed these issues are often intrinsic limitations of isotope tracer experiments, but from the rebuttal responses provided in Page 11 and 13 of the rebuttal PDF, it makes me worry that the authors are unaware of these points first of all.

Thus Adenosine is indeed able to influence the newly-synthesised ATP from both de novo and salvage pathway. If one increases adenosine by exogenous addition, for instance, it will raise the cellular ATP levels and should phenocopy what the authors are reporting (my previous Q4). To nail down that this is salvage pathway independent they can easily perform these adenosine-administered experiments in APRT native and KO/KD cells see if phenotypes are replicated.

b) potential interference of endogenous wild-type protein variant that could impose dominant negative effects differentially on overexpressed wt and mutant constructs remains a concern. This is not eliminated by the revisions due to inability to obtain KD/KO lines of this protein. To this end: I wonder if the authors have attempted transient KD by siRNA, instead of near-complete KO or integrated shRNA KD.

Should transient KD still result in complete lack of any level of KD achievable in these cells, then this issue above resulting in entirely overexpression-focused studies, should be mentioned in the paper. Reasons behind why KD/KO lines are inaccessible by both lentiviral CRISPR/gRNA, shRNA as well as siRNAs, also should be briefly engaged in the discussion as part of the usual ' limitations of the study ' paragraph..

More generally, some of the limitations and assumptions outlined in the rebuttal should be integrated into the manuscript to help the ultimate readers.

Likewise, for the choice of the use of GDP over ADP should be briefly explained in the relevant results section.

Similarly, technical limitations in measuring the changes in the amount of newly-made ATP in the absence of PRPS2 feedback inhibition should be mentioned.

c) siRNA KD efficiencies of all siRNAs used in fig 8E should be measured using western blot and results shown and quantified and reported in Supplemental Info, against ctrl-siRNA-administered and non-siRNA-administered native cells

d) my Q8 previously asked for quantification of the bands since the changes are minute / modest. The authors have not yet quantified nor provided the quantitative changes to the half-life of protein (derived from appropriate half-life ($t_{1/2}$) curve fitting) under different conditions. Please quantify the bands across multiple independent replicates along with appropriate statistical treatments for data in both A and B of Supp Fig S17

e) it is an important due diligence that all antibodies be validated by knockdown/knockout unless they have been previously validated as such by either previous data in the published literature, or by the image data available on manufacturers' websites. Especially that there are multiple bands, and the authors responded by saying that these are due to non-specific bands: how does one 100% sure that any non-specific bands do not accidentally appear on the same/similar MW/kD as the protein being probed. This coincidental issue is far too common for many antibodies (where the band of correct MW fails to disappear after targeted KO/KD!) and has resulted in a lot of data misinterpretations of western blot data with numerous antibodies. Thus the authors should thus do the due diligence of KO/KD validations of Abs that report on their key enzymes and isoforms at least PRPS1/2 and MAT2A, key proteins essential for their mechanistic model. This is the only valid way to verify the outcomes.

f) please also provide full-view blots of all blots shown as supplementary information, which is not an uncommon practice in biological publications.

g) WRITING: writing improvement was requested previously (Q28)-- it would be necessary that professional proof-editor be deployed to help improve the writing, grammatical correctness and idiomatic use of the language across the board. There are several places where language used could lead to incorrect interpretations: just as one-of-many examples, the authors consistently use in the manuscript text and rebuttal document, the word 'in' mistakenly, whenever they tend to mean that something is ("in") consistent with something else. Should the space be accidentally missing between "in" and "consistent", this would mean the complete opposite of what they intend to describe!

I must say I do not quite understand why the authors replicated their responses to the referees in a repeated manner (in some cases already 3 times: see, for instance "We do have looked through the metabolites in multiple metabolic pathways, such as glycolysis, TCA cycle, NADPH metabolic pathway, etc.," on page 11, page 13, and page 16 – one would wonder why?). In general, what was already said in Page 11 (and indeed read by this reviewer with great care and interest) was repeatedly described in subsequent pages, yet once again. This makes the rebuttal look doubly lengthy and of greater importance, it is not helpful and confusing to say the least for the reviewers to sieve through new information if any against repeating texts...

I am also unsure if no. of years spent on a project is a valid yard-stick to justify to what extent the work is rigorous. I understand that the authors have done a large extent of work but it is better that the responses be centered upon scientific points that can be integrated to the manuscript where necessary.

On the flip side, it makes the readers worry that after spending over 8 yrs into this project, why then the authors did not think about clearly possible solutions that would render a lot of these investigations / results to be much more solid: such as, on the multiple reviewers' overlapping questions, where the response has been: "commercial PRPS1 or PRPS2 antibodies could not discriminate PRPS1 from PRPS2".... >8 yrs is several folds more than sufficient time for the authors to raise isoform-specific antibodies in-house or have them be custom-made.